# Toward Scalable and Valid Conditional Independence Testing with Spectral Representations

**Alek Fröhlich** [1 2]   **Vladimir Kostić** [1 3]   **Karim Lounici** [4]   **Daniel Perazzo** [1 2]   **Daniel Tiezzi** [5]   **Massimiliano Pontil** [1 6]

## Abstract

Conditional independence (CI) is central to causal inference, feature selection, and graphical modeling, yet it is untestable in many settings without additional assumptions. Existing CI tests often rely on restrictive structural conditions, limiting their validity. Kernel methods using partial covariance operators offer a more principled approach but suffer from limited adaptivity and scalability. In this work, we explore whether representation learning can help address these limitations. Specifically, we focus on representations derived from the singular value decomposition of partial covariance operators and use them to construct a simple test statistic. We also introduce a bi-level contrastive algorithm to learn these representations. Our theory links representation learning error to test performance and establishes asymptotic validity and power guarantees. Experiments on real and synthetic data suggest that this approach offers a principled and statistically grounded path toward scalable CI testing, bridging kernel-based theory with modern representation learning.[1]

## 1. Introduction

Given random variables $X$, $Y$, and $Z$, the goal of conditional independence (CI) testing is to decide between

$$\mathcal{H}_0 : X \perp\!\!\!\perp Y \mid Z \quad \text{vs.} \quad \mathcal{H}_1 : X \not\perp\!\!\!\perp Y \mid Z, \qquad (1)$$

using an i.i.d. sample from their joint distribution $P_{X,Y,Z}$. CI is a fundamental concept in statistics (Dawid, 1979) and

machine learning (Schölkopf et al., 2021), underlying causal inference (Pearl, 2009; Spirtes et al., 2001; Imbens & Rubin, 2015), graphical models (Lauritzen, 1996; Koller & Friedman, 2009), and variable selection (Candès et al., 2018; Huang et al., 2022). A concrete example arises in computational pathology when integrating multi-modal data $(X, Z)$ to predict patient outcomes $Y$. Let $X$ represent a tumor's molecular profile and $Z$ its visual (histological) features. Tumor phenotype is known to reflect the underlying molecular state (Thennavan et al., 2021). As a result, while $X$ may correlate with $Y$, this association can be redundant if the histological patterns encoded in $Z$ already capture the same biological signal. A CI test determines whether $X$ offers incremental predictive power for $Y$ beyond the information already contained in the image features $Z$.

Despite its far-ranging applications, CI testing in nonparametric settings is known to be fundamentally challenging. Shah & Peters (2020) established that any test controlling type I error uniformly over all conditionally independent distributions lacks power against *any* alternative, helping explain why existing CI tests can fail to control type I error in practice. The difficulty stems from the counterintuitive fact that any conditionally dependent sample can be approximated arbitrarily well by one that is conditionally independent.[2] In other words, CI distributions are so rich that even strongly dependent data could have arisen from a CI model, unless structural assumptions are imposed on $P_{X,Y,Z}$, for example that $P_{X,Y|Z=z}$ varies continuously with $z$. This stands in stark contrast to the unconditional case, for which there exist uniformly valid tests, even in finite samples (Hoeffding, 1948a; Berrett & Samworth, 2019).

This realization has motivated a shift from designing universally valid tests to methods tailored for particular settings. Kernel-based tests such as **KCIT** (Zhang et al., 2011) and **RCIT** (Strobl et al., 2019) rely on regressions from $Z$ to $X$ and $Y$, remaining valid as long as these regressions can be learned sufficiently well (Shah & Peters, 2020; He et al., 2025; Pogodin et al., 2025). Model-x tests such as **GCIT** (Bellot & van der Schaar, 2019) or **DGCIT** (Shi et al., 2021) assume access to $P(X|Z)$, remaining valid when this distribution or a reliable approximation is available (Katsevich

---

[1]CSML, Istituto Italiano di Tecnologia, Genoa, Italy [2]DIBRIS, University of Genoa, Genoa, Italy [3]Faculty of Science, University of Novi Sad, Novi Sad, Serbia [4]CMAP, École Polytechnique, Palaiseau, France [5]Breast Disease Division, University of São Paulo, Ribeirão Preto, Brazil [6]Department of Computer Science, University College London, London, UK. Correspondence to: Alek Fröhlich <alek.frohlich@iit.it>.

*Proceedings of the 43rd International Conference on Machine Learning*, Seoul, South Korea. PMLR 306, 2026. Copyright 2026 by the author(s).

[1]Code repository: https://github.com/alekfrohlich/SCIT.

---

[2]Cf. (Neykov et al., 2021, Lemmas 3.2 & A.1).

*Figure 1.* **SpectralCIT**'s testing pipeline. First the features $\widehat{u}_\theta, \widehat{v}_\theta, \widehat{w}_\theta$ are learned using Algorithm 1 with whitening over the training set. Then, the test statistic $\widehat{T}_n$ is computing using Eq. (10) over the test set. Finally, a decision is made using the $1 - \alpha$ quantile of the chi-squared distribution with $d^2$ degrees of freedom, where $d$ is the output dimension of the networks.

& Ramdas, 2022; Zhang et al., 2025). Local permutation tests such as **NNLSCIT** (Li et al., 2023) generate empirical p-values by permuting samples with clustered/binned $Z$ values, but incur substantial computational cost and depend on smoothness assumptions (Kim et al., 2022). We defer a broader discussion on existing methods to Section 6.

Among these approaches, classical kernel-based tests stand out for modeling conditional dependence through the partial covariance operator, whose properties implicitly encode a wide range of structural assumptions on the joint distribution of $(X, Y, Z)$, including smoothness, sparsity, low-rank, and latent variable models. Despite the generality of this operator-based framework, its practical impact has remained limited due to the lack of adaptivity and scalability of kernel methods (Pogodin et al., 2025; Ramdas et al., 2015).

**From kernels to representation learning.** Recently, methods based on learning leading spectral features of statistical operators have shown promise across nonparametric inference tasks, including causal effect estimation (Sun et al., 2025; Wang et al., 2022; Meunier et al., 2025b;a), reinforcement learning (Hu et al., 2024), and learning dynamical systems (Turri et al., 2026), while remaining simple and scalable due to their connection to contrastive learning. Motivated by the above examples and classical kernel CI tests, in this paper we explore whether spectral representation learning can address the limitations of kernel methods for scalable and valid CI testing. In particular, we provide a contrastive learning algorithm to learn the leading spectral features of the partial covariance operator, using them to construct a simple test statistic reminiscent of the Hilbert-Schmidt Independence Criterion (Gretton et al., 2005). We then analyze the behavior of the test statistic under both null and alternative, proving that it converges asymptotically to a chi-squared distribution under the null, and achieves power under the alternative. Our approach is illustrated in Fig. 1.

**Contributions.** In summary, our main contributions are:
**i.** We introduce a simple, scalable algorithm that learns the leading spectral features of the partial covariance operator, overcoming the adaptivity and scalability bottlenecks of classical kernel CI tests.
**ii.** We provide a comprehensive theoretical analysis of CI testing using the learned representations in combination with a simple test statistic. This includes type I error and power guarantees along with the characterization of the null distribution as asymptotically chi-squared.
**iii.** We validate our theory on challenging real and synthetic data, including a novel nonsmooth, high-dimensional variant of the post-nonlinear model and multi-modal breast cancer data (The Cancer Genome Atlas Network, 2012), showing that our approach offers a principled and statistically grounded path toward scalable CI testing, bridging kernel-based theory with modern representation learning.

**Paper organization.** In Section 2, we introduce notation and review statistical hypothesis testing, covariance operators, and spectral representation learning. Section 3 presents an algorithm for learning partial covariance operators. In Section 4, we present our operator framework for CI testing alongside our main theoretical results. Section 5 reports numerical experiments. Section 6 reviews related work. All proofs are deferred to the appendix.

## 2. Background

**Data spaces.** $X, Y, Z$ take values in measurable spaces $(\mathcal{X}, \mathcal{F}_\mathcal{X})$, $(\mathcal{Y}, \mathcal{F}_\mathcal{Y})$, $(\mathcal{Z}, \mathcal{F}_\mathcal{Z})$ with marginals $P_X, P_Y, P_Z$ and joint distributions $P_{X,Y}, P_{X,Z}, P_{Y,Z}, P_{X,Y,Z}$. In particular, $X, Y, Z$ might be continuous, discrete, or mixed random vectors of arbitrary dimensions $d_X, d_Y, d_Z$. Throughout the paper, we take $\ddot{Y} = (Y, Z)$ and assume $P_{XYZ} \ll P_X \times P_{YZ}$, $P_{XZ} \ll P_X \times P_Z$ and $P_{YZ} \ll P_Y \times P_Z$, where $P \ll Q$ denotes $P$ is absolutely continuous w.r.t. $Q$.

**Statistical hypothesis testing.** We are given a sample $\mathcal{D}_n = \{(X_i, Y_i, Z_i)\}_{i=1}^n$ and seek to decide between a null hypothesis $\mathcal{H}_0$ and an alternative hypothesis $\mathcal{H}_1$. This decision is based on a *test statistic* $\widehat{T}_n = T(\mathcal{D}_n)$, with large values of $\widehat{T}_n$ typically providing evidence against $\mathcal{H}_0$. The test rejects $\mathcal{H}_0$ whenever $\widehat{T}_n \geq c_\alpha$ where the *critical value* $c_\alpha$ is chosen so that the *Type I error*—the probability of incorrectly rejecting $\mathcal{H}_0$ when it is true—is controlled at a pre-

scribed significance level $\alpha \in (0, 1)$: $\mathbb{P}_{\mathcal{H}_0}(\widehat{T}_n \geq c_\alpha) = \alpha$.

Under the alternative, the *power* of the test is the probability of correctly rejecting $\mathcal{H}_0$: Power $= \mathbb{P}_{\mathcal{H}_1}(\widehat{T}_n \geq c_\alpha)$. The goal is to design tests that achieve high power while controlling the Type I error at level $\alpha$. In nonparametric settings, this trade-off is particularly challenging due to the broad class of possible alternatives. Our objective is thus to develop conditional independence tests that rigorously control Type I error and retain competitive power in a wide range of structural scenarios. We refer to Appendix B.1 for more details on statistical hypothesis testing.

**Function spaces.** For a random variable $A$, $L^2(A)$ denotes the space of square-integrable functions ($\mathbb{E}[f(A)^2] < \infty$).

**Operators on Hilbert spaces.** Let $\mathcal{F}, \mathcal{G}$ be Hilbert spaces. For a bounded operator $\mathsf{T} : \mathcal{G} \to \mathcal{F}$, we denote $\|\mathsf{T}\|$ its operator norm, $\|\mathsf{T}\|_{\mathrm{HS}}$ its Hilbert-Schmidt norm, and $\mathsf{T}^*$ its adjoint. Given a second operator $\mathsf{S} : \mathcal{G} \to \mathcal{F}$, we denote $\langle \mathsf{T}, \mathsf{S} \rangle_{\mathrm{HS}} = \mathrm{tr}(\mathsf{T}\mathsf{S}^*)$ the Hilbert-Schmidt inner product. For matrices, the Hilbert-Schmidt norm coincides with Frobenious norm, which we denote $\|M\|_{\mathrm{F}}$ for a matrix $M$. For $f \in L^2(A), g, h \in L^2(B)$, the rank-one operator $f \otimes g$ is defined as $[f \otimes g](h) = \langle g, h \rangle f$, generalizing $fg^\top$ for vectors $f, g$.

**Covariance operators.** For random variables $A, B$ such that $P_{AB}$ is absolutely continuous with respect to $P_A \times P_B$, we define the cross-covariance operator as the unique bounded linear operator satisfying[3]

$$\langle f, \Sigma_{AB} g \rangle = \mathbb{E}\big[f(A)g(B)\big] - \mathbb{E}\big[f(A)\big]\mathbb{E}\big[g(B)\big], \quad (2)$$

for every $f \in L^2(A), g \in L^2(B)$. Covariance operators generalize (centered) covariance matrices $C_{AB} = \mathbb{E}(A - \mathbb{E}A)(B - \mathbb{E}B)^\top$ and characterize independence ($A \perp\!\!\!\perp B$ iff $\|\Sigma_{AB}\|_{\mathrm{HS}} = 0$). Given a third random variable $C$, and assuming $\Sigma_{AB}, \Sigma_{AC}, \Sigma_{CB}$ are well-defined, we define the partial cross-covariance operator as[4]

$$\Sigma_{AB \cdot C} = \Sigma_{AB} - \Sigma_{AC}\Sigma_{CB}. \quad (3)$$

Partial covariance operators generalize partial covariances and characterize conditional independence ($A \perp\!\!\!\perp B | C$ iff $\|\Sigma_{A(B,C) \cdot C}\|_{\mathrm{HS}} = 0$; (Strobl et al., 2019)).

**Singular value decomposition (SVD).** A compact operator between Hilbert spaces $\mathsf{T} : \mathcal{G} \to \mathcal{F}$ can be written as $\mathsf{T} = \sum_{i=1}^\infty \sigma_i \phi_i \otimes \psi_i$, where $\sigma_1 \geq \sigma_2 \geq \cdots \geq 0, \sigma_i \to 0$, are scalars (singular values) and $\phi_i, \psi_j$ are orthonormal functions (singular functions) in $\mathcal{F}, \mathcal{G}$. Furthermore, by

---

[3]This is a direct consequence of the Riesz representation theorem (Reed & Simon, 1981, Theorem II.4).

[4]This definition differs from RKHS-based formulations involving $\Sigma_{CC}^{-1}$ (e.g. (Strobl et al., 2019)). Working directly on $L^2$ spaces, $\Sigma_{CC}$ acts as the identity on the subspace of centered functions, eliminating the need for an explicit inverse.

Eckart-Young-Mirsky's theorem (Eckart & Young, 1936), the best rank-$d$ approximation of $\mathsf{T}$ with respect to any unitarily invariant norm is given by $[\![\mathsf{T}]\!]_d = \sum_{i=1}^d \sigma_i \phi_i \otimes \psi_i = \Phi \Sigma \Psi^\star$, where $\Phi = [\phi_1| \cdots |\phi_d] : \mathbb{R}^d \to \mathcal{F}$ and $\Psi = [\psi_1| \cdots |\psi_d] : \mathbb{R}^d \to \mathcal{G}$ act via $\Phi \alpha = \sum_{i=1}^d \alpha_i \phi_i$ and $\Psi \beta = \sum_{i=1}^d \beta_i \psi_i$, and $\Sigma = \mathrm{diag}(\sigma_1, \ldots, \sigma_d)$.

**Spectral representation learning.** To learn the rank-$d$ truncated SVD of a compact covariance operator $\Sigma_{AB}$, Kostic et al. (2024) have shown that it suffices to minimize the following regularized loss for $\gamma > 0$

$$\widehat{\mathcal{L}}_\gamma(\theta) = -\frac{2}{m} \sum_{i=1}^m \langle \overline{u}_\theta(A_i), M_\theta \overline{v}_\theta(B_i) \rangle^2$$

$$+ \frac{1}{m(m-1)} \sum_{i \neq j}^m \langle \overline{u}_\theta(A_i), M_\theta \overline{v}_\theta(B_j) \rangle + \gamma \, \widehat{\Omega}(\theta),$$

where $\{(A_i, B_i)\}_{i=1}^m$ is a batch sampled i.i.d. from $P_{AB}$, $\overline{u}_\theta, \overline{v}_\theta$ are neural nets whose outputs' have been empirically centered ($\overline{u}_\theta(A_i) = u_\theta(A_i) - \frac{1}{m}\sum_{j=1}^m u_\theta(A_j)$), $M_\theta$ is a matrix, and $\widehat{\Omega}(\cdot)$ is orthonormality regularization

$$\widehat{\Omega}(\theta) = \|\widehat{C}_{U_\theta U_\theta} - I_d\|_F^2 + \|\widehat{C}_{V_\theta V_\theta} - I_d\|_F^2.$$

The connection with the SVD of $\Sigma_{AB}$ follows from the Eckart–Young–Mirsky theorem (cf. Theorem A.1). In particular, minimizing

$$\mathcal{L}_\gamma(\theta) := \|\Sigma_{AB} - \mathsf{U}_\theta \mathsf{M}_\theta \mathsf{V}_\theta^*\|_{\mathrm{HS}}^2 - \|\Sigma_{AB}\|_{\mathrm{HS}}^2 + \gamma \, \Omega(\theta)$$

yields the rank-$d$ truncated SVD of $\Sigma_{AB}$: $\widehat{u}_{\theta,i}$ and $\widehat{v}_{\theta,i}$, $i \in [d]$, correspond to the left and right singular functions of $\Sigma_{AB}$, while $\mathsf{M}_\theta$ is the diagonal matrix containing the associated singular values. The empirical loss $\widehat{\mathcal{L}}_0(\theta)$ is simply a U-statistic estimator of $\mathcal{L}_0(\theta)$ (Kostic et al., 2024; Sun et al., 2025).

Prior work on spectral representation learning focuses on estimation and uncertainty quantification, rather than on independence or conditional independence testing. Extending this framework to CI testing is nontrivial: the partial covariance operator $\Sigma_{A(B,C) \cdot C}$ involves residualization with respect to $C$, which is not directly observable from data and must be handled implicitly. This fundamentally changes the representation learning problem and leads to the bilevel formulation introduced in Section 3. As a result, both the representation learning stage and the subsequent statistical analysis in Section 4 differ substantially from existing literature.

## 3. Learning Partial Covariance Operators

In this section, we describe a algorithm for learning the leading spectral features of the partial covariance operator $\Sigma_{X\ddot{Y} \cdot Z}$ associated with a random triple $(X, Y, Z)$, using

an i.i.d. sample $\mathcal{D}_m^{(\text{train})} = \{(X_i, Y_i, Z_i)\}_{i=1}^m$ from their joint distribution $P_{X,Y,Z}$. Following Kostic et al. (2024), we make the following mild regularity assumption for the SVD of $\Sigma_{X\ddot{Y}\cdot Z}$ to be well-defined.

**Assumption 3.1.** $\Sigma_{X\ddot{Y}\cdot Z} : L^2(\ddot{Y}) \to L^2(X)$ *is a compact operator.*

This assumption holds for a large class of discrete and continuous distributions, including those without densities with respect to the Lebesgue measure. A sufficient condition is given by the Radon–Nikodym derivatives $\kappa_{X,\ddot{Y}} = \frac{dP_{X,\ddot{Y}}}{d(P_X \times P_{\ddot{Y}})}$ and $\kappa_{X,Z} = \frac{dP_{X,Z}}{d(P_X \times P_Z)}$ being square-integrable: $\mathbb{E}_{P_{X,\ddot{Y}}}[\kappa_{X,\ddot{Y}}(X,\ddot{Y})^2], \mathbb{E}_{P_{X,Z}}[\kappa_{X,Z}(X,Z)^2] < \infty$.

**Variational formulation of SVD.** Given $d \in \mathbb{N}$, our aim is to learn the best rank-$d$ approximation of $\Sigma_{X\ddot{Y}\cdot Z}$, that is,

$$[\![\Sigma_{X\ddot{Y}\cdot Z}]\!]_d = \sum_{i=1}^d \sigma_i u_i \otimes v_i, \tag{4}$$

where $\sigma_i \geq 0$, $u_i \in L^2(X)$, and $v_i \in L^2(\ddot{Y})$ are the leading singular values and the corresponding left and right singular functions of $\Sigma_{X\ddot{Y}\cdot Z}$, respectively.

As in Section 2, we characterize (4) via a variational formulation. Let $U = [u_1(X), \ldots, u_d(X)]^\top$, $V = [v_1(\ddot{Y}), \ldots, v_d(\ddot{Y})]^\top$, and $\mathsf{M} = \text{diag}(\sigma_1, \ldots, \sigma_d)$, and denote by $\mathsf{U} = [u_1 | \cdots | u_d] : \mathbb{R}^d \to L^2(X)$ and $\mathsf{V} = [v_1 | \cdots | v_d] : \mathbb{R}^d \to L^2(\ddot{Y})$ the associated feature operators. The truncated SVD of $\Sigma_{X\ddot{Y}\cdot Z}$ can then be expressed as

$$\begin{aligned}(\mathsf{U}, \mathsf{M}, \mathsf{V}) \in \underset{\tilde{\mathsf{U}}, \tilde{\mathsf{M}}, \tilde{\mathsf{V}}}{\arg\min}\, &\text{tr}\Big[ C_{\tilde{U}\tilde{U}} \tilde{\mathsf{M}} C_{\tilde{V}\tilde{V}} \tilde{\mathsf{M}}^\top \Big] \\ &- 2\,\text{tr}\Big[ C_{\tilde{U}\tilde{V}} \tilde{\mathsf{M}}^\top \Big] \\ &+ 2\,\text{tr}\Big[ \tilde{\mathsf{M}}^\top \tilde{\mathsf{U}}^* \Sigma_{XZ} \Sigma_{Z\ddot{Y}} \tilde{\mathsf{V}} \Big], \end{aligned} \tag{5}$$

where the covariance matrices are centered. In contrast to the unconditional setting, the final term in (5) involves a composition of covariance operators, $\Sigma_{XZ} \Sigma_{Z\ddot{Y}}$, and cannot be directly estimated from data. *This constitutes the main technical challenge in learning $\Sigma_{X\ddot{Y}\cdot Z}$.*

**Low-rank auxiliary problem.** To overcome this, we use the cyclic property of the trace and observe that the operator $\Sigma_{Z\ddot{Y}} \tilde{\mathsf{V}} \tilde{\mathsf{M}}^\top \tilde{\mathsf{U}}^* \Sigma_{XZ} : L^2(Z) \to L^2(Z)$ is of rank at most $d$. Therefore, its symmetrization is of rank at most $2d$, and hence admits a singular value decomposition of the form $\mathsf{WNW}^*$, where $\mathsf{W} = [w_1 | \cdots | w_{2d}] : \mathbb{R}^{2d} \to L^2(Z)$ for an orthonormal system $w_i \in L^2(Z)$ and $\mathsf{N}$ is a diagonal matrix with nonnegative entries. As the trace is linear and invariant under transposition, applying the same variational principle as in Section 2 gives

$$2\,\text{tr}\Big[ \tilde{\mathsf{M}}^\top \tilde{\mathsf{U}}^* \Sigma_{XZ} \Sigma_{Z\ddot{Y}} \tilde{\mathsf{V}} \Big] = \text{tr}\big[ \mathsf{WNW}^* \big], \tag{6}$$

where $(\mathsf{W}, \mathsf{N})$ solves the inner optimization problem

$$\begin{aligned}(\mathsf{W}, \mathsf{N}) \in \underset{\tilde{\mathsf{W}}, \tilde{\mathsf{N}}}{\arg\min}\, &\text{tr}\Big[ \tilde{\mathsf{N}} C_{\tilde{W}\tilde{W}} \tilde{\mathsf{N}}^\top C_{\tilde{W}\tilde{W}} \Big] \\ &- \text{tr}\Big[ (\tilde{\mathsf{N}}^\top + \tilde{\mathsf{N}}) C_{\tilde{W}\tilde{U}} \tilde{\mathsf{M}} C_{\tilde{V}\tilde{W}} \Big], \end{aligned} \tag{7}$$

enabling us to compute the last term in (5) via (6). Solving the bi-level optimization[5] problem obtained by combining Eqs. (5) and (7) yields the representation

$$[\![\Sigma_{X\ddot{Y}\cdot Z}]\!]_d = \mathsf{U}[C_{UV} - C_{UW} C_{WV}]\mathsf{V}^*. \tag{8}$$

This expression highlights that the conditional dependence structure between $X$ and $Y$ given $Z$ is captured by the matrix $C_{UV} - C_{UW} C_{WV}$, provided that appropriate representations $(U, V, W) = (u(X), v(\ddot{Y}), w(Z))$ are used. A derivation of the above bi-level formulation is given in Appendix A.

**Spectral representation learning.** Using the bi-level variational formulation of the truncated SVD of the partial cross-covariance operator in (8), we extend the spectral contrastive algorithm of Kostic et al. (2024) to learn the spectral features $u, v, w$. We parametrize these features with neural networks $u_\theta : \mathcal{X} \to \mathbb{R}^d$, $v_\theta : \mathcal{Y} \times \mathcal{Z} \to \mathbb{R}^d$, $w_\theta : \mathcal{Z} \to \mathbb{R}^{2d}$, and minimize empirical losses derived from Eqs. (5) and (7) using U-statistics (Hoeffding, 1948b). The full-batch losses are shown below.

$$\begin{aligned}\widehat{\mathcal{L}}_{\text{out}} = &\frac{1}{m(m-1)} \sum_{i \neq j}^m \langle \overline{u}_i, M\overline{v}_j \rangle^2 - \frac{2}{m} \sum_{i=1}^m \langle \overline{u}_i, M\overline{v}_i \rangle \\ &+ \frac{2}{m(m-1)} \sum_{i \neq j}^m \langle \overline{u}_i, M\overline{v}_j \rangle \langle \overline{w}_i, \overline{w}_j \rangle. \end{aligned}$$

$$\begin{aligned}\widehat{\mathcal{L}}_{\text{in}} = &\frac{1}{m(m-1)} \sum_{i \neq j}^m \langle \overline{w}_i, N\overline{w}_j \rangle^2 \\ &- \frac{2}{m(m-1)} \sum_{i \neq j}^m \langle \overline{u}_i, M\overline{v}_j \rangle \langle \overline{w}_i, N\overline{w}_j \rangle, \end{aligned}$$

where we've written $\overline{u}_i = \overline{u}_\theta(X_i)$, $\overline{v}_j = \overline{v}_\theta(\ddot{Y}_j)$, and $\overline{w}_k = \overline{w}_\theta(Z_k)$, $M = M_\theta$, $N = (N_\theta + N_\theta^\top)/2$, where $\ddot{Y} = (Y, Z)$ and $\bar{\cdot}$ denotes centering. The corresponding representation learning algorithm is shown in Algorithm 1.

**Orthonormality regularization.** We also employ orthonormality (whitening) regularization with strength $\gamma > 0$, e.g.,

$$\begin{aligned}\widehat{\Omega}_{\text{out}}(\theta) &= \|\widehat{C}_{U_\theta U_\theta} - I_d\|_{\text{F}}^2 + \|\widehat{C}_{V_\theta V_\theta} - I_d\|_{\text{F}}^2, \\ \widehat{\Omega}_{\text{in}}(\theta) &= \|\widehat{C}_{W_\theta W_\theta} - I_{2d}\|_{\text{F}}^2. \end{aligned}$$

---

[5] For a refresher on bi-level optimization, see (Franceschi et al., 2025) and references therein.

---

**Algorithm 1** Bi-level spectral representation learning

---

**Input:** train set $\mathcal{D}_m^{(\text{train})}$; regularization strength $\gamma > 0$
*# Learn representations*
**for** for $t = 1, \ldots, n_{\text{steps}}$ **do**
  *# Step inner model*
  **for** $s = 1, \ldots, n_{\text{steps\_inner}}$ **do**
    Sample mini-batch $\mathbf{B}$ from $\mathcal{D}_m^{(\text{train})}$
    $g_{\text{in}} \leftarrow \nabla_\theta [\widehat{\mathcal{L}}_{\text{in}}(\theta, \mathbf{B}) + \gamma \, \widehat{\Omega}_{\text{in}}(\theta, \mathbf{B})]$
    Update $w_\theta$ with $g_{\text{in}}$
  **end for**

  *# Step outer model*
  Sample mini-batch $\mathbf{B}'$ from $\mathcal{D}_m^{(\text{train})}$
  $g_{\text{out}} \leftarrow \nabla_\theta [\widehat{\mathcal{L}}_{\text{out}}(\theta, \mathbf{B}') + \gamma \, \widehat{\Omega}_{\text{out}}(\theta, \mathbf{B}')]$
  Update $u_\theta, v_\theta$ with $g_{\text{out}}$
**end for**

*# Whiten representations*
$\widehat{u}_\theta, \widehat{v}_\theta, \widehat{w}_\theta \leftarrow \widehat{C}_{\widetilde{U}_\theta \widetilde{U}_\theta}^{-1/2} \widetilde{u}_\theta, \widehat{C}_{\widetilde{V}_\theta \widetilde{V}_\theta}^{-1/2} \widetilde{v}_\theta, \widehat{C}_{\widetilde{W}_\theta \widetilde{W}_\theta}^{-1/2} \widetilde{w}_\theta$
**Output:** estimated leading spectral features $\widehat{u}_\theta, \widehat{v}_\theta, \widehat{w}_\theta$

---

We use orthonormality regularization to ensure that empirical covariance matrices remain well-conditioned and can be safely inverted during whitening.

**Whitening post-processing.** Let $\widetilde{u}_\theta, \widetilde{v}_\theta, \widetilde{w}_\theta$ denote the features learned by Algorithm 1. Since orthonormality is enforced only at the batch level and competes with other objectives, we apply an additional post-processing step to ensure that $\{\widehat{u}_{\theta,i}\}_{i=1}^d$, $\{\widehat{v}_{\theta,j}\}_{j=1}^d$, and $\{\widehat{w}_{\theta,k}\}_{k=1}^{2d}$ are empirically orthonormal. Specifically, we whiten $\widetilde{u}_\theta(X)$, $\widetilde{v}_\theta(\ddot{Y})$, and $\widetilde{w}_\theta(Z)$ via $\widehat{u}_\theta(X) = \widehat{C}_{\widetilde{U}_\theta \widetilde{U}_\theta}^{-1/2} \widetilde{u}_\theta(X)$ (and analogously for $\widehat{v}_\theta$ and $\widehat{w}_\theta$), where covariance matrices are estimated over the entire training set. This transformation preserves the learned subspaces (e.g., $\text{range}(\widehat{U}_\theta) = \text{range}(\widetilde{U}_\theta)$) while improving the geometry of the basis functions, yielding empirically orthonormal representations.

## 4. CI Testing with Spectral Representations

We begin this section by recasting nonparametric CI testing in terms of the partial covariance operator:

$$\mathcal{H}_0 : \|\Sigma_{X\ddot{Y} \cdot Z}\|_{\text{HS}}^2 = 0 \text{ vs. } \mathcal{H}_{1,n,d} : \|\Sigma_{X\ddot{Y} \cdot Z}\|_{\text{HS}}^2 \geq \epsilon_n. \quad (9)$$

Under $\mathcal{H}_0$, we consider all distributions $P_{X,Y,Z}$ for which $\Sigma_{X\ddot{Y} \cdot Z}$ is well-defined and vanishes. Under $\mathcal{H}_{1,n,d}$, we consider local alternatives consisting of distributions $P_{X,Y,Z}$ whose partial covariance operator is well-defined and has Hilbert-Schmidt norm at least $\epsilon_n$, where the separation threshold $\epsilon_n$ may decay with the test sample size $n$ at a rate that will be described in Theorem 4.2. The dependence

on $d$ comes from the requirement that $\sigma_d > \sigma_{d+1} \geq 0$ and $u_i(X), v_i(\ddot{Y}), w_j(Z)$ be sub-Gaussian for $i \in [d], j \in [2d]$.

In light of the no-free-lunch theorem of Shah & Peters (2020), no test can distinguish $\mathcal{H}_0$ from $\mathcal{H}_{1,n,d}$ uniformly fast. Rather than imposing restrictive structural conditions–such as Lipschitzness of $P_{X|Z=z}$ and $P_{Y|Z=z}$ w.r.t. $z$ as in (Neykov et al., 2021; Kim et al., 2022)–we adopt a more practical perspective, framing test validity and power in terms of the quality of the learned representations $\widehat{u}_\theta, \widehat{v}_\theta, \widehat{w}_\theta$ obtained via Algorithm 1. Formally, we measure this via:

$$\mathcal{E}_m^{\text{val}} = \max \Big\{ \|C_{\widehat{U}_\theta \widehat{U}_\theta} - I_d\|, \|C_{\widehat{V}_\theta \widehat{V}_\theta} - I_d\|,$$
$$\|C_{\widehat{W}_\theta \widehat{W}_\theta} - I_{2d}\| \Big\},$$
$$\mathcal{E}_m^{\text{pow}} = \| [\![\Sigma_{X\ddot{Y} \cdot Z}]\!]_d - \mathsf{U}_\theta \mathsf{M}_\theta \mathsf{V}_\theta^* \|.$$

As shown in Appendix B, under the null the validity of our test is governed by the central limit theorem applied to the empirical partial covariance $\widehat{C}_{\widehat{U}_\theta \widehat{V}_\theta} - \widehat{C}_{\widehat{U}_\theta \widehat{W}_\theta} \widehat{C}_{\widehat{W}_\theta \widehat{V}_\theta}$ computed from the learned features. In particular, validity holds as long as $\mathcal{E}_m^{\text{val}}$ is sufficiently small. Compared to regression-based tests (Shah & Peters, 2020; He et al., 2025), our requirement is less restrictive, as it does not rely on estimating conditional expectations at a prescribed rate. This is consistent with the classical distinction between testing and estimation (Ingster, 1993): more precisely, rather than solving a full regression problem, our approach only requires capturing a low-dimensional spectral subspace of $\Sigma_{X\ddot{Y} \cdot Z}$. Under the alternative, power is governed by $\mathcal{E}_m^{\text{pow}}$, which measures how well the learned representations capture $[\![\Sigma_{X\ddot{Y} \cdot Z}]\!]_d$. Thus, $\mathcal{E}_m^{\text{val}}$ controls calibration under the null, while $\mathcal{E}_m^{\text{pow}}$ controls signal retention under the alternative.

We now introduce our proposed CI test, **SpectralCIT**.

**Spectral conditional independence test.** Let $N = m + n$ and let $\{(X_i, Y_i, Z_i)\}_{i=1}^N$ be an i.i.d. sample from $P_{X,Y,Z}$, split into train and test sets,

$$\mathcal{D}_m^{(\text{train})} = \{(X_i, Y_i, Z_i)\}_{i=1}^m, \mathcal{D}_n^{(\text{test})} = \{(X_i, Y_i, Z_i)\}_{i=m+1}^N.$$

**SpectralCIT** works as follows. First, Algorithm 1 is applied to $\mathcal{D}_m^{(\text{train})}$ to learn the feature maps $\widehat{u}_\theta, \widehat{v}_\theta$, and $\widehat{w}_\theta$. Using these representations and the test set $\mathcal{D}_n^{(\text{test})}$, we compute the test statistic

$$\widehat{T}_n = n \, \|\widehat{C}_{\widehat{U}_\theta \widehat{V}_\theta} - \widehat{C}_{\widehat{U}_\theta \widehat{W}_\theta} \widehat{C}_{\widehat{W}_\theta \widehat{V}_\theta}\|_F^2. \quad (10)$$

**Validity of testing with spectral features.** The statistical guarantees are derived under the following mild regularity assumptions, which can be ensured by choosing bounded activation functions (e.g., `Tanh`).

**Assumption 4.1.** *Let* $\|\widehat{u}_\theta(X)\|, \|\widehat{v}_\theta(\ddot{Y})\|$ *and* $\|\widehat{w}_\theta(Z)\|$ *be* $K$*–sub-Gaussian random variables.*

**Theorem 4.1** (Validity). *Let Assumption 4.1 be satisfied. Assume in addition that $\mathcal{E}_m^{\text{val}} \to 0$ as $m \to \infty$. Then under the null hypothesis, $\widehat{T}_n = n \, \|\widehat{C}_{\widehat{U}_\theta \widehat{V}_\theta} - \widehat{C}_{\widehat{U}_\theta \widehat{W}_\theta} \widehat{C}_{\widehat{W}_\theta \widehat{V}_\theta}\|_F^2$ converges in distribution to a chi-square distribution with $d^2$ degrees of freedom as $m, n \to \infty$.*

Theorem 4.1 shows that $\widehat{T}_n \xrightarrow{d} \chi^2(d^2)$ as $m, n \to \infty$ as long as $\mathcal{E}_m^{\text{val}} \to 0$ as $m \to \infty$. Intuitively, after whitening the learned features are approximately orthonormal, so $\widehat{C}_{\widehat{U}_\theta \widehat{V}_\theta} - \widehat{C}_{\widehat{U}_\theta \widehat{W}_\theta} \widehat{C}_{\widehat{W}_\theta \widehat{V}_\theta}$ has approximately identity covariance structure under the null, making $\widehat{T}_n$ approximately a sum of $d^2$ squared independent standard Gaussians (see Appendix B.7). Therefore, for a significance level $\alpha \in (0,1)$, the method rejects the null hypothesis whenever $\widehat{T}_n \geq c_\alpha$, where $c_\alpha = q_{1-\alpha}(\chi^2(d^2))$ denotes the $1 - \alpha$ quantile of the $\chi^2(d^2)$ distribution.

**Power of testing with spectral features.** Theorem 4.2 characterizes the minimum signal strength $\epsilon_n$ required for our method to reliably detect conditional dependence under the alternative hypothesis.

**Theorem 4.2** (Power). *Let Assumption 4.1 be satisfied. Let $\delta \in (0,1)$ and take $d$ large enough such that $\sum_{j=1}^d \sigma_j^2 \geq \|\Sigma_{X\ddot{Y}\cdot Z}\|_{\text{HS}}^2/2$. Assume that $\epsilon_n^2 \geq c\left(d\,(\mathcal{E}_m^{\text{pow}})^2 + \frac{d^2 + d\log(\delta^{-1})}{n}\right)$ for some large enough numerical constant $c > 0$. Then for $\mathcal{H}_{1,n,d}$ defined in Eq. (9), we have $\mathbb{P}_{\mathcal{H}_{1,n,d}}\left(\widehat{T}_n > c_\alpha\right) \geq 1 - \delta$.*

## 5. Experiments

In this section, we report results from numerical experiments on challenging synthetic and real-world datasets. In Section 5.1, we benchmark a range of existing CI tests against our proposed method under the post-nonlinear model (Yang et al., 2025; Scetbon et al., 2022; Zhang et al., 2011; Li et al., 2023). In Section 5.2, we apply our approach to further investigate the role of visual (histological) features in breast tumor progression (Li et al., 2024).

### 5.1. Synthetic Data

**Benchmark on post-nonlinear model.** We first evaluate the type I error and power of **SpectralCIT** against established baselines: **KCIT** (Zhang et al., 2011), **RCIT** (Strobl et al., 2019), **GCIT** (Bellot & van der Schaar, 2019), **DGCIT** (Shi et al., 2021), and **NNLSCIT** (Li et al., 2023) using the post-nonlinear model of Yang et al. (2025):

$$\mathcal{H}_0 : X = f(\bar{Z} + \varepsilon_X/4), \, Y = g(\bar{Z} + \varepsilon_Y/4),$$
$$\mathcal{H}_1 : X = f(\bar{Z} + \varepsilon_X/4) + \varepsilon/2, \, Y = g(\bar{Z} + \varepsilon_Y/4) + \varepsilon/2,$$

where $\bar{Z} = \frac{1}{d_Z}\sum_{i=1}^{d_Z} Z_i$, and $Z_i, \varepsilon_X, \varepsilon_Y, \varepsilon$ are i.i.d. standard Gaussian for $i \in [d_Z]$. The nonlinearities are $f(w) =$

$w^3$ and $g(w) = \tanh(w)$. We fix the sample size $N = 1000$ and vary the conditioning dimension $d_Z \in [50, 300]$. Each setting is repeated 100 times. Hyperparameter selection is described in Appendix C.

Results in Fig. 2 at significance level $\alpha = 0.05$ show that **KCIT**, **RCIT**, and **GCIT** achieved high power, but failed to control for type I errors. **DGCIT** performed substantially worse, completely losing type I error control. In contrast, **NNLSCIT** and **SpectralCIT** consistently maintained robust type I error control while achieving high power across all dimensions, consistent with the findings in Li et al. (2023).

**On the role of structural assumptions.** A key advantage of our approach is its ability to adapt to diverse structural scenarios. To illustrate this, we evaluate the type I error of our method (**SpectralCIT**) against **NNLSCIT** using a nonsmooth, high-dimensional data model defined as:

$$X = f(Z/2 + \varepsilon_X), \quad Y = g(Z/2 + \varepsilon_Y),$$

where $Z$, $\varepsilon_X$, and $\varepsilon_Y$ are sampled independently from $N(0, I_{100})$. We define the nonlinearities $f(w) = h_2(w)$ and $g(w) = h_3(w)$ to be highly oscillatory near the origin:

$$h_k(w) = \begin{cases} w^k & \text{for } |w| \geq 1, \\ \cos\left(\frac{2\pi}{w}\right) & \text{for } 0 < |w| < 1, \\ 1 & \text{for } w = 0. \end{cases}$$

We set the dimensionality of $Z$ to 100 and vary the sample size from 500 to 1500. Each setting is repeated 100 times. Hyperparameters are the same as in the previous experiment.

Results are reported in Fig. 3 with a significance level of $\alpha = 0.05$. Notably, the highly oscillatory behavior of $f$ and $g$ near $w = 0$ creates a lack of smoothness on the conditional distributions that violates the core assumptions of **NNLSCIT** (Kim et al., 2022), leading to a complete breakdown of its type I error control. In contrast, our operator-based approach remains robust.

**Overcoming limitations of kernel methods.** Finally, we evaluate the scalability and adaptivity of **SpectralCIT** against established kernel-based CI tests, including **KCIT** (Zhang et al., 2011), **RCIT** (Strobl et al., 2019), **LPCIT** (Scetbon et al., 2022), and **GCM** (Shah & Peters, 2020). Data is generated according to (Shi et al., 2021):

$$X = \sin(a_X^\top Z + \varepsilon_X/2), \, Y = \cos(a_Y^\top Z + bX + \varepsilon_Y/2).$$

Here, $Z$, $\varepsilon_X$, and $\varepsilon_Y$ are i.i.d. standard Gaussian and the entries of $a_{(\cdot)}$ are sampled uniformly from $[0, 1]$ and then normalized to unit $\ell_1$ norm. The parameter $b$ controls the degree of conditional dependence: $b = 0$ corresponds to the null hypothesis $\mathcal{H}_0$, while $b \neq 0$ implies $\mathcal{H}_1$. We set the sample size to $N = 1000$ and vary the dimensionality of $Z$, $d_Z$, from 50 to 300. Each setting is repeated 100 times. Hyperparameters are selected following Appendix C.

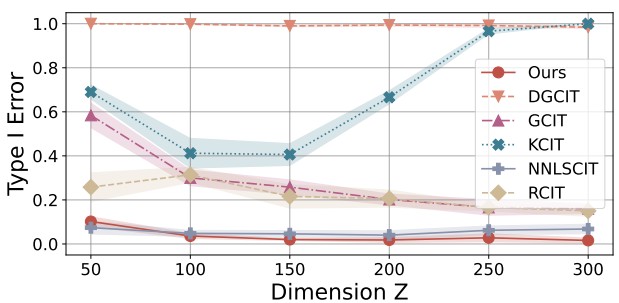
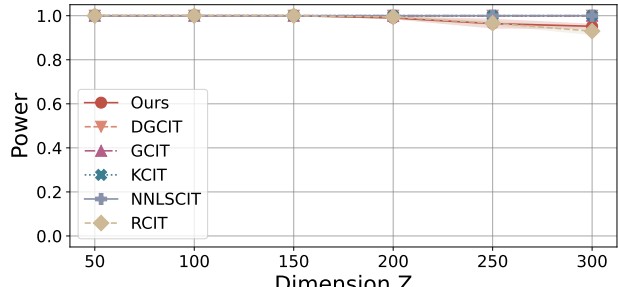

*Figure 2.* Type I error and power of our method (**SpectralCIT**) compared to state-of-the-art conditional independence tests across varying dimensionality of the conditioning set $Z$.

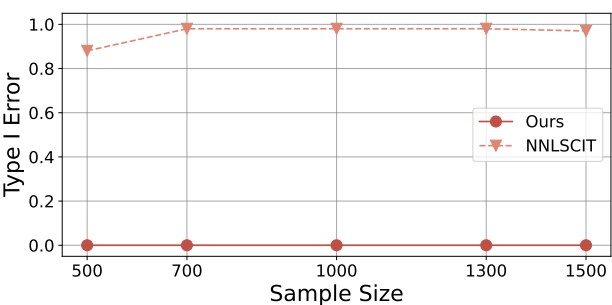

*Figure 3.* Type 1 error of our method (**SpectralCIT**) compared to **NNLSCIT** across varying sample sizes.

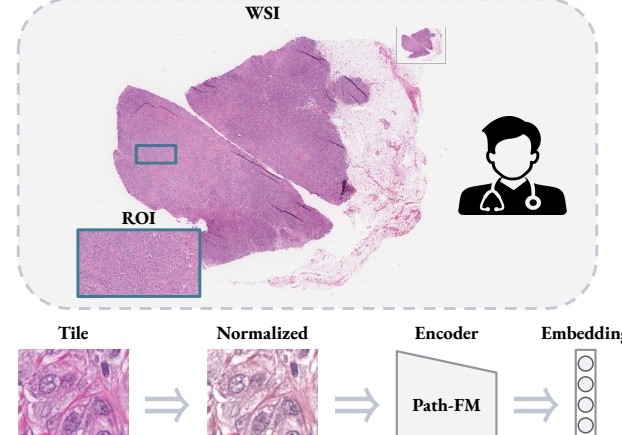

*Figure 4.* Data processing pipeline for breast cancer experiment.

Results at significance level $\alpha = 0.05$ and $b \in \{0, 2\}$ are shown in Fig. 5. Consistent with Ramdas et al. (2015), we observe a general loss of power for kernel-based CI tests. While **KCIT** achieved higher power than **SpectralCIT** for $d_Z \in \{250, 300\}$, it failed to maintain type I error control, in line with (Shi et al., 2021). In contrast, **GCM** preserved type I error validity but exhibited very low power, again consistent with (Shi et al., 2021). The method of Scetbon et al. (2022) was excluded due to excessive runtime.[6]

### 5.2. Breast Cancer Data

Predicting outcomes in cancer patients is crucial for clinical decision-making and precision medicine. Gene expression profiling can identify high-risk patients (Buyse et al., 2006; Sparano et al., 2018), but it remains costly and time-consuming, especially in underserved communities (Cabello et al., 2019). Histological images, routinely examined by pathologists to inform prognosis and treatment, inherently reflect underlying molecular phenotypes (Thennavan et al., 2021). Recent advances in computational pathology suggest that machine learning can extract prognostic signals beyond human perception (Farahmand et al., 2022; Wang et al., 2023), raising the possibility that sufficiently rich image representations may capture much of the prognostic infor-

---
[6]Cf. (Yang et al., 2025, Fig. 7).

mation encoded in molecular profiles. This leads to a key question: how much additional value do molecular profiles provide beyond image-based representations? Clinically, genomic assays such as MammaPrint (Buyse et al., 2006) and OncotypeDx (Sparano et al., 2018) provide prognostic information beyond standard pathological assessment, but it remains unclear whether modern foundation-model representations of histology already capture most of this signal. Motivated by this question, we combine pathology foundation models with our conditional independence testing framework to formally assess whether molecular profiles provide prognostic information beyond that captured by histological images.

**Dataset.** We selected all invasive carcinoma whole slide images (WSIs) from The Cancer Genome Atlas Breast Invasive Carcinoma (TCGA-BRCA) dataset (The Cancer Genome Atlas Network, 2012; Ciriello et al., 2015) with available survival data indicating either death within three years of diagnosis ($Y = 1$) or survival beyond five years of follow-up ($Y = 0$), resulting in 135 WSIs. For each WSI, a representative malignant tumor region (target area) was annotated by a medical doctor using QuPath (Bankhead et al., 2017). Target areas were then divided into $244 \times 244$ tiles, H&E stain-

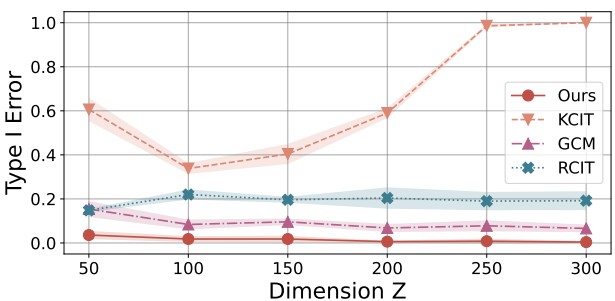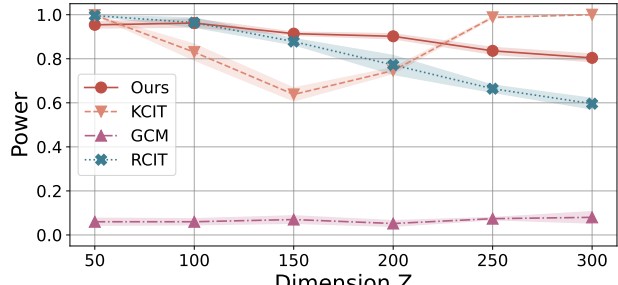

*Figure 5.* Type I error and power of our method (**SpectralCIT**) compared to state-of-the-art kernel-based conditional independence tests across varying dimensionality of the conditioning set $Z$.

normalized following (Macenko et al., 2009), and processed using the Path Foundation model (Lai et al., 2023) to extract latent representations encoding local nuclear morphology and micro-environmental context. Up to 10 target-area patches per WSI were selected based on Euclidean distance in the latent space, yielding a dataset of $N = 1341$ triplets $(X, Y, Z)$, where $X \in \mathbb{R}^3$ represents Her2, Luminal, and Basal metagene scores from (Tiezzi et al., 2025), $Y \in \{0, 1\}$ is the survival outcome, and $Z \in \mathbb{R}^{384}$ represents the high-dimensional image features. The data processing pipeline is illustrated in Fig. 4.

**Conditional independence testing.** We first examined linear associations via partial correlations between each coordinate $X$ and $Y$ given $Z$, with results indicating negligible linear dependence ($-0.006$, $-0.088$, $0.07$). In line with this observation, a logistic regression model incorporating both $X$ and $Z$ achieved slightly lower predictive accuracy than a model using $Z$ alone ($0.86$ vs. $0.87$). We then applied **KCIT**, **NNLSCIT**, and **SpectralCIT** to assess conditional independence beyond linear relationships. **KCIT** and **NNLSCIT** failed to reject the null hypothesis at significance level $\alpha = 0.05$, whereas **SpectralCIT** strongly rejected it (see Tab. 1). Training an XGBoost model (Chen & Guestrin, 2016) confirmed residual predictive information in $X$: accuracy rose from $0.91$ with $Z$ alone to $0.95$ with $X$ and $Z$. This indicates non-linear associations between metagene scores and survival not captured by latent representations, consistent with recent prognostic models that incorporate multi-modal data (Tolaney et al., 2024), and highlights SpectralCIT's ability to detect complex dependencies in high-dimensional biological data.

*Table 1.* P-values returned by the conditional independence tests.

| SpectralCIT | KCIT | NNLSCIT |
|---|---|---|
| $< 10^{-3}$ | $6.8 \times 10^{-2}$ | $3.8 \times 10^{-1}$ |

## 6. Related Work

We provide a brief overview of existing CI tests and spectral representation learning, and refer the reader to (Li & Fan, 2019) for a comprehensive discussion. We note that the CI testing literature is highly method-centric, with a lack of standardized benchmarks and well-established hyperparameter tuning protocols (Poinsot et al., 2025).

**Spectral representation learning.** Expansions in orthogonal bases such as Fourier, wavelets, and splines have a long history in nonparametric statistics (Efromovich, 1999), but perform poorly in high dimensions and lack adaptivity to the data distribution. This limitation motivated the use of bases derived from the spectral decomposition of linear integral operators defined by symmetric positive semi-definite kernels (Izbicki, 2014). More recently, research has shifted toward learned spectral bases, or features, which have been applied across diverse settings including dynamical systems (Turri et al., 2026), reinforcement learning (Hu et al., 2024), causal inference (Wang et al., 2022; Sun et al., 2025; Meunier et al., 2025b;a), and geometric deep learning (Ordoñez-Apraez et al., 2025).

**Kernel tests of CI.** Kernel-based measures of conditional dependence were first introduced in (Fukumizu et al., 2007), building on the partial cross-covariance operator from (Fukumizu et al., 2004). The first kernel-based CI test, **KCIT** (Zhang et al., 2011), uses the Hilbert–Schmidt norm of this operator and provides a characterization of the asymptotic null distribution, but without power guarantees. To improve scalability, Strobl et al. (2019) introduced **RCIT** and **RCoT**, which approximate **KCIT** using random Fourier features. More recently, kernel partial correlation (Huang et al., 2022), generalized covariance measure (**GCM**) (Shah & Peters, 2020), and tests based on analytical kernel mean embeddings (**LPCIT**) (Scetbon et al., 2022) have been developed. A causal representation learning method leveraging a kernel-based statistic has been proposed in (Pogodin et al., 2023). Practical strategies for kernel selection have been discussed (Wang et al., 2026), though at notable computational cost. Despite their simplicity and effectiveness, kernel-based tests

rely on kernel mean embeddings (Muandet et al., 2017), resulting in slow convergence rates (Pogodin et al., 2025) and loss of power as the dimensionality increases (Ramdas et al., 2015).

**Model-x, Local permutation, and other tests of CI.** Model-X tests rely on knowledge of the conditional distribution $P(X|Z)$ and were first introduced in (Candès et al., 2018), showing that valid p-values are obtained when $P(X|Z)$ is known exactly by sampling from it. Berrett et al. (2020) extended this framework by designing a permutation test that requires only approximate knowledge of the conditional distribution. Motivated by advances in generative modeling with neural networks, Bellot & van der Schaar (2019) proposed learning $P(X|Z)$ and performing a permutation test (**GCIT**), with theoretical guarantees based on the GAN loss. However, Zhang et al. (2025) showed that these bounds can be loose even in simple linear settings and proposed a doubly robust alternative, while Shi et al. (2021) offered another double generative approach (**DGCIT**). More recently, Yang et al. (2025) introduced a diffusion-based variant. Local permutation tests (Runge, 2018; Kim et al., 2022; Bellot & van der Schaar, 2019; Fukumizu et al., 2007; Sen et al., 2017; Li et al., 2023) exploit the factorization $P_{X|Y=y,Z=z} = P_{X|Z=z}$ by permuting $X$ within binned or clustered $Z$ values, producing approximately valid samples from $P_{XYZ}$ under the null under smoothness assumptions and sufficiently populated clusters (Kim et al., 2022).

## 7. Conclusions

In this paper, we revisited conditional independence testing with partial covariance operators through the lens of learned spectral features, addressing the scalability and adaptivity limitations of kernel-based tests. We proposed a scalable bi-level contrastive algorithm to learn leading spectral features of partial covariance operators, enabling simple linear test statistics while maintaining validity and power. Under the null, we showed that validity depends on central limit theorem rates for the test statistic's matrix, which can be easily controlled using bounded activations such as `Tanh`. Under alternative, we established that power relies on the quality of the learned representations–an equivalent but more machine learning-aligned perspective. Finally, we evaluated the proposed test on high-dimensional synthetic and real-world datasets, including breast cancer data from (The Cancer Genome Atlas Network, 2012), showing that our approach offers a practical and statistically grounded path toward scalable CI testing, bridging kernel-based theory with modern representation learning.

**Limitations.** Our work shares the following limitations with the CI testing and spectral representation learning literatures. First, there is no standard protocol for hyperparameter selection in conditional independence testing. As described in Appendix C, we therefore either use established values or apply a common tuning protocol with equal computational budgets across methods, varying only hyperparameters exposed in the original implementations. Second, unlike the unconditional setting where permutation methods provide exact calibration (Berrett & Samworth, 2019), existing tests require additional assumptions: model-X methods depend on knowledge of $P_{X|Z}$, local permutation methods rely on smoothness, and regression-based approaches require sufficiently fast regression estimation. Our method is no exception. Although our assumptions for validity are somewhat weaker, requiring only moment control of learned features, we still observe conservative calibration in practice, which can reduce power. Finally, our guarantees depend on the quality of the learned representations through the $\mathcal{E}_m$ terms in Section 4. We make this dependence explicit rather than tying the analysis to a particular architecture or optimization scheme. Obtaining explicit non-asymptotic rates for learned spectral representations remains an important open problem beyond our specific method (Meunier et al., 2025a; Kostic et al., 2024).

## Acknowledgments

This work was supported by the EU Project ELIAS (grant No. 101120237), and by the European Union – NextGenerationEU and the Italian National Recovery and Resilience Plan through the Ministry of University and Research (MUR), under Project PE0000013 CUP J53C22003010006. KL acknowledges support from the French National Research Agency for the DECATTLON project (ANR-24-CE40-3341).

## Impact Statement

This paper presents work whose goal is to advance the field of machine learning. There are many potential societal consequences of our work, none of which we feel must be specifically highlighted here.

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

## A. Derivation of Losses and Algorithm for Learning the Partial Covariance Operator

In this section, we present the full derivation of the losses and the bi-level contrastive algorithm presented in Section 3.

**From Eckart-Young-Mirsky to a loss.** Let $d \in \mathbb{N}$. Our goal is to learn $[\![\Sigma_{X\ddot{Y} \cdot Z}]\!]_d$ from an i.i.d. sample from $P_{X,Y,Z}$ using representation learning. To do so, we require a loss whose minimizer is $[\![\Sigma_{X\ddot{Y} \cdot Z}]\!]_d$. We base our loss on the following reformulation of Eckart-Young-Mirsky theorem from (Kostic et al., 2024).

**Theorem A.1** (Eckart-Young-Mirsky: loss form). *Let $\mathsf{A} : \mathcal{G} \to \mathcal{F}$ be a compact operator and $\gamma > 0$, then the solution of*

$$\min_{\mathsf{U,M,V}} \quad \mathcal{L}(\mathsf{U}, \mathsf{M}, \mathsf{V}) + \gamma\, \Omega(\mathsf{U}, \mathsf{V}),$$
$$where \quad \mathcal{L}(\mathsf{U}, \mathsf{M}, \mathsf{V}) = \|\mathsf{A} - \mathsf{UMV}^*\|_{\mathrm{HS}}^2 - \|\mathsf{A}\|_{\mathrm{HS}}^2$$
$$\Omega(\mathsf{U}, \mathsf{V}) = \|\mathsf{U}^*\mathsf{U} - \mathsf{Id}\|_{\mathrm{HS}}^2 + \|\mathsf{V}^*\mathsf{V} - \mathsf{Id}\|_{\mathrm{HS}}^2,$$

*and $\mathsf{U} : \mathbb{R}^d \to \mathcal{F}, \mathsf{M} : \mathbb{R}^d \to \mathbb{R}^d, \mathsf{V} : \mathbb{R}^d \to \mathcal{G}$ are linear operators, is given by the rank-$d$ truncated SVD of $\mathsf{A}$.*

Applied to our problem, Theorem A.1 yields the following optimization problem:

$$\min_{\mathsf{U,M,V}} \quad \mathcal{L}(\mathsf{U}, \mathsf{M}, \mathsf{V}) + \gamma\, \Omega(\mathsf{U}, \mathsf{V})$$
$$\text{where} \quad \mathcal{L}(\mathsf{U}, \mathsf{M}, \mathsf{V}) = \|\Sigma_{X\ddot{Y} \cdot Z} - \mathsf{UMV}^*\|_{\mathrm{HS}}^2 - \|\Sigma_{X\ddot{Y} \cdot Z}\|_{\mathrm{HS}}^2 \quad (11)$$
$$\Omega(\mathsf{U}, \mathsf{V}) = \|\mathsf{U}^*\mathsf{U} - \mathsf{Id}\|_{\mathrm{HS}}^2 + \|\mathsf{V}^*\mathsf{V} - \mathsf{Id}\|_{\mathrm{HS}}^2.$$

As before, we take $\mathsf{U} : \mathbb{R}^d \to L^2(X)$, $\mathsf{V} : \mathbb{R}^d \to L^2(\ddot{Y})$, and $\mathsf{M} : \mathbb{R}^d \to \mathbb{R}^d$ to be linear maps spanning (at most) $d$-dimensional subspaces of $L^2(X)$, $L^2(\ddot{Y})$, and $\mathbb{R}^d$. We further assume that the functions spanned by $\mathsf{U}$ and $\mathsf{V}$ are centered, i.e., $\mathrm{range}(\mathsf{U}), \mathrm{range}(\mathsf{V}) \subset \mathrm{span}(1)^\perp$. This reduces the search space for $\mathsf{U}$ and $\mathsf{V}$, effectively simplifying the problem, and is justified by $1 \in \mathrm{kernel}(\Sigma_{X\ddot{Y} \cdot Z}) \cap \mathrm{kernel}(\Sigma_{X\ddot{Y} \cdot Z}^*)$ and the left and right singular functions of $\Sigma_{X\ddot{Y} \cdot Z}$ being contained in $\mathrm{kernel}(\Sigma_{X\ddot{Y} \cdot Z}^*)^\perp$ and $\mathrm{kernel}(\Sigma_{X\ddot{Y} \cdot Z})^\perp$, respectively.

By subtracting the constant $\|\Sigma_{X\ddot{Y} \cdot Z}\|_{\mathrm{HS}}^2 < \infty$, we can expand the loss function as follows

$$\|\Sigma_{X\ddot{Y} \cdot Z} - \mathsf{UMV}^*\|_{\mathrm{HS}}^2 - \|\Sigma_{X\ddot{Y} \cdot Z}\|_{\mathrm{HS}}^2$$
$$= \|\mathsf{UMV}^*\|_{\mathrm{HS}}^2 - 2\langle\Sigma_{X\ddot{Y}}, \mathsf{UMV}^*\rangle_{\mathrm{HS}} + 2\langle\Sigma_{XZ}\Sigma_{Z\ddot{Y}}, \mathsf{UMV}^*\rangle_{\mathrm{HS}} \quad (12)$$
$$= \mathrm{tr}(C_{UU}MC_{VV}M^T) - 2\,\mathrm{tr}(C_{UV}M^T) + 2\,\mathrm{tr}(\Sigma_{XZ}\Sigma_{Z\ddot{Y}}\mathsf{VM}^*\mathsf{U}^*),$$

where $C_{UU}$ and $C_{VV}$ are the covariance matrices of $U$ and $V$, $C_{UV}$ is the cross-covariance matrix of $U = [u_1(X) \ldots u_d(X)]^\top$ and $V = [v_1(\ddot{Y}) \ldots v_d(\ddot{Y})]^\top$, and $M$ is the matrix of $\mathsf{M}$ w.r.t. the canonical basis of $\mathbb{R}^d$. To deal with the last term $\overline{\mathsf{F}} = \Sigma_{Z\ddot{Y}}\mathsf{VM}^*\mathsf{U}^*\Sigma_{XZ}$, we first symmetrize it $\mathsf{F} = (\overline{\mathsf{F}} + \overline{\mathsf{F}}^*)/2$, and then rely on the fact that $\mathrm{tr}(\mathsf{F}) = \mathrm{tr}(\mathsf{WW}^*\mathsf{F})$, for any linear map $\mathsf{W} : \mathbb{R}^{2d} \to L^2(Z)$ satisfying $\mathsf{W}^*\mathsf{W} = I_{2d}$ and $\mathrm{range}(\mathsf{F}) \subseteq \mathrm{range}(\mathsf{W})$, i.e., any orthogonal projector $\mathsf{WW}^*$ onto $\mathrm{range}(\mathsf{F})$. Again, since the constant function $\mathbf{1} \in L^2(Z)$ satisfies $\mathbf{1} \in \mathrm{kernel}(\mathsf{F})$, we search for centered functions by requiring $\mathrm{range}(\mathsf{W}) \subset \mathrm{span}(\mathbf{1})^\perp$. With this, the last term reduces to $2\,\mathrm{tr}(C_{UW}C_{WV}M^T)$, where $C_{UW}$ and $C_{WV}$ are the cross-covariance matrices of $U, W$ and $W, V$. The regularization terms can be expanded similarly

$$\|\mathsf{U}^*\mathsf{U} - \mathsf{Id}\|_{\mathrm{HS}}^2 = \|C_{UU} - I_d\|_{\mathrm{F}}^2, \qquad \|\mathsf{V}^*\mathsf{V} - \mathsf{Id}\|_{\mathrm{HS}}^2 = \|C_{VV} - I_d\|_{\mathrm{F}}^2. \quad (13)$$

**Bi-level formulation.** Since evaluating $\mathrm{tr}(\mathsf{F})$ requires learning an orthogonal projector $\mathsf{WW}^*$ such that $\mathrm{range}(\mathsf{W}) \supset \mathrm{range}(\mathsf{F})$, we adopt the following bi-level optimization problem moving forward

$$\min_{U,M,V} \quad \mathcal{L}_{\mathrm{out}}(U, M, V, W_{\mathrm{in}}) + \gamma\, \Omega_{\mathrm{out}}(U, V)$$
$$\text{s.t.} \quad (W_{\mathrm{in}}, N_{\mathrm{in}}) \in \underset{W, N}{\arg\min}\, \mathcal{L}_{\mathrm{in}}(W, N) + \gamma\, \Omega_{\mathrm{in}}(W) \quad (14)$$
$$\text{where} \quad \mathcal{L}_{\mathrm{out}}(U, M, V, W_{\mathrm{in}}) = \mathrm{tr}(C_{UU}MC_{VV}M^T) - 2\,\mathrm{tr}(C_{UV}M^T) + 2\,\mathrm{tr}(C_{UW}C_{WV}M^T)$$
$$\mathcal{L}_{\mathrm{in}}(W, N) = \mathrm{tr}(C_{WW}NC_{WW}N^T) - \mathrm{tr}(C_{UW}(N + N^T)C_{WV}M^T),$$

where $C_{WW}$ is the covariance matrix of $W = [w_1(Z) \ldots w_{2d}(Z)]^\top$ and $\mathsf{N} : \mathbb{R}^{2d} \to \mathbb{R}^{2d}$ is a linear map. The inner loss is the same regularized Hilbert-Schmidt loss as in (11), but applied to learning the finite-rank operator $\mathsf{F}$ with $\mathsf{WNW}^*$.

**Empirical resolution.** In practice, we can represent the finite-rank operators U, V, W, M, and N using neural networks $u_\theta : \mathcal{X} \to \mathbb{R}^d$, $v_\theta : \mathcal{Y} \times \mathcal{Z} \to \mathbb{R}^d$, and $w_\theta : \mathcal{Z} \to \mathbb{R}^{2d}$, along with learnable weight matrices $M_\theta \in \mathbb{R}^{d \times d}$, $N_\theta \in \mathbb{R}^{2d \times 2d}$. These parameterizations define subspaces of $L^2(X)$, $L^2(\ddot{Y})$, and $L^2(Z)$, and model the action of the operators as linear maps between them. Moreover, the bi-level optimization problem (14) corresponds to simultaneously minimizing two regularized contrastive losses, which is amenable to traditional gradient-based optimization and features manifold connections to contrastive learning (HaoChen et al., 2021)

$$
\begin{aligned}
\widehat{\mathcal{L}}_{\text{out}}(\theta) = {} & \frac{1}{m(m-1)} \sum_{i \neq j}^m \langle \overline{u}_\theta(X_i), M_\theta \overline{v}_\theta(\ddot{Y}_j) \rangle^2 \\
& - \frac{2}{m} \sum_{i=1}^m \langle \overline{u}_\theta(X_i), M_\theta \overline{v}_\theta(\ddot{Y}_i) \rangle \\
& + \frac{2}{m(m-1)} \sum_{i \neq j}^m \langle \overline{u}_\theta(X_i), M_\theta \overline{v}_\theta(\ddot{Y}_j) \rangle \langle \overline{w}_\theta(Z_i), \overline{w}_\theta(Z_j) \rangle, \\
\widehat{\mathcal{L}}_{\text{in}}(\theta) = {} & \frac{1}{m(m-1)} \sum_{i \neq j}^m \langle \overline{w}_\theta(Z_i), N_\theta \overline{w}_\theta(Z_j) \rangle^2 \\
& - \frac{2}{m(m-1)} \sum_{i \neq j}^m \langle \overline{u}_\theta(X_i), M_\theta \overline{v}_\theta(\ddot{Y}_j) \rangle \langle \overline{w}_\theta(Z_i), N_\theta \overline{w}_\theta(Z_j) \rangle,
\end{aligned}
$$

where $\overline{\cdot}$ denotes centering and $N_\theta$ is taken to be symmetric. Furthermore, orthonormality regularizers such as the following are used with strength $\gamma > 0$

$$
\begin{aligned}
\widehat{\Omega}_{\text{out}}(\theta) &= \|\widehat{C}_{U_\theta U_\theta} - I_d\|_{\text{F}}^2 + \|\widehat{C}_{V_\theta V_\theta} - I_d\|_{\text{F}}^2, \\
\widehat{\Omega}_{\text{in}}(\theta) &= \|\widehat{C}_{W_\theta W_\theta} - I_{2d}\|_{\text{F}}^2.
\end{aligned}
$$

# B. Proofs of Statistical Results

## B.1. Background on statistical hypothesis testing

**Statistical hypothesis testing.** We briefly recall the fundamental concepts of statistical hypothesis testing, as they will be used throughout the paper.

Given a sample $\mathcal{D}_n = \{(X_i, Y_i, Z_i)\}_{i=1}^n$ drawn i.i.d. from a joint distribution $P_{XYZ}$, a *statistical test* is a decision rule that determines whether to reject a null hypothesis $H_0$ in favor of an alternative hypothesis $H_1$. This decision is typically based on a *test statistic* $T_n = T(\mathcal{D}_n)$, a real-valued function designed to capture evidence against $H_0$.

The test proceeds by defining a *rejection region* $\mathcal{R}_n \subset \mathbb{R}$ such that:

$$
\text{Reject } H_0 \quad \Longleftrightarrow \quad T_n \in \mathcal{R}_n. \tag{15}
$$

Commonly, $\mathcal{R}_n$ takes the form of a threshold rule $\mathcal{R}_n = \{t \in \mathbb{R} : t \geq t_\alpha\}$, where $t_\alpha$ is the *critical value* chosen to control the *Type I error rate* at a desired significance level $\alpha \in (0,1)$. Specifically, the Type I error corresponds to the probability of incorrectly rejecting $H_0$ when it is true:

$$
\alpha = \mathbb{P}_{H_0}(T_n \in \mathcal{R}_n). \tag{16}
$$

Conversely, the *Type II error* is the probability of failing to reject $H_0$ when the alternative hypothesis holds:

$$
\beta(P_{XYZ}) = \mathbb{P}_{H_1}(T_n \notin \mathcal{R}_n), \tag{17}
$$

where we emphasize that $\beta$ depends on the specific alternative distribution $P_{XYZ} \in H_1$. The *power* of the test is defined as the probability of correctly rejecting $H_0$ under $H_1$:

$$
\text{Power} = 1 - \beta(P_{XYZ}). \tag{18}
$$

The objective in hypothesis testing is to design test statistics and rejection regions that simultaneously control the Type I error at level $\alpha$ and achieve high power against relevant alternatives. In nonparametric settings, where minimal assumptions are made on $P_{XYZ}$, this is particularly challenging.

A fundamental notion in the asymptotic analysis of hypothesis tests is that of *contiguity* (van der Vaart, 1998). A sequence of distributions $\{Q_n\}$ is said to be contiguous with respect to the sequence of distributions $\{P_n\}$ if, for every sequence of events $\{A_n\}$, we have:

$$P_n(A_n) \to 0 \implies Q_n(A_n) \to 0, \tag{19}$$

as $n \to \infty$. In the context of statistical testing, contiguity implies that the alternative becomes asymptotically indistinguishable from the null: no test statistic can achieve both vanishing Type I and Type II error rates.

### B.2. Background on sub-Gaussian random variables

We recall the definition of a sub-Gaussian random variable and some of its useful properties.

Let $\psi_2(x) = e^{x^2} - 1$, $x \geq 0$. We define the $\psi_2$-Orlicz norm of a random variable $\eta$ as

$$\|\eta\|_{\psi_2} := \inf \left\{ C > 0 \,:\, \mathbb{E}\left[\psi_2\left(\frac{|\eta|}{C}\right)\right] \leq 1 \right\}.$$

**Definition B.1** (Sub-Gaussian random vector). *A centered random vector $X \in \mathbb{R}^d$, with probability distribution denoted $\mu_X$, will be called sub-Gaussian iff, for all $u \in \mathbb{R}^d$,*

$$\|\langle X, u\rangle\|_{\psi_2} \lesssim \|\langle X, u\rangle\|_{L_2(\mu_X)}.$$

**Lemma B.1** ((Sub-Gaussian random variable) Lemma 5.5. in (Vershynin, 2011)). *Let $Z$ be a random variable. Then, the following assertions are equivalent with parameters $K_i > 0$ differing from each other by at most an absolute constant factor.*

1. *Tails: $\mathbb{P}\{|Z| > t\} \leq \exp(1 - t^2/K_1^2)$ for all $t \geq 0$;*

2. *Moments: $(\mathbb{E}|Z|^p)^{1/p} \leq K_2\sqrt{p}$ for all $p \geq 1$;*

3. *Super-exponential moment: $\mathbb{E}\exp(Z^2/K_3^2) \leq 2$.*

*A random variable $Z$ satisfying any of the above assertions is called a sub-Gaussian random variable. We will denote by $K_3$ the sub-Gaussian norm.*

Consequently, a sub-Gaussian random variable satisfies the following equivalence of moments property. There exists an absolute constant $c > 0$ such that for any $m \geq 2$,

$$\left(\mathbb{E}|Z|^m\right)^{1/m} \leq cK_3\sqrt{m}\left(\mathbb{E}|Z|^2\right)^{1/2}. \tag{20}$$

Let $\eta_1, \ldots, \eta_n, \eta$ be i.i.d $K$-sub-Gaussian random vectors in $\mathbf{R}^d$. Then there exists an absolute constant $C > 0$ such that for all $\delta \in (0, 1)$,

$$\mathbb{P}\left(\|\frac{1}{n}\sum_{I=1}^n \eta_i - \mathbb{E}[\eta]\| \leq CK\left(\sqrt{\frac{d}{n}} + \sqrt{\frac{\log(1/\delta)}{n}}\right)\right) \geq 1 - \delta.$$

where $\|\cdot\|$ is the Euclidean norm.

### B.3. Background on Lindeberg-Feller Multivariate Central Limit Theorem

We use the multivariate CLT for independent (not necessarily identically distributed) vector-valued random variables:

**Theorem B.1** (Multivariate Central Limit Theorem). *Let $\{W_i\}_{i=1}^n \subset \mathbb{R}^m$ be independent, mean-zero random vectors. Assume that*

$$\frac{1}{n}\sum_{i=1}^n \mathbb{E}[W_i W_i^\top] \to \Sigma \quad \text{and} \quad \text{for all } \varepsilon > 0, \quad \frac{1}{n}\sum_{i=1}^n \mathbb{E}\left[\|W_i\|^2 \cdot \mathbb{1}_{\{\|W_i\| > \varepsilon\sqrt{n}\}}\right] \to 0.$$

*Then*

$$\frac{1}{\sqrt{n}}\sum_{i=1}^n W_i \xrightarrow{d} \mathcal{N}(0, \Sigma).$$

## B.4. Technical results

We prove now several technical results that will be used in several parts of the proof of our main results. For the sake of brevity and to avoid cumbersome notation, we denote the learned features $\widehat{u}_\theta$, $\widehat{v}_\theta$, and $\widehat{w}_\theta$ obtained after the representation learning step by $u_\theta$, $v_\theta$, and $w_\theta$, respectively, and keep them fixed thereafter. In the testing phase, we compute the test statistic with new data $(\{x_i, y_i, z_i\}_{i=1}^n)$ not used during the representation learning phase.

**Spectral structure of $\widehat{P}_\theta$**    We define of the $n \times d$ matrices $\widehat{U}_\theta$, $\widehat{V}_\theta$ and the $n \times (2d)$ matrix $\widehat{W}_\theta$ as follows:

$$\widehat{U}_\theta = \frac{1}{\sqrt{n}} [u_\theta(x_1)|\cdots|u_\theta(x_n)]^\top, \quad \widehat{V}_\theta = \frac{1}{\sqrt{n}} [v_\theta(\ddot{y}_1)|\cdots|v_\theta(\ddot{y}_n)]^\top, \quad \widehat{W}_\theta = \frac{1}{\sqrt{n}} [w_\theta(z_1)|\cdots|w_\theta(z_n)]^\top .$$

Set also

$$\widehat{P}_\theta = I_n - \widehat{W}_\theta \widehat{W}_\theta^\top .$$

We assume from now on that $n > 2d$. Note first that the nonzero eigenvalues of $\widehat{W}_\theta \widehat{W}_\theta^\top$ are the same as those of $\widehat{W}_\theta^\top \widehat{W}_\theta$. We set $\mathrm{Cov}(w_\theta) = \mathbb{E}[\widehat{W}_\theta^\top \widehat{W}_\theta]$. Next we have

$$\|\widehat{W}_\theta^\top \widehat{W}_\theta - I_d\| \leq \|\widehat{W}_\theta^\top \widehat{W}_\theta - \mathbb{E}[\widehat{W}_\theta^\top \widehat{W}_\theta]\| + \|\mathbb{E}[\widehat{W}_\theta^\top \widehat{W}_\theta] - I_d\|$$
$$= \underbrace{\|\widehat{W}_\theta^\top \widehat{W}_\theta - \mathbb{E}[\widehat{W}_\theta^\top \widehat{W}_\theta]\|}_{(I)} + \underbrace{\|\mathrm{Cov}(w_\theta) - I_d\|}_{(II)} .$$

Recall the optimization gap condition:

$$(II) \leq \mathcal{E}_m.$$

We study now $(I)$. Note that

$$\widehat{W}_\theta^\top \widehat{W}_\theta - \mathbb{E}[\widehat{W}_\theta^\top \widehat{W}_\theta] = \frac{1}{n} \sum_{i \in [n]} w_\theta(z_i) \otimes w_\theta(z_i) - \mathrm{Cov}(w_\theta(z)).$$

The effective rank of a symmetric positive semi-definite matrix $A$ is

$$\mathbf{r}(A) = \frac{\mathrm{tr}(A)}{\|A\|}.$$

Define the rate

$$\psi_n(\delta) := c_K \|\mathrm{Cov}(w_\theta(z))\| \left( \sqrt{\frac{\mathbf{r}(\mathrm{Cov}(w_\theta(z)))}{n}} \bigvee \frac{\mathbf{r}(\mathrm{Cov}(w_\theta(z)))}{n} \bigvee \sqrt{\frac{\log \delta^{-1}}{n}} \bigvee \frac{\log \delta^{-1}}{n} \right),$$

where $c_K > 0$ is a numerical constant that can depend only $K$.

Using well-known concentration results for covariance operators like Koltchinskii & Lounici (2017, Corollary 2), we get for any $\delta \in (0,1)$, w.p.a.l. $1 - \delta$

$$\|\widehat{W}_\theta^\top \widehat{W}_\theta - \mathrm{Cov}(w_\theta(z))\| \leq \psi_n(\delta). \tag{21}$$

Hence we get for any $\delta \in (0,1)$, w.p.a.l. $1 - \delta$

$$\|\widehat{W}_\theta^\top \widehat{W}_\theta - I_d\| \leq \mathcal{E}_m + \psi_n(\delta).$$

Hence we get on the same probability event:

$$\max_{j \in [2d]} |\widehat{\lambda}_j - 1| \leq \mathcal{E}_m + \psi_n(\delta). \tag{22}$$

Recall that the nonzero eigenvalues of $\widehat{W}_\theta \widehat{W}_\theta^\top$ are the same as the nonzero eigenvalues of $\widehat{W}_\theta^\top \widehat{W}_\theta$ (assuming the worst case that $\widehat{W}_\theta^\top$ is full rank). Hence the eigenvalues of $\widehat{W}_\theta \widehat{W}_\theta^\top$ are 0 with multiplicity $n - 2d$ and the above $\widehat{\lambda}_j$'s. Consequently $\widehat{P}_\theta$ admits spectral decomposition with eigenvalue 1 of multiplicity $n - 2d$ and the remaining $d$ eigenvalues $\mu_j$ satisfying:

$$|\mu_j(\widehat{P}_\theta)| \leq \psi_n(\delta) + \mathcal{E}_m, \quad \forall j \in [n - 2d + 1, n], \tag{23}$$

and $\widehat{P}_\theta$ admits an orthonormal family $\{a_j\}_{j \in [n]}$ in $\mathbb{R}^n$ of eigenvectors such that

$$\widehat{P}_\theta = \sum_{j=1}^{n-2d} a_j \otimes a_j + \sum_{j=n-2d+1}^{n} \mu_j \, a_j \otimes a_j. \tag{24}$$

We discuss now the rate $\psi_n(\delta)$. We always have $\mathbf{r}(\mathrm{Cov}(w_\theta(z))) \leq 2d$. Furthermore, under the optimization gap condition, we have $\|\mathrm{Cov}(w_\theta(z))\| \leq 1 + \mathcal{E}_m$. Hence we have have the following upper bound for any $\delta \in (0, 1)$ such that $n \geq d \vee \log \delta^{-1}$:

$$\psi_n(\delta) \leq c_K (1 + \mathcal{E}_m) \left( \sqrt{\frac{2d}{n}} \bigvee \sqrt{\frac{\log \delta^{-1}}{n}} \right). \tag{25}$$

**Asymptotic covariance structure under the null hypothesis.** Using the tower property of conditional expectation, the conditional independence of $x_i$ and $\ddot{y}_i$ given $z_i$ under the null hypothesis, and the fact that $\{x_i, \ddot{y}_i, z_i\}_{i=1}^n$ is an i.i.d. sample set, we have for any $i \in [n]$:

$$\mathbb{E}\left[ \mathrm{vec}(u_\theta(x_i) \otimes v_\theta(\ddot{y}_i)) \, \mathrm{vec}(u_\theta(x_i) \otimes v_\theta(\ddot{y}_i))^\top \right]$$
$$= \mathbb{E}\left[ (u_\theta(x_i) \otimes v_\theta(\ddot{y}_i))(u_\theta(x_i) \otimes v_\theta(\ddot{y}_i))^\top \right]$$
$$= \mathbb{E}\left[ u_\theta(x_i) u_\theta(x_i)^\top \right] \otimes \mathbb{E}\left[ v_\theta(\ddot{y}_i) v_\theta(\ddot{y}_i)^\top \right]$$
$$= \mathrm{Cov}(u_\theta(x)) \otimes \mathrm{Cov}(v_\theta(\ddot{y})) \in \mathbb{R}^{d^2 \times d^2}, \tag{26}$$

where we used the identity $(a \otimes b)(a \otimes b)^\top = (aa^\top) \otimes (bb^\top)$ in the third line and the independence of $u_\theta(x_i)$ and $v_\theta(\ddot{y}_i)$ (conditional on $z_i$).

We now study the spectral structure of $\mathrm{Cov}(u_\theta(x)) \otimes \mathrm{Cov}(v_\theta(\ddot{y}))$. Denote by $\{\lambda_j(u_\theta)\}_{j \in [d]}$ and $\{\lambda_j(v_\theta)\}_{j \in [d]}$ the eigenvalues of $\mathrm{Cov}(u_\theta(x))$ and $\mathrm{Cov}(v_\theta(\ddot{y}))$ respectively. By properties of Kronecker products, $\mathrm{Cov}(u_\theta(x)) \otimes \mathrm{Cov}(v_\theta(\ddot{y}))$ admits eigenvalues $\{\lambda_j(u_\theta)\lambda_k(v_\theta)\}_{j,k \in [d]}$.

Now we prove that $\mathrm{Cov}(u_\theta(x)) \otimes \mathrm{Cov}(v_\theta(\ddot{y}))$ is approximately equal to $I_{d^2}$. To this end, we introduce the true representation basis $u, v$:

$$u(\cdot) = [u_1(\cdot), \ldots, u_d(\cdot)]^\top, \quad v(\cdot) = [v_1(\cdot), \ldots, v_d(\cdot)]^\top,$$

and we recall that $u$ and $v$ are orthonormal families w.r.t. $L^2_{\mu_x}$ and $L^2_{\mu_{\ddot{y}}}$, meaning that

$$\mathrm{Cov}(u(x)) \otimes \mathrm{Cov}(v(\ddot{y})) = I_d \otimes I_d = I_{d^2}.$$

Next we have

$$\mathrm{Cov}(u_\theta(x)) \otimes \mathrm{Cov}(v_\theta(\ddot{y})) - \mathrm{Cov}(u(x)) \otimes \mathrm{Cov}(v(\ddot{y}))$$
$$= \big( \mathrm{Cov}(u_\theta(x)) - \mathrm{Cov}(u(x)) \big) \otimes \mathrm{Cov}(v_\theta(\ddot{y}))$$
$$+ \mathrm{Cov}(u(x)) \otimes \big( \mathrm{Cov}(v_\theta(\ddot{y})) - \mathrm{Cov}(v(\ddot{y})) \big).$$

Taking the operator norm and using that $\|A \otimes B\|_{op} = \|A\|_{op} \|B\|_{op}$, we get

$$\|\mathrm{Cov}(u_\theta(x)) \otimes \mathrm{Cov}(v_\theta(\ddot{y})) - \mathrm{Cov}(u(x)) \otimes \mathrm{Cov}(v(\ddot{y}))\|_{op}$$
$$\leq \|\mathrm{Cov}(u_\theta(x)) - \mathrm{Cov}(u(x))\|_{op} \|\mathrm{Cov}(v_\theta(\ddot{y}))\|_{op}$$
$$+ \|\mathrm{Cov}(u(x))\|_{op} \|\mathrm{Cov}(v_\theta(\ddot{y})) - \mathrm{Cov}(v(\ddot{y}))\|_{op}$$
$$= \|\mathrm{Cov}(u_\theta(x)) - I_d\|_{op} \|\mathrm{Cov}(v_\theta(\ddot{y}))\|_{op} + \|I_d\|_{op} \|\mathrm{Cov}(v_\theta(\ddot{y})) - I_d\|_{op}.$$

We recall that by definition of the optimization gap

$$\mathcal{E}_m \geq \max \left\{ \|\mathrm{Cov}(u_\theta(x)) - I_d\|_{op}, \|\mathrm{Cov}(v_\theta(\ddot{y})) - I_d\|_{op} \right\}.$$

Combining the last two displays gives

$$\|\mathrm{Cov}(u_\theta(x)) \otimes \mathrm{Cov}(v_\theta(\ddot{y})) - I_{d^2}\|_{op} \leq (2 + \mathcal{E}_m)\mathcal{E}_m.$$

By standard perturbation bounds for eigenvalues, we deduce:

$$\max_{j,k\in[d]} \left\{ |\lambda_j(u_\theta)\lambda_k(v_\theta) - 1| \right\} \leq (2 + \mathcal{E}_m)\mathcal{E}_m. \tag{27}$$

Similarly, using that $\mathrm{tr}(A \otimes B) = \mathrm{tr}(A)\mathrm{tr}(B)$, we have:

$$\mathrm{tr}\left(\mathrm{Cov}(u_\theta(x)) \otimes \mathrm{Cov}(v_\theta(\ddot{y}))\right) = \mathrm{tr}\left(\mathrm{Cov}(u_\theta(x))\right)\mathrm{tr}\left(\mathrm{Cov}(v_\theta(\ddot{y}))\right) \leq (1 + \mathcal{E}_m)^2 d^2. \tag{28}$$

**Non-asymptotic bound in probability.**

**Lemma B.2.** *Let $\Gamma = \sum_{j=1}^r \lambda_j a_j \otimes a_j \in \mathbb{R}^{n\times n}$ where $\{a_j\}_{j\in r}$ is an orthonormal family of $\mathbb{R}^n$. We recall that the $\{x_i, \ddot{y}_i, z_i\}_{i\in[n]}$ are i.i.d. copies of $(X, \ddot{Y}, Z)$ and that $x \perp\!\!\!\perp \ddot{y} \,|\, z$ under the null hypothesis . Then, for any $\epsilon > 0$, we have*

$$\mathbb{P}\left(n\,\|\widehat{U}_\theta^\top \Gamma \widehat{V}_\theta\|_F^2 \geq \epsilon\right) \lesssim \frac{1}{\epsilon}\left[K^4 r\|\Gamma\|^2 \left(\frac{d^2}{n} + \frac{\log^2(n)}{n}\right)\right] + \frac{1}{n}. \tag{29}$$

*Proof.* We start with some preliminary observations exploiting sub-Gaussianity of the mappings $u_\theta(x)$ and $v_\theta(\ddot{y})$.

**Properties of mean and empirical means of $u_\theta(x)$ and $v_\theta(\ddot{y})$.** In view of Assumption 4.1 we have, by sub-Gaussian concentration, there exists an absolute constant $C > 0$ such that, w.p.a.l. $1 - \delta$,

$$\left\|\mathbb{E}[u_\theta(x)] - \frac{1}{n}\sum_{i=1}^n u_\theta(x_i)\right\| \bigvee \left\|\mathbb{E}[v_\theta(\ddot{y})] - \frac{1}{n}\sum_{i=1}^n v_\theta(\ddot{y}_i)\right\| \leq CK\left(\sqrt{\frac{d}{n}} + \sqrt{\frac{\log(\delta^{-1})}{n}}\right). \tag{30}$$

Recall that the functions $u_\theta$ and $v_\theta$ are empirically centered. Hence we get for any $\delta \in (0,1)$, w.p.a.l. $1 - \delta$

$$\left\|\mathbb{E}[u_\theta(x)]\right\| \bigvee \left\|\mathbb{E}[v_\theta(\ddot{y})]\right\| \leq CK\left(\sqrt{\frac{d}{n}} + \sqrt{\frac{\log(\delta^{-1})}{n}}\right). \tag{31}$$

Let $A := u_\theta(x) - \mathbb{E}[u_\theta(x)] \in \mathbb{R}^d$. Note that $A$ is $2K$-sub-Gaussian, i.e.,

$$\sup_{v\in\mathbb{S}^{d-1}} \left\|\langle A, v\rangle\right\|_{\psi_2} \leq 2K.$$

By the equivalence of moment property of sub-Gaussian distribution, there exists an absolute constant $C > 0$ such that, for any $m \geq 3$, we have

$$\mathbb{E}\|A\|^m \leq C^m (K\sqrt{d})^m m^{m/2} \left(\mathbb{E}\|A\|^2\right)^{m/2}.$$

Now consider the uncentered moment $\mathbb{E}\|u_\theta(x)\|^m = \mathbb{E}\|\mathbb{E}[u_\theta(x)] + A\|^m$. By the triangle inequality and the convexity of $t \mapsto t^m$ on $[0,\infty)$,

$$\|\mathbb{E}[u_\theta(x)] + A\|^m \leq \left(\|\mathbb{E}[u_\theta(x)]\| + \|A\|\right)^m \leq 2^{m-1}\left(\|\mathbb{E}[u_\theta(x)]\|^m + \|A\|^m\right).$$

Taking expectations yields

$$\mathbb{E}\|u_\theta(x)\|^m \leq 2^{m-1}\left(\|\mathbb{E}[u_\theta(x)]\|^m + C^m (K\sqrt{d})^m m^{m/2} \left(\mathbb{E}\|A\|^2\right)^{m/2}\right). \tag{32}$$

**Representation of the rescaled Frobenius norm.** Note first that $[\Gamma]_{i,k} = \sum_{j=1}^{r} \lambda_j a_{i,j} a_{k,j}$. We have

$$\sqrt{n}\widehat{U}_\theta^\top \Gamma \widehat{V}_\theta = \frac{1}{\sqrt{n}} \sum_{j=1}^{r} \sum_{i_1,i_2=1}^{n} \lambda_j a_{i_1,j} a_{i_2,j} u_\theta(x_{i_1}) \otimes v_\theta(\ddot{y}_{i_2}) = \frac{1}{\sqrt{n}} \sum_{i_1,i_2=1}^{n} [\Gamma]_{i_1,i_2} u_\theta(x_{i_1}) \otimes v_\theta(\ddot{y}_{i_2}).$$

The Frobenius norm of $\sqrt{n}\widehat{U}_\theta^\top \Gamma \widehat{V}_\theta$ is

$$n\|\widehat{U}_\theta^\top \Gamma \widehat{V}_\theta\|_F^2 = \frac{1}{n} \sum_{i_1,i_2,i_3,i_4=1}^{n} [\Gamma]_{i_1,i_2}[\Gamma]_{i_3,i_4} \langle u_\theta(x_{i_1}), u_\theta(x_{i_3})\rangle \langle v_\theta(\ddot{y}_{i_2}), v_\theta(\ddot{y}_{i_4})\rangle \tag{33}$$

**Analysis of the rescaled Frobenius norm under the null hypotheis.**

We exploit now the conditional independence of $x$ and $\ddot{y}$ given $z$. We take the expectation in (33). Exploiting the tower property of conditional expectation, we get

$$\mathbb{E}\left[\langle u_\theta(x_{i_1}), u_\theta(x_{i_3})\rangle \langle v_\theta(\ddot{y}_{i_2}), v_\theta(\ddot{y}_{i_4})\rangle\right]$$
$$= \mathbb{E}\left[\mathbb{E}\left[\langle u_\theta(x_{i_1}), u_\theta(x_{i_3})\rangle \langle v_\theta(\ddot{y}_{i_2}), v_\theta(\ddot{y}_{i_4})\rangle \mid z_{i_1}, z_{i_2}, z_{i_3}, z_{i_4}\right]\right]$$
$$= \mathbb{E}\left[\langle u_\theta(x_{i_1}), u_\theta(x_{i_3})\rangle\right] \mathbb{E}\left[\langle v_\theta(\ddot{y}_{i_2}), v_\theta(\ddot{y}_{i_4})\rangle\right]$$
$$= \begin{cases} \langle \mathbb{E}\left[u_\theta(x_{i_1})\right], \mathbb{E}\left[u_\theta(x_{i_3})\right]\rangle \langle \mathbb{E}\left[v_\theta(\ddot{y}_{i_2})\right], \mathbb{E}\left[v_\theta(\ddot{y}_{i_4})\right]\rangle & \text{if } i_1 \neq i_3 \text{ and } i_2 \neq i_4 \\ \mathbb{E}\left[\|u_\theta(x_{i_1})\|^2\right] \mathbb{E}\left[\|v_\theta(\ddot{y}_{i_2})\|^2\right] & \text{if } i_1 = i_3 \text{ and } i_2 = i_4 \end{cases}.$$

Hence

$$\left|\mathbb{E}\left[\langle u_\theta(x_{i_1}), u_\theta(x_{i_3})\rangle \langle v_\theta(\ddot{y}_{i_2}), v_\theta(\ddot{y}_{i_4})\rangle\right]\right|$$
$$\leq \begin{cases} \|\mathbb{E}\left[u_\theta(x_{i_1})\right]\|^2 \|\mathbb{E}\left[v_\theta(\ddot{y}_{i_2})\right]\|^2 & \text{if } i_1 \neq i_3 \text{ and } i_2 \neq i_4 \\ \mathbb{E}\left[\|u_\theta(x_{i_1})\|^2\right] \mathbb{E}\left[\|v_\theta(\ddot{y}_{i_2})\|^2\right] & \text{if } i_1 = i_3 \text{ and } i_2 = i_4 \end{cases}$$

Recall that Assumption 4.1 implies $\mathbb{E}[\|u_\theta(x)\|^2] \leq CK^2 d$ and $\mathbb{E}[\|v_\theta(y)\|^2] \leq CK^2 d$. We also have in view of (31), for any $\delta \in (0,1)$, w.p.a.l. $1-\delta$, for any indices satisfying $i_1 \neq i_3$ and $i_2 \neq i_4$,

$$\left|\mathbb{E}\left[\langle u_\theta(x_{i_1}), u_\theta(x_{i_3})\rangle \langle v_\theta(\ddot{y}_{i_2}), v_\theta(\ddot{y}_{i_4})\rangle\right]\right| \leq C^4 K^4 \left(\sqrt{\frac{d}{n}} + \sqrt{\frac{\log(\delta^{-1})}{n}}\right)^4, \tag{34}$$

Taking expectations in (33) and decomposing the sum into diagonal and off-diagonal terms, we obtain

$$\mathbb{E}\left[n\|\widehat{U}_\theta^\top \Gamma \widehat{V}_\theta\|_F^2\right] = \frac{1}{n} \sum_{i_1,i_2=1}^{n} [\Gamma]_{i_1,i_2}[\Gamma]_{i_1,i_2} \mathbb{E}\left[\|u_\theta(x_{i_1})\|^2\right] \mathbb{E}\left[\|v_\theta(\ddot{y}_{i_2})\|^2\right]$$
$$+ \frac{1}{n} \sum_{\substack{i_1,i_2,i_3,i_4=1 \\ (i_1,i_2)\neq(i_3,i_4)}}^{n} [\Gamma]_{i_1,i_2}[\Gamma]_{i_3,i_4} \mathbb{E}\left[\langle u_\theta(x_{i_1}), u_\theta(x_{i_3})\rangle \langle v_\theta(\ddot{y}_{i_2}), v_\theta(\ddot{y}_{i_4})\rangle\right].$$

For the diagonal terms, we have

$$\left|\frac{1}{n} \sum_{i_1,i_2=1}^{n} [\Gamma]_{i_1,i_2}^2 \mathbb{E}\left[\|u_\theta(x)\|^2\right] \mathbb{E}\left[\|v_\theta(y)\|^2\right]\right| \leq \frac{CK^4 d^2}{n} \sum_{i_1,i_2=1}^{n} [\Gamma]_{i_1,i_2}^2 = CK^4 d^2 \frac{\|\Gamma\|_F^2}{n} \leq CK^4 d^2 \frac{r\|\Gamma\|^2}{n}.$$

For the off-diagonal terms, using (34) with $\delta = 1/n$, w.p.a.l. $1 - 1/n$,

$$\left| \frac{1}{n} \sum_{\substack{i_1,i_2,i_3,i_4=1 \\ (i_1,i_2) \neq (i_3,i_4)}}^{n} [\Gamma]_{i_1,i_2} [\Gamma]_{i_3,i_4} \mathbb{E}\left[ \langle u_\theta(x_{i_1}), u_\theta(x_{i_3}) \rangle \langle v_\theta(\ddot{y}_{i_2}), v_\theta(\ddot{y}_{i_4}) \rangle \right] \right|$$

$$\leq \frac{C^4 K^4}{n} \left( \sqrt{\frac{d}{n}} + \sqrt{\frac{\log(n)}{n}} \right)^4 \sum_{i_1,i_2,i_3,i_4=1}^{n} |[\Gamma]_{i_1,i_2}||[\Gamma]_{i_3,i_4}|$$

$$\leq \frac{C^4 K^4}{n} \left( \sqrt{\frac{d}{n}} + \sqrt{\frac{\log(n)}{n}} \right)^4 \left( \sum_{i_1,i_2=1}^{n} |[\Gamma]_{i_1,i_2}| \right)^2$$

$$\leq \frac{C^4 K^4}{n} \left( \sqrt{\frac{d}{n}} + \sqrt{\frac{\log(n)}{n}} \right)^4 n^2 \|\Gamma\|_F^2$$

$$\leq C^4 K^4 nr \|\Gamma\|^2 \left( \frac{d^2}{n^2} + \frac{\log^2(n)}{n^2} \right).$$

Combining both terms, we obtain w.p.a.l. $1 - 1/n$,

$$\left| \mathbb{E}\left[ n \|\widehat{U}_\theta^\top \Gamma \widehat{V}_\theta\|_F^2 \right] \right| \lesssim K^4 r \|\Gamma\|^2 \left( \frac{d^2}{n} + \frac{\log^2(n)}{n} \right). \tag{35}$$

**Final step.** Let $B_n$ denote the event

$$B_n := \left\{ \left\| \mathbb{E}[u_\theta(x)] \right\| \bigvee \left\| \mathbb{E}[v_\theta(\ddot{y})] \right\| \leq CK \left( \sqrt{\frac{d}{n}} + \sqrt{\frac{\log(n)}{n}} \right) \right\}.$$

By (31) with $\delta = 1/n$, we have $\mathbb{P}(B_n^c) \leq 1/n$.

On the event $B_n$, by (35), we have

$$\mathbb{E}[S\mathbf{1}_{B_n}] \lesssim K^4 r \|\Gamma\|^2 \left( \frac{d^2}{n} + \frac{\log^2(n)}{n} \right). \tag{36}$$

For any $\epsilon > 0$, we have

$$\mathbb{P}\left( n \|\widehat{U}_\theta^\top \Gamma \widehat{V}_\theta\|_F^2 \geq \epsilon \right) = \mathbb{P}(S \geq \epsilon)$$

$$= \mathbb{P}(S \geq \epsilon, B_n) + \mathbb{P}(S \geq \epsilon, B_n^c)$$

$$\leq \mathbb{P}(S \geq \epsilon, B_n) + \mathbb{P}(B_n^c)$$

$$\leq \mathbb{P}(S \geq \epsilon, B_n) + \frac{1}{n}.$$

By Markov's inequality,

$$\mathbb{P}(S \geq \epsilon, B_n) = \mathbb{P}(S\mathbf{1}_{B_n} \geq \epsilon) \leq \frac{\mathbb{E}[S\mathbf{1}_{B_n}]}{\epsilon}.$$

Combining with (36), we obtain

$$\mathbb{P}\left( n \|\widehat{U}_\theta^\top \Gamma \widehat{V}_\theta\|_F^2 \geq \epsilon \right) \lesssim \frac{K^4 r \|\Gamma\|^2}{\epsilon} \left( \frac{d^2}{n} + \frac{\log^2(n)}{n} \right) + \frac{1}{n}. \tag{37}$$

$\square$

**B.5. Approximation of $\widehat{T}_n$.**

We recall that

$$\widehat{T}_n = n\|\widehat{U}_\theta^\top \widehat{V}_\theta - \widehat{U}_\theta^\top \widehat{W}_\theta \widehat{W}_\theta^\top \widehat{V}_\theta\|_F^2 = n\|\widehat{U}_\theta^\top [I_n - \widehat{W}_\theta \widehat{W}_\theta^\top]\widehat{V}_\theta\|_F^2 = n\|\widehat{U}_\theta^\top \widehat{P}_\theta \widehat{V}_\theta\|_F^2.$$

Set $Q := \sum_{j=1}^{n-2d} a_j \otimes a_j$ and $\Gamma := \widehat{P}_\theta - Q = \sum_{j=n-2d+1}^{n} \mu_j \, a_j \otimes a_j$. Next we get

$$\widehat{T}_n = n\|\widehat{U}_\theta^\top Q\widehat{V}_\theta\|_F^2 + 2n\langle \widehat{U}_\theta^\top \Gamma \widehat{V}_\theta, \widehat{U}_\theta^\top Q\widehat{V}_\theta\rangle_{HS} + n\|\widehat{U}_\theta^\top \Gamma \widehat{V}_\theta\|_F^2. \tag{38}$$

The next result guarantees that $\widehat{T}_n$ and $n\|\widehat{U}_\theta^\top Q\widehat{V}_\theta\|_F^2$ have the same limiting distribution.

**Lemma B.3.** *Set $Q := \sum_{j=1}^{n-2d} a_j \otimes a_j$. Assume the conditional independence assumption $x \perp \ddot{y} \mid z$. Assume that $d = o(\sqrt{n})$ and $\sqrt{d}\,\mathcal{E}_{\theta_m} \to 0$ as $m, n \to \infty$. Assume that $\sqrt{n}\|\widehat{U}_\theta^\top Q\widehat{V}_\theta\|_F$ converges in distribution. Then we have:*

$$\widehat{T}_n = n\,\|\widehat{U}_\theta^\top Q\widehat{V}_\theta\|_F^2 + o_\mathbb{P}\,(1)\,.$$

*Proof.* Equation (29) in Lemma B.2 combined with (23) guarantee that (with $r = 2d$ and $\delta = n^{-1}$)

$$\mathbb{P}\left(n\,\|\widehat{U}_\theta^\top \Gamma \widehat{V}_\theta\|_F^2 \geq \epsilon\right) \lesssim \frac{1}{\epsilon}\left[K^4 d\left(\psi_n(n^{-1}) + \mathcal{E}_m\right)^2 \left(\frac{d^2}{n} + \frac{\log^2(n)}{n}\right)\right] + \frac{1}{n}.$$

Under the assumption $d = o(\sqrt{n})$, we note that $\sqrt{d}\,\psi_n(n^{-1}) \to 0$ in view of (25). Furthermore assuming that $\sqrt{d}\,\mathcal{E}_{\theta_m} \to 0$ as $m, n \to \infty$, the previous display guarantees that

$$n\,\|\widehat{U}_\theta^\top \Gamma \widehat{V}_\theta\|_F^2 \xrightarrow{\mathbb{P}} 0, \quad \text{as } m, n \to \infty.$$

Next the Cauchy-Schwarz inequality gives

$$n|\langle \widehat{U}_\theta^\top \Gamma \widehat{V}_\theta, \widehat{U}_\theta^\top Q\widehat{V}_\theta\rangle_{HS}| \leq \sqrt{n}\|\widehat{U}_\theta^\top \Gamma \widehat{V}_\theta\|_F \cdot \sqrt{n}\|\widehat{U}_\theta^\top Q\widehat{V}_\theta\|_F$$

Assume that $\sqrt{n}\|\widehat{U}_\theta^\top Q\widehat{V}_\theta\|_F$ converges in distribution. Set $a_n := \sqrt{n}\|\widehat{U}_\theta^\top \Gamma \widehat{V}_\theta\|_F$ and $b_n := \sqrt{n}\|\widehat{U}_\theta^\top Q\widehat{V}_\theta\|_F$. By the previous analysis, $a_n \xrightarrow{\mathbb{P}} 0$. By Slutsky's theorem, $a_n b_n \xrightarrow{d} 0$, and since the limit is constant, $a_n b_n \xrightarrow{\mathbb{P}} 0$.

$\square$

**B.6. Diagonal Concentration of the Projection Matrix $Q$**

We end this subsection with a concentration result on $Q$.

Define

$$\epsilon_n(i) := \sum_{j=n-2d+1}^{n} a_{i,j}^2 \quad \forall i \in [n]. \tag{39}$$

**Lemma B.4.** *Assume that $\mathcal{E}_m + \psi_n(2\delta) < 1$. Then, with probability at least $1 - \delta$, we have*

$$\max_{i \in [n]}\{\epsilon_n(i)\} \leq \frac{c^2 K^2}{\sqrt{1 - \mathcal{E}_m - \psi_n(2\delta)}} \frac{d^2}{n} \log(4nd\delta^{-1}). \tag{40}$$

*Proof.* By definition of $\widehat{P}_\theta$ and (24), we deduce that the SVD of $\widehat{W}_\theta$ is

$$\widehat{W}_\theta = \sum_{j=1}^{2d} \sqrt{\widehat{\lambda}_j}\, a_{n-2d+j} \otimes b_j,$$

where $\{a_{n-2d+j}\}_j$ and $\{b_j\}_j$ are the empirical left and right singular vectors. This yields

$$\sqrt{\widehat{\lambda}_j}\, a_{n-2d+j} = \widehat{W}_\theta b_j = \frac{1}{\sqrt{n}} \begin{pmatrix} w_\theta(z_1)^\top \\ \vdots \\ w_\theta(z_n)^\top \end{pmatrix} b_j.$$

By the Cauchy-Schwarz inequality and concentration of $w_\theta(z_i)$, with probability at least $1 - \delta/2$, for any $i \in [n]$ and $j \in [2d]$,

$$\sqrt{\widehat{\lambda}_j}\, |a_{i,n-2d+j}| \leq \frac{|\langle w_\theta(z_i), b_j \rangle|}{\sqrt{n}} \leq \frac{\|w_\theta(z_i)\|}{\sqrt{n}} \leq cK \sqrt{\frac{d}{n} \log(4nd\delta^{-1})}.$$

By (22), we have w.p.a.l $1 - \delta/2$ that $\widehat{\lambda}_j \geq 1 - \mathcal{E}_m - \psi_n(2\delta)$ for any $j \in [2d]$. Under the assumption that $\mathcal{E}_m + \psi_n(2\delta) < 1$, we obtain $\sqrt{\widehat{\lambda}_j} \geq \sqrt{1 - \mathcal{E}_m - \psi_n(2\delta)}$.

Therefore, with probability at least $1 - \delta$, for any $i \in [n]$ and $j \in [2d]$,

$$|a_{i,n-2d+j}| \leq \frac{cK}{\sqrt{1 - \mathcal{E}_m - \psi_n(2\delta)}} \sqrt{\frac{d}{n} \log(4nd\delta^{-1})}.$$

A union bound over gives, with probability at least $1 - \delta$,

$$\max_{i \in [n], j \in [2d]} |a_{i,n-2d+j}| \leq \frac{cK}{\sqrt{1 - \mathcal{E}_m - \psi_n(2\delta)}} \sqrt{\frac{d}{n} \log(4nd\delta^{-1})}.$$

Recall $\epsilon_n(i) = \sum_{j=1}^{2d} a_{i,n-2d+j}^2$. For the previous display, we obtain w.p.a.l. $1 - \delta$:

$$\max_{i \in [n]} \{\epsilon_n(i)\} \leq 2d \cdot \max_{i \in [n], j \in [d]} a_{i,n-2d+j}^2 \leq \frac{c^2 K^2}{1 - \mathcal{E}_m - \psi_n(2\delta)} \frac{d^2}{n} \log(4nd\delta^{-1}),$$

which completes the proof. $\qquad\square$

**Asymptotic behavior.**    We now discuss the asymptotic implications of Lemma B.4. Assume that $\mathcal{E}_m + \psi_n(2n^{-1}) \leq 1/2$ and that $d^2 \log(n)/n \to 0$ as $n \to \infty$. Under these conditions, the bound in Lemma B.4 implies

$$\max_{i \in [n]} \{\epsilon_n(i)\} = O\left(\frac{d^2 \log(n)}{n}\right) \xrightarrow{\mathbb{P}} 0, \quad \text{as } n \to \infty.$$

Recalling that $[Q]_{i,i} = \sum_{j=1}^{n-2d} a_{i,j}^2 = 1 - \epsilon_n(i)$ for any $i \in [n]$, we immediately obtain

$$\min_{i \in [n]} \{[Q]_{i,i}\} = 1 - \max_{i \in [n]} \{\epsilon_n(i)\} \xrightarrow{\mathbb{P}} 1, \quad \text{as } n \to \infty. \tag{41}$$

This establishes that the diagonal entries of $Q$ converge to $1$ in probability, indicating that asymptotically, the projection matrix $Q$ becomes increasingly close to the identity. This asymptotic property is crucial for establishing consistency results in the regime where $d^2 = o(n/\log n)$.

### B.7. Proof of Theorem 4.1

*Proof of Theorem 4.1.* Throughout the proof, $c > 0$ will denote an absolute constant, which may vary from one occurrence to another.

By Lemma B.3, we have
$$\widehat{T}_n = n\, \|\widehat{U}_\theta^\top Q \widehat{V}_\theta\|_F^2 + o_{\mathbb{P}}(1).$$

**Asymptotic distribution of** $n\|\widehat{U}_\theta^\top Q \widehat{V}_\theta\|_F^2$. Set $a_j = (a_{i,j})_{i\in n} \in \mathbb{R}^n$. By definition of $Q$, we have $[Q]_{i_1,i_2} = \sum_{j=1}^{n-2d} a_{i_1,j} a_{i_2,j}$. We consider

$$
\begin{aligned}
M_n := \sqrt{n}\widehat{U}_\theta^\top Q \widehat{V}_\theta &= \frac{1}{\sqrt{n}} \sum_{j=1}^{n-2d} \sum_{i_1,i_2=1}^{n} a_{i_1,j} a_{i_2,j} u_\theta(x_{i_1}) \otimes v_\theta(\ddot{y}_{i_2}) \\
&= \frac{1}{\sqrt{n}} \sum_{i_1,i_2=1}^{n} [Q]_{i_1,i_2} u_\theta(x_{i_1}) \otimes v_\theta(\ddot{y}_{i_2}) \\
&= \frac{1}{\sqrt{n}} \sum_{i=1}^{n} [Q]_{i,i} u_\theta(x_i) \otimes v_\theta(\ddot{y}_i) + \frac{1}{\sqrt{n}} \sum_{i_1,i_2=1, i_1\neq i_2}^{n} [Q]_{i_1,i_2} u_\theta(x_{i_1}) \otimes v_\theta(\ddot{y}_{i_2}).
\end{aligned}
\tag{42}
$$

We prove that the second off-diagonal sum is negligible in front of the diagonal sum in the previous display. Indeed computing the variance of the off-diagonal term, using the tower property of conditional expectation and the conditional independence of $x_i$ and $\ddot{y}_i$ given $z_i$ under the null hypothesis, and the fact that $\{x_i, \ddot{y}_i, z_i\}_{i=1}^n$ is an i.i.d. sample set, we get:

$$
\begin{aligned}
\mathrm{Cov}&\left(\sum_{i_1\neq i_2} [Q]_{i_1,i_2} u_\theta(x_{i_1}) \otimes v_\theta(\ddot{y}_{i_2})\right) \\
&= \sum_{i_1\neq i_2} \sum_{i_3\neq i_4} [Q]_{i_1,i_2} [Q]_{i_3,i_4} \mathrm{Cov}\left(u_\theta(x_{i_1}) \otimes v_\theta(\ddot{y}_{i_2}), u_\theta(x_{i_3}) \otimes v_\theta(\ddot{y}_{i_4})\right) \\
&= \sum_{i_1\neq i_2} \sum_{i_3\neq i_4} [Q]_{i_1,i_2} [Q]_{i_3,i_4} \Big[ \mathbb{E}\left[u_\theta(x_{i_1}) \otimes v_\theta(\ddot{y}_{i_2}) \otimes u_\theta(x_{i_3}) \otimes v_\theta(\ddot{y}_{i_4})\right] \\
&\qquad\qquad - \mathbb{E}[u_\theta(x_{i_1}) \otimes v_\theta(\ddot{y}_{i_2})] \otimes \mathbb{E}[u_\theta(x_{i_3}) \otimes v_\theta(\ddot{y}_{i_4})]\Big].
\end{aligned}
$$

Using the conditional independence and i.i.d. structure, $\mathrm{Cov}\left(u_\theta(x_{i_1}) \otimes v_\theta(\ddot{y}_{i_2}), u_\theta(x_{i_3}) \otimes v_\theta(\ddot{y}_{i_4})\right)$ is non-zero only when $(i_1, i_2) = (i_3, i_4)$, giving:

$$
\mathrm{Cov}\left(\sum_{i_1\neq i_2} [Q]_{i_1,i_2} u_\theta(x_{i_1}) \otimes v_\theta(\ddot{y}_{i_2})\right) = \left(\sum_{i_1\neq i_2} [Q]_{i_1,i_2}^2\right) \mathrm{Cov}(u_\theta(x)) \otimes \mathrm{Cov}(v_\theta(\ddot{y})).
$$

Recall next that that $Q$ is an orthogonal projection of rank $n - 2d$, we have $\|Q\|_F^2 = \mathrm{tr}(Q) = n - 2d$. Hence:

$$
\begin{aligned}
\mathrm{Cov}\left(\sum_{i_1\neq i_2} [Q]_{i_1,i_2} u_\theta(x_{i_1}) \otimes v_\theta(\ddot{y}_{i_2})\right) &\preceq \left(\sum_{i_1\neq i_2} [Q]_{i_1,i_2}^2\right) \mathrm{Cov}(u_\theta(x)) \otimes \mathrm{Cov}(v_\theta(\ddot{y})) \\
&= \left(\|Q\|_F^2 - \sum_{i=1}^{n} [Q]_{i,i}^2\right) \mathrm{Cov}(u_\theta(x)) \otimes \mathrm{Cov}(v_\theta(\ddot{y})) \\
&\preceq \left((n-2d) - \min_{i\in[n]}\{Q_{i,i}\} \mathrm{tr}(Q)\right) \mathrm{Cov}(u_\theta(x)) \otimes \mathrm{Cov}(v_\theta(\ddot{y})) \\
&= \left(1 - \min_{i\in[n]}\{Q_{i,i}\}\right) (n-2d) \mathrm{Cov}(u_\theta(x)) \otimes \mathrm{Cov}(v_\theta(\ddot{y})).
\end{aligned}
\tag{43}
$$

Similarly, for the diagonal term:

$$
\begin{aligned}
\text{Cov}\left(\sum_{i=1}^{n}[Q]_{i,i}u_\theta(x_i)\otimes v_\theta(\ddot{y}_i)\right) &= \sum_{i=1}^{n}[Q]_{i,i}^2\,\text{Cov}\left(u_\theta(x_i)\otimes v_\theta(\ddot{y}_i)\right) \\
&= \left(\sum_{i=1}^{n}[Q]_{i,i}^2\right)\text{Cov}(u_\theta(x))\otimes\text{Cov}(v_\theta(\ddot{y})) \\
&\succeq \min_{i\in[n]}\{Q_{i,i}\}\,\text{tr}(Q)\,\text{Cov}(u_\theta(x))\otimes\text{Cov}(v_\theta(\ddot{y})) \\
&= \min_{i\in[n]}\{Q_{i,i}\}\,(n-2d)\,\text{Cov}(u_\theta(x))\otimes\text{Cov}(v_\theta(\ddot{y})).
\end{aligned}
\tag{44}
$$

Recall that $[Q]_{i,i} = \sum_{j=1}^{n-2d}a_{i,j}^2 = 1-\epsilon_n(i)$ for any $i\in[n]$. We proved in Lemma B.4 and in (41) that that

$$
\min_i\{[Q]_{i,i}\} \xrightarrow{\mathbb{P}} 1, \quad \text{as } n\to\infty.
$$

Thanks to the previous property, we just proved that the covariance in (43) is completely dominated and negligible in front of the covariance in (44).

To prove convergence in probability, we analyze both the mean and variance of the rescaled off-diagonal term.

**Variance analysis:** The variance of the rescaled term satisfies:

$$
\begin{aligned}
\text{Cov}\left(\frac{1}{\sqrt{n}}\sum_{i_1\neq i_2}[Q]_{i_1,i_2}u_\theta(x_{i_1})\otimes v_\theta(\ddot{y}_{i_2})\right) &= \frac{1}{n}\text{Cov}\left(\sum_{i_1\neq i_2}[Q]_{i_1,i_2}u_\theta(x_{i_1})\otimes v_\theta(\ddot{y}_{i_2})\right) \\
&\preceq \frac{1}{n}\left(1-\min_{i\in[n]}\{Q_{i,i}\}\right)(n-2d)\,\text{Cov}(u_\theta(x))\otimes\text{Cov}(v_\theta(\ddot{y})) \\
&= \frac{n-2d}{n}\left(1-\min_{i\in[n]}\{Q_{i,i}\}\right)\text{Cov}(u_\theta(x))\otimes\text{Cov}(v_\theta(\ddot{y})).
\end{aligned}
$$

Since $\min_{i\in[n]}\{Q_{i,i}\}\xrightarrow{\mathbb{P}}1$ as $n\to\infty$, and $\frac{n-2d}{n}\to 1$ under the regime $d=o(n)$, we have

$$
\frac{n-2d}{n}\left(1-\min_{i\in[n]}\{Q_{i,i}\}\right)\xrightarrow{\mathbb{P}}0.
$$

Therefore, the covariance converges to zero in probability.

**Mean analysis:** For the mean, using conditional independence for $i_1\neq i_2$:

$$
\mathbb{E}\left[\frac{1}{\sqrt{n}}\sum_{i_1\neq i_2}[Q]_{i_1,i_2}u_\theta(x_{i_1})\otimes v_\theta(\ddot{y}_{i_2})\right] = \frac{1}{\sqrt{n}}\sum_{i_1\neq i_2}[Q]_{i_1,i_2}\mathbb{E}[u_\theta(x)]\otimes\mathbb{E}[v_\theta(\ddot{y})].
$$

By the empirical centering property (31), we have w.p.a.l. $1-\delta$:

$$
\|\mathbb{E}[u_\theta(x)]\|\bigvee\|\mathbb{E}[v_\theta(\ddot{y})]\| \leq CK\left(\sqrt{\frac{d}{n}}+\sqrt{\frac{\log(\delta^{-1})}{n}}\right).
$$

Using $\|Q\|_1 \leq \|Q\|_F \sqrt{n} = \sqrt{(n-2d)n}$, we obtain:

$$\left\| \mathbb{E}\left[ \frac{1}{\sqrt{n}} \sum_{i_1 \neq i_2} [Q]_{i_1,i_2} u_\theta(x_{i_1}) \otimes v_\theta(\ddot{y}_{i_2}) \right] \right\| \leq \frac{1}{\sqrt{n}} \|Q\|_1 \|\mathbb{E}[u_\theta(x)]\| \|\mathbb{E}[v_\theta(\ddot{y})]\|$$

$$\leq \sqrt{n-2d} \cdot C^2 K^2 \left( \sqrt{\frac{2d}{n}} + \sqrt{\frac{\log(\delta^{-1})}{n}} \right)^2$$

$$= O\left( \sqrt{n-2d} \cdot \frac{2d}{n} \right) = O\left( \frac{d}{\sqrt{n}} \right) \to 0,$$

as $n \to \infty$ under the regime $d = o(\sqrt{n})$.

**Conclusion:** Since both the mean and variance of the rescaled off-diagonal term converge to zero, we conclude:

$$\frac{1}{\sqrt{n}} \sum_{i_1,i_2=1, i_1 \neq i_2}^{n} [Q]_{i_1,i_2} u_\theta(x_{i_1}) \otimes v_\theta(\ddot{y}_{i_2}) \overset{\mathbb{P}}{\to} 0, \quad \text{as } n \to \infty.$$

We consider now the diagonal term:

$$\frac{1}{\sqrt{n}} \sum_{i=1}^{n} [Q]_{i,i} u_\theta(x_i) \otimes v_\theta(\ddot{y}_i).$$

We apply the Lindeberg-Feller CLT to get the following result. See Appendix B.8 for the proof.

**Lemma B.5.** *Assume that there exists a large enough absolute constant $c > 0$ such that $n \geq c K^4 d^2 \log n$ as $n \to \infty$. Then*

$$\frac{1}{\sqrt{n}} \sum_{i=1}^{n} [Q]_{i,i} u_\theta(x_i) \otimes v_\theta(\ddot{y}_i) \overset{\mathcal{D}}{\to} \mathcal{Z} \in \mathbb{R}^{d \times d}, \quad \text{where } \mathcal{Z} \sim \mathcal{N}(0, \Sigma),$$

*with $\Sigma = \mathrm{Cov}(u_\theta(x)) \otimes \mathrm{Cov}(v_\theta(\ddot{y}))$.*

Recall that $\{\lambda_j(u_\theta)\}_{j \in [d]}$ and $\{\lambda_j(v_\theta)\}_{j \in [d]}$ are the eigenvalues of $\mathrm{Cov}(u_\theta(x_1))$ and $\mathrm{Cov}(v_\theta(\ddot{y}_1))$ respectively.

Slutsky's theorem implies the following convergence in distribution:

$$n \|\widehat{U}_\theta^\top Q \widehat{V}_\theta\|_F^2 \overset{\mathcal{D}}{\to} \sum_{i=1}^{d} \sum_{j=1}^{d} \lambda_i(u_\theta) \lambda_j(v_\theta) \chi_{i,j}^2(1) =: \xi, \tag{45}$$

as $n \to \infty$ where the $\chi_{i,j}^2(1)$ are independent random variables following a chi-square distribution of degree 1.

Noting that the $\chi_{i,j}^2(1)$ are nonnegative for any $i,j$ and assuming that $(2 + \mathcal{E}_m)\mathcal{E}_m < 1$, we get for any $t \in \mathbb{R}$ that

$$\mathbb{P}\left( \chi^2(d^2) \leq \frac{t}{1 + (2 + \mathcal{E}_m)\mathcal{E}_m} \right) \leq \mathbb{P}(\xi \leq t) \leq \mathbb{P}\left( \chi^2(d^2) \leq \frac{t}{1 - (2 + \mathcal{E}_m)\mathcal{E}_m} \right), \tag{46}$$

where $\chi^2(d^2)$ is a chi-square distribution with $d^2$ degrees of freedom.

Note that the $\chi^2(d^2)$ distribution admits the following density w.r.t the Lebesgue measure

$$f(t) = \frac{1}{2^{d^2/2} \Gamma(d^2/2)} t^{d^2/2 - 1} e^{-t/2}.$$

Since we assumed that $\epsilon := (2 + \mathcal{E}_m)\mathcal{E}_m < 1$, we have

$$\frac{t}{1 - \epsilon} = t + \frac{\epsilon t}{1 - \epsilon} = t(1 + \Diamond), \quad \text{with} \quad \Diamond = \frac{\epsilon}{1 - \epsilon}.$$

Consequently for any $t > 0$

$$\mathbb{P}\left(\chi^2(d^2) \leq t\,(1 + \Diamond)\right) = \mathbb{P}\left(\chi^2(d^2) \leq t\right) + \int_t^{t(1+\Diamond)} f(s)ds$$

We proceed similarly for the lower bound. Indeed we first note that $\frac{t}{1+\epsilon} \geq t(1 - \epsilon)$. Hence,

$$\mathbb{P}\left(\chi^2(d^2) \leq t\,(1 - \epsilon)\right) = \mathbb{P}\left(\chi^2(d^2) \leq t\right) - \int_{t(1-\epsilon)}^{t} f(s)ds.$$

Combining the last two displays with (46), we get for any $t > 0$ that

$$\left|\mathbb{P}\left(\xi \leq t\right) - \mathbb{P}\left(\chi^2(d^2) \leq t\right)\right| \leq \max\left\{\int_{t(1-\epsilon)}^{t} f(s)ds, \int_t^{t(1+\Diamond)} f(s)ds\right\} \tag{47}$$

Since $\mathcal{E}_m = \mathcal{E}_{\theta_m} \to 0$ as $m \to \infty$ and the density $f$ is uniformly bounded on $\mathbb{R}^+$, the right-hand-side in (47) goes to 0 as $m \to \infty$ for any fixed $t > 0$. This concludes the proof.

$\square$

## B.8. Proof of Lemma B.5

Define

$$Z_n := \frac{1}{\sqrt{n}} \sum_{i=1}^{n} [Q]_{i,i}\, u_\theta(x_i) \otimes v_\theta(\ddot{y}_i),$$

*Proof of Lemma B.5.* Define

$$Z_n := \frac{1}{\sqrt{n}} \sum_{i=1}^{n} [Q]_{i,i}\, u_\theta(x_i) \otimes v_\theta(\ddot{y}_i),$$

and consider the vectorized form:

$$\mathrm{vec}(Z_n) = \frac{1}{\sqrt{n}} \sum_{i=1}^{n} [Q]_{i,i} \cdot \mathrm{vec}\left(u_\theta(x_i) \otimes v_\theta(\ddot{y}_i)\right) \in \mathbb{R}^{d^2}.$$

Let us denote

$$w_i := \mathrm{vec}\left(u_\theta(x_i) \otimes v_\theta(\ddot{y}_i)\right), \quad \text{so that} \quad \mathrm{vec}(Z_n) = \frac{1}{\sqrt{n}} \sum_{i=1}^{n} [Q]_{i,i} w_i.$$

We now verify the Lindeberg condition:

$$\frac{1}{n} \sum_{i=1}^{n} \mathbb{E}\left[\|[Q]_{i,i} w_i\|^2 \cdot \mathbb{1}_{\{\|[Q]_{i,i} w_i\| > \varepsilon\sqrt{n}\}}\right] \to 0 \quad \text{for all } \varepsilon > 0.$$

Since $\|[Q]_{i,i} w_i\| = [Q]_{i,i} \cdot \|w_i\|$, and $[Q]_{i,i} \leq 1$, it suffices to show:

$$\frac{1}{n} \sum_{i=1}^{n} \mathbb{E}\left[\|w_i\|^2 \cdot \mathbb{1}_{\{\|w_i\| > \varepsilon\sqrt{n}\}}\right] \to 0.$$

Note that:

$$\|w_i\| = \|\mathrm{vec}(u_\theta(x_i) \otimes v_\theta(\ddot{y}_i))\| = \|u_\theta(x_i)\| \cdot \|v_\theta(\ddot{y}_i)\|.$$

Let us define:

$$A_i := \|u_\theta(x_i)\|, \quad B_i := \|v_\theta(\ddot{y}_i)\|, \quad \text{so that } \|w_i\| = A_i B_i.$$

Since $u_\theta(x_i)$ and $v_\theta(\ddot{y}_i)$ are $K$–sub-Gaussian, their norms $A_i$, $B_i$ are $K\sqrt{d}$–sub-Gaussian random variables. Therefore, their product $A_i B_i$ is $2K^2 d$–sub-exponential.

As a result, for all $t > 0$,

$$\mathbb{P}(A_i B_i > t) \leq c e^{-c' \frac{t}{K^2 d}},$$

for some absolute constants $c, c' > 0$.

Therefore, elementary calculus (based on integration by parts for Gamma function) gives the following upper bound

$$\mathbb{E}\left[(A_i B_i)^2 \cdot \mathbb{1}_{\{A_i B_i > \varepsilon \sqrt{n}\}}\right] \leq c K^2 d n \epsilon^2 e^{-c' \frac{\epsilon \sqrt{n}}{K^2 d}},$$

for absolute constants $c, c' > 0$.

Thus,

$$\frac{1}{n} \sum_{i=1}^n \mathbb{E}\left[\|w_i\|^2 \cdot \mathbb{1}_{\{\|w_i\| > \varepsilon \sqrt{n}\}}\right] = \mathbb{E}\left[\|w_1\|^2 \cdot \mathbb{1}_{\{\|w_1\| > \varepsilon \sqrt{n}\}}\right] \leq c K^2 d n \epsilon^2 e^{-c' \frac{\epsilon \sqrt{n}}{K^2 d}} \to 0,$$

provided that $n \geq \frac{3}{2\epsilon^2 c'} K^4 d^2 \log n$ as $n \to \infty$ so that the exponential term is dominating and thus leading the right-hand-side to 0.

Since the Lindeberg condition is satisfied, by the multivariate CLT, the sequence $\text{vec}(Z_n)$ converges in distribution to a multivariate Gaussian random vector. Therefore,

$$Z_n \xrightarrow{d} \mathcal{Z} \in \mathbb{R}^{d \times d}, \quad \text{where } \text{vec}(\mathcal{Z}) \sim \mathcal{N}(0, \Sigma),$$

with

$$\Sigma = \text{Cov}(u_\theta(x)) \otimes \text{Cov}(v_\theta(\ddot{y})).$$

$\square$

### B.9. Proof of Theorem 4.2

Recall that

$$[\![\Sigma_{X\ddot{Y} \cdot Z}]\!]_d = C_{UV} - C_{UW} C_{WV}.$$

Assuming the representation learning step has been successful, we can assume that (10) is valid with small $\mathcal{E}_m$ for learned features $\widehat{U}_\theta$, $\widehat{V}_\theta$ and $\widehat{W}_\theta$, meaning that

$$\| [\![\Sigma_{X\ddot{Y} \cdot Z}]\!]_d - \left(C_{\widehat{U}_\theta \widehat{V}_\theta} - C_{\widehat{U}_\theta \widehat{W}_\theta} C_{\widehat{W}_\theta \widehat{V}_\theta}\right) \| \leq \mathcal{E}_m \ll 1.$$

We have

$$\widehat{C}_{\widehat{U}_\theta \widehat{V}_\theta} - \widehat{C}_{\widehat{U}_\theta \widehat{W}_\theta} \widehat{C}_{\widehat{W}_\theta \widehat{V}_\theta} = [\![\Sigma_{X\ddot{Y} \cdot Z}]\!]_d + \left(C_{\widehat{U}_\theta \widehat{V}_\theta} - C_{\widehat{U}_\theta \widehat{W}_\theta} C_{\widehat{W}_\theta \widehat{V}_\theta}\right) - [\![\Sigma_{X\ddot{Y} \cdot Z}]\!]_d)$$
$$= [\![\Sigma_{X\ddot{Y} \cdot Z}]\!]_d + (\widehat{C}_{\widehat{U}_\theta \widehat{V}_\theta} - C_{\widehat{U}_\theta \widehat{V}_\theta}) - (\widehat{C}_{\widehat{U}_\theta \widehat{W}_\theta} - C_{\widehat{U}_\theta \widehat{W}_\theta}) \widehat{C}_{\widehat{W}_\theta \widehat{V}_\theta} + C_{\widehat{U}_\theta \widehat{W}_\theta} (C_{\widehat{W}_\theta \widehat{V}_\theta} - \widehat{C}_{\widehat{W}_\theta \widehat{V}_\theta}).$$

Taking the Frobenius norm and using the triangular inequality, we get

$$\|\widehat{C}_{\widehat{U}_\theta \widehat{V}_\theta} - \widehat{C}_{\widehat{U}_\theta \widehat{W}_\theta} \widehat{C}_{\widehat{W}_\theta \widehat{V}_\theta}\|_F \geq \| [\![\Sigma_{X\ddot{Y} \cdot Z}]\!]_d \|_F - \|\left(C_{\widehat{U}_\theta \widehat{V}_\theta} - C_{\widehat{U}_\theta \widehat{W}_\theta} C_{\widehat{W}_\theta \widehat{V}_\theta}\right) - [\![\Sigma_{X\ddot{Y} \cdot Z}]\!]_d)\|_F$$
$$- \|\widehat{C}_{\widehat{U}_\theta \widehat{V}_\theta} - C_{\widehat{U}_\theta \widehat{V}_\theta}\|_F - \|(\widehat{C}_{\widehat{U}_\theta \widehat{W}_\theta} - C_{\widehat{U}_\theta \widehat{W}_\theta}) \widehat{C}_{\widehat{W}_\theta \widehat{V}_\theta}\|_F - \|C_{\widehat{U}_\theta \widehat{W}_\theta} (C_{\widehat{W}_\theta \widehat{V}_\theta} - \widehat{C}_{\widehat{W}_\theta \widehat{V}_\theta})\|_F$$
$$\geq \| [\![\Sigma_{X\ddot{Y} \cdot Z}]\!]_d \|_F - \|\left(C_{\widehat{U}_\theta \widehat{V}_\theta} - C_{\widehat{U}_\theta \widehat{W}_\theta} C_{\widehat{W}_\theta \widehat{V}_\theta}\right) - [\![\Sigma_{X\ddot{Y} \cdot Z}]\!]_d\|_F$$
$$- \|\widehat{C}_{\widehat{U}_\theta \widehat{V}_\theta} - C_{UV}\|_F - \|(\widehat{C}_{\widehat{U}_\theta \widehat{W}_\theta} - C_{UW})\|_F \|\widehat{C}_{\widehat{W}_\theta \widehat{V}_\theta}\| - \|C_{UW}\| \|(C_{WV} - \widehat{C}_{\widehat{W}_\theta \widehat{V}_\theta})\|_F.$$

Using (10), we get

$$\| [\![\Sigma_{X\ddot{Y} \cdot Z}]\!]_d - \left(C_{\widehat{U}_\theta \widehat{V}_\theta} - C_{\widehat{U}_\theta \widehat{U}_\theta} C_{\widehat{W}_\theta \widehat{V}_\theta}\right)\|_F \leq \sqrt{d} \| [\![\Sigma_{X\ddot{Y} \cdot Z}]\!]_d - \left(C_{\widehat{U}_\theta \widehat{V}_\theta} - C_{\widehat{U}_\theta \widehat{U}_\theta} C_{\widehat{W}_\theta \widehat{V}_\theta}\right)\| \leq \sqrt{d} \mathcal{E}_m.$$

Furthermore, exploiting the sub-Gaussian properties of features (Assumption 4.1), standard matrix concentration inequalities for the cross-variance estimators and an union bound, we obtain w.p.a.l. $1 - \delta$

$$\frac{\|\widehat{C}_{\widehat{U}_\theta \widehat{V}_\theta} - C_{\widehat{U}_\theta \widehat{V}_\theta}\|}{\|C_{\widehat{U}_\theta \widehat{V}_\theta}\|} \vee \frac{\|\widehat{C}_{\widehat{U}_\theta \widehat{W}_\theta} - C_{\widehat{U}_\theta \widehat{W}_\theta}\|}{\|C_{\widehat{U}_\theta \widehat{W}_\theta}\|} \vee \frac{\|\widehat{C}_{\widehat{W}_\theta \widehat{V}_\theta} - C_{\widehat{W}_\theta \widehat{V}_\theta}\|}{\|C_{\widehat{W}_\theta \widehat{V}_\theta}\|} \lesssim K^2 \left( \sqrt{\frac{d}{n}} \vee \sqrt{\frac{\log(\delta^{-1})}{n}} \right) =: \psi_n(\delta),$$

for some large enough absolute constant $C > 0$.

Using (10) again, we get $\|C_{\widehat{U}_\theta \widehat{V}_\theta}\| \leq \sqrt{\|C_{\widehat{U}_\theta \widehat{U}_\theta}\|}\sqrt{\|C_{\widehat{V}_\theta \widehat{V}_\theta}\|} \leq 1 + \mathcal{E}_m$. We have the same upper bound for all 3 cross-covariances in operator norm.

Combining the last four displays, we get w.p.a.l. $1 - \delta$

$$\|\widehat{C}_{\widehat{U}_\theta \widehat{V}_\theta} - \widehat{C}_{\widehat{U}_\theta \widehat{W}_\theta} \widehat{C}_{\widehat{W}_\theta \widehat{V}_\theta}\|_F \geq \|[\![\Sigma_{X\ddot{Y} \cdot Z}]\!]_d\|_F - \sqrt{d}\left(\mathcal{E}_m + (1 + \mathcal{E}_m)\psi_n(\delta) + 2(1 + \mathcal{E}_m)^2 \psi_n(\delta)\right)$$
$$\geq \|[\![\Sigma_{X\ddot{Y} \cdot Z}]\!]_d\|_F - \sqrt{d}\left(\mathcal{E}_m + 3(1 + \mathcal{E}_m)^2 \psi_n(\delta)\right).$$

We recall that

$$\|[\![\Sigma_{X\ddot{Y} \cdot Z}]\!]_d\|_F^2 = \sum_{j=1}^{d} \sigma_j^2,$$

and $\Sigma_{X\ddot{Y} \cdot Z}$ is assumed Hilbert-Schmidt. This implies that if we take $d$ large enough, then we capture enough signal, that is $\sum_{j=1}^{d} \sigma_j^2 \geq (\sum_{j=1}^{\infty} \sigma_j^2)/2 = \|\Sigma_{XY \cdot Z}\|_F^2/2 = \epsilon_n^2/2$. Hence the smoothness of the operator $\Sigma_{XY \cdot Z}$ measured through the decay rate of its eigenvalues provides a theoretical guidelines on the choice of $d$ and the necessary number of samples to obtain guarantees on the power of the test.

Recall that $c_\alpha$ is the quantile of the chi-square distribution $\chi_{d^2}^2$. Hence if

$$\epsilon_n^2 > 2d\,\mathcal{E}_m^2 + \frac{2C^2 K^2 (d^2 + d\log(\delta^{-1})) + 2c_\alpha}{n},$$

then we get

$$\mathbb{P}_{\mathcal{H}_1}\left(\widehat{T}_n > c_\alpha\right) \geq 1 - \delta.$$

Using the large-$d^2$ quantile approximation of chi-square distribution $\chi_{d^2}^2$: $c_\alpha \approx d^2 + 2dz_{1-\alpha}$, where $z_{1-\alpha}$ is the quantile of the standard normal. Hence the threshold on $\epsilon_n$ becomes :

$$\epsilon_n^2 > 2d\,\mathcal{E}_m^2 + \frac{2C^2 K^2 (d^2 + d\log(\delta^{-1})) + 2d^2 + 4dz_{1-\alpha}}{n}.$$

We use the normal approximation for the $\chi_{d^2}^2$ quantile:

$$c_\alpha \approx d^2 + 2d\,z_{1-\alpha},$$

where $z_{1-\alpha}$ is the $(1 - \alpha)$ standard normal quantile. This gives the condition

$$\epsilon_n^2 > d\,\mathcal{E}_m^2 + \frac{C^2 K^2 (d^2 + d\log(\delta^{-1})) + d^2 + 2d\,z_{1-\alpha}}{n}.$$

This inequality balances three things:

1. the signal strength $\epsilon_n^2 = \|\Sigma_{X\ddot{Y} \cdot Z}\|_F^2$,

2. the error from representation learning ($d\,\mathcal{E}_m^2$), and

3. the statistical error from estimating covariances, which scales like $d^2/n$ (plus logarithmic terms).

For the test to have non-vacuous power, we need the signal to dominate both the representation error and the estimation noise. In particular, if

$$\epsilon_n^2 \gtrsim d\,\mathcal{E}_m^2 + \frac{d^2 + d\log(\delta^{-1})}{n},$$

then the test achieves power at least $1 - \delta$.

## C. Experimental Details

**Implementation details.** To parameterize $u_\theta$, $v_\theta$, and $w_\theta$, we used multilayer perceptrons (MLPs) with `n_hidden_{u,v,w}` hidden layers, each containing `layer_size_{u,v,w}` neurons. All networks used `Tanh` activation functions and Xavier initialization. Optimization was performed using Adam with default hyperparameters, except for the learning rate, which is reported in Tab. 2. Training was carried out with varying `batch_size` as reported in Tab. 2 for 400 epochs. The inner model was warmed up for 100 steps. During bi-level optimization, we alternated a single update of the inner model with a single update of the outer model. The dataset was split into 80% training and 20% test sets.

**Dimension pruning.** For added stability, all models were trained using a latent dimension of `output_dim`. At test time, we computed a lower-rank test statistic by performing an SVD of the test-statistic matrix and retaining only the leading $\lfloor$`perc_dim_prune` $\times$ `output_dim`$\rfloor$ singular triplets. The resulting statistic was evaluated using a corrected $\chi^2$ distribution with degrees of freedom equal to the retained (pruned) dimension.

**Hyperparameter optimization.** There is no standard protocol for performing hyperparameter optimization of conditional independence tests. In this work, we either use established hyperparameter values or follow a shared tuning protocol with equal computational budgets for all compared methods. Only hyperparameters reported in the original papers and exposed in the code repositories were varied. For example, **DGCIT** does not expose architectural hyperparameters it in its training algorithm, which limits its adaptivity and our evaluation. We deliberately avoided altering or re-implementing standard code.

Given these considerations, hyperparameters were selected as follows. For kernel-based methods, we employed a Gaussian kernel, with the length scale selected using the median heuristic, following prior work (Bellot & van der Schaar, 2019; Shi et al., 2021; Scetbon et al., 2022; Strobl et al., 2019). For **NNLSCIT**, we adopted the hyperparameters reported in the original paper for the same data-generating model (Li et al., 2023). The methods **SpectralCIT**, **RCIT**, **GCIT**, and **DGCIT** were tuned using Weights & Biases (Biewald, 2020) with `method` set to `bayes` and a budget of 30 trials over the hyperparameter grids reported in Tabs. 2 to 5. For **GCIT** and **DGCIT**, the listed hyperparameters correspond to those identified as most influential by the authors, while all other hyperparameters were fixed to the values used in the original papers (Bellot & van der Schaar, 2019; Shi et al., 2021).

$$S_{\text{val}}(m) = \widehat{\alpha}(m) - \widehat{\pi}(m),$$

where $\widehat{\alpha}$ and $\widehat{\pi}$ denote the empirical type I error and power of a model $m$, respectively. The sample size was set to $N = 1000$. For each trial, $\widehat{\alpha}$ and $\widehat{\pi}$ were estimated from 20 repetitions under both $\mathcal{H}_0$ and $\mathcal{H}_1$ for each conditioning dimension $d_Z \in \{50, 100, 150, 200, 250, 300\}$, and the resulting values were averaged across $d_Z$.

*Table 2.* Hyperparameter grid for **SpectralCIT**.

| Hyperparameter | Values | Description |
|---|---|---|
| `output_dim` | $2, 4, 6, 8, 10$ | SVD truncation rank $d$ |
| `perc_dim_prune` | $\text{Uniform}(0.85, 1)$ | Percentage of dimensions to prune |
| `n_hidden_{u,v,w}` | $1, 2, 3, 4$ | Number of hidden layers of $u_\theta, v_\theta, w_\theta$ |
| `layer_size_{u,v,w}` | $128, 256, 512$ | Layer size of $u_\theta, v_\theta, w_\theta$ |
| `lr_{inner,outer}` | $\text{LogUniform}(3 \times 10^{-5}, 1 \times 10^{-2})$ | Learning rate for inner and outer models |
| `reg_str_{inner,outer}` | $\text{LogUniform}(1, 100)$ | Regularization strength for inner and outer models |
| `batch_size` | $128, 256, 512, 1024$ | Batch size |

*Table 3.* Hyperparameter grid for **RCIT**.

| Hyperparameter | Values | Description |
|---|---|---|
| `approx` | Lindsay-Pilla-Basak, Bootstrap | Null approximation scheme |
| `num_f` | $50, 100, 150, 200$ | Number of Fourier features for $Z$ |
| `num_f2` | $5, 10, 15, 20$ | Number of Fourier features for $\ddot{X}, X, Y$ |

**Computational resources.** All experiments were conducted on a machine running Ubuntu 22.04.5 LTS with Linux kernel 5.15.0-134-generic. The system was equipped with two Intel Xeon E5-2695 v4 CPUs (36 physical cores total) and 251 GB

*Table 4.* Hyperparameter grid for **GCIT**.

| Hyperparameter | Values | |
| --- | --- | --- |
| lamda | $0, 5, 10, 15, 20$ | Regularization parameter $\lambda$ |
| statistic_type | MMD, PCC, DC, KS, RDC | Statistic |

*Table 5.* Hyperparameter grid for **DGCIT**.

| Hyperparameter | Values | |
| --- | --- | --- |
| batch_size | 16, 32, 64 | Batch size |
| b | 30, 35, 40, 45, 50 | Number of transformation functions $B$ |
| j | 500, 750, 1000, 1250 | Number of bootstrap samples $J$ |

of system memory. GPU acceleration was provided by seven NVIDIA GeForce GTX 1080 Ti GPUs, each with 11 GB of memory. The software stack consisted of Python 3.12.9, PyTorch 2.6.0, and CUDA 12.4.

**Hyperparameter sensitivity.** We evaluate the robustness of **SpectralCIT** with respect to its main hyperparameters: the truncation dimension $d$, the learning rate $\eta$, and the regularization strength $\gamma$. For the truncation dimension, we vary $d$ around the nominal choice $d_* = 10$ used in the post-nonlinear benchmark (cf. Fig. 2). As shown in Fig. 6, performance stabilizes for $d \geq 8$, with well-controlled Type I error and consistently high power across conditioning dimensions. For optimization, we scale the learning rate as $\eta = \alpha \eta_*$, where $\eta_* = (\eta_{*,\text{inner}}, \eta_{*,\text{outer}}) = (3 \times 10^{-5}, 2.1 \times 10^{-3})$ is the reference value from Fig. 2, and $\alpha \in \{0.1, 0.3, 1, 3, 10\}$. Figure 7 shows that Type I error remains controlled for $\alpha \geq 0.3$, with only minor degradation in power at extreme values, indicating limited sensitivity once $\eta$ is sufficiently large. Similarly, we vary the regularization strength as $\gamma = \alpha \gamma_*$, with $\alpha \in \{0.1, 0.3, 1, 10\}$ and $\gamma_* = (\gamma_{*,\text{inner}}, \gamma_{*,\text{outer}}) = (3.3, 1.9)$. As shown in Fig. 8, both Type I error and power remain stable across this range. Overall, these results indicate that **SpectralCIT** is robust to moderate perturbations of $d$, $\eta$, and $\gamma$, with a broad range of values yielding comparable performance.

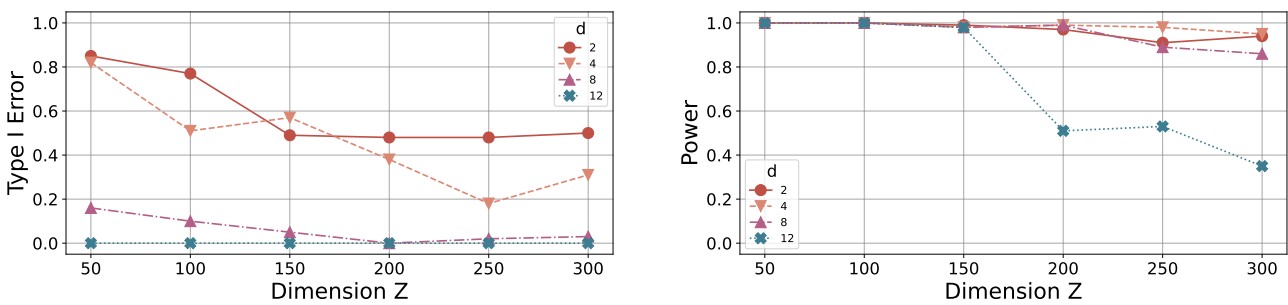

*Figure 6.* Type I error and power of our method (**SpectralCIT**) varying truncation dimension $d$ around $d_* = 10$.

**Computational complexity.** We further examine the computational cost of **SpectralCIT** relative to existing kernel-based CI tests. Experiments are conducted on the post-nonlinear model of Shi et al. (2021), using the same setup as in Fig. 5. Each configuration is repeated 10 times, and we report mean and standard deviation results in log seconds. We fix the sample size to $N = 1000$ and vary the conditioning dimension $d_Z \in \{50, 100, 200, 300\}$. As shown in Fig. 9, **SpectralCIT** is consistently about $2\times$ faster than **KCIT** across all values of $d_Z$. This behavior is expected: although **SpectralCIT** incurs an additional cost for representation learning, this cost is amortized, and the overall complexity is dominated by low-rank operations in the truncation dimension $d$, rather than by kernel matrix computations scaling with $N$. While **RCIT** achieves lower runtime, it fails to maintain Type I error control in high dimensions (cf. Fig. 10). Overall, **SpectralCIT** provides the best trade-off between computational efficiency and validity.

**New high-dimensional benchmark.** To strengthen the empirical evaluation, we introduce a high-dimensional benchmark in which the dimensions of $X$ and $Y$ scale with that of $Z$, and the shared noise has low variance (standard deviation 0.15).

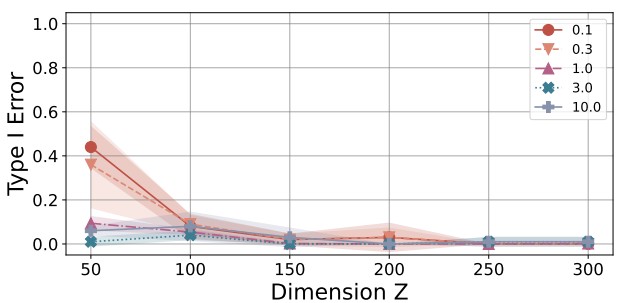 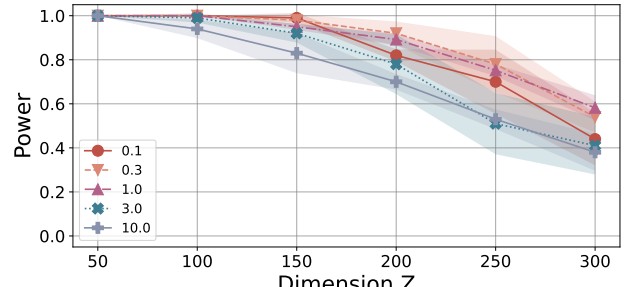

*Figure 7.* Type I error and power of our method (**SpectralCIT**) varying the learning rate $\eta$ as $\alpha\eta_*$, where $\alpha \in [0.1, 0.3, 1, 3, 10]$.

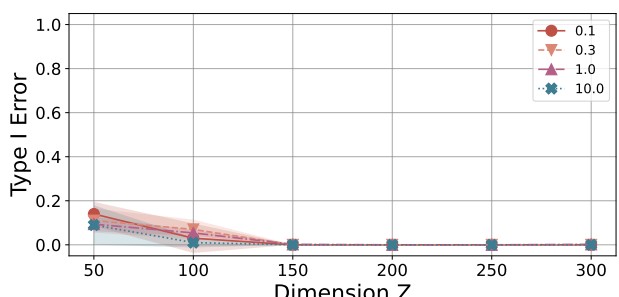 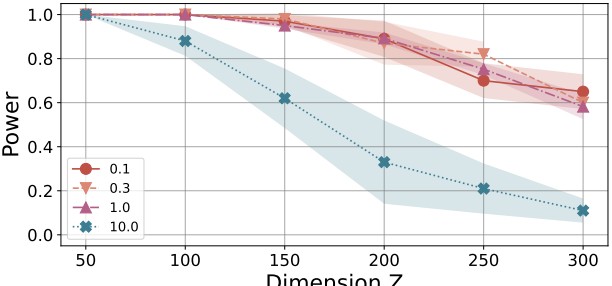

*Figure 8.* Type I error and power of our method (**SpectralCIT**) varying the regularization strength $\gamma$ as $\alpha\gamma_*$, where $\alpha \in [0.1, 0.3, 1, 10]$.

We consider the data-generating process

$$\mathcal{H}_0 : X = \sin(Z + \varepsilon_X), \quad Y = \cos(Z + \varepsilon_Y),$$
$$\mathcal{H}_1 : X = \sin(Z + \varepsilon_X) + \eta, \quad Y = \cos(Z + \varepsilon_Y) + \eta,$$

where $Z \sim \mathcal{N}(0, \texttt{str\_Z}^2 I_{d_z})$, $\varepsilon_X \sim \mathcal{N}(0, \texttt{noise\_str}^2 I_{d_z})$, $\varepsilon_Y \sim \mathcal{N}(0, \texttt{noise\_str}^2 I_{d_z})$, and $\eta \sim \mathcal{N}(0, \texttt{str\_con\_dep}^2 I_{d_z})$ are independent. We set $\texttt{str\_Z} = 0.1$, $\texttt{noise\_str} = 0.25$, and $\texttt{str\_cond\_dep} = 0.15$. For each dimension, we run each method over 500 repetitions and report the mean and standard error of the Type I error and power (see Fig. 10). Hyperparameters are identical to those in the original benchmark. In this more challenging regime, competing methods exhibit loss of validity, whereas **SpectralCIT** remains valid, albeit slightly conservative.

**On the role of signal strength.** We present a final ablation showcasing that the dominant factor governing the power of our method is the signal strength under the alternative, rather than the dimensionality of $Z$.

Concretely, using the same nonlinear equations as in the high-dimensional benchmark, but now with $d_Z = 3$, we isolate the effect of signal strength in a controlled low-dimensional setting. We vary the signal strength parameter $\texttt{str\_cond\_dep} \in \{0.05, 0.15, 0.5\}$ while keeping $\texttt{str\_Z}$ and $\texttt{noise\_str}$ fixed as before. Fig. 11 shows that, as the signal increases, the power of the test improves substantially, while Type I error remains controlled throughout. In particular, weak-signal regimes yield low power, whereas moderate-to-strong signal regimes recover high power.

These observations are consistent with our theoretical guarantees (cf. Theorem 4.2), which predict that power is governed by the signal strength relative to representation error. This experiment therefore provides empirical support that signal strength, rather than dimensionality, is the primary driver of performance in this setting.

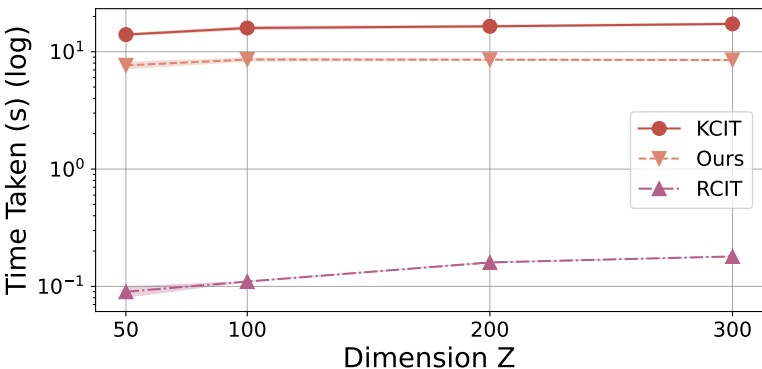

*Figure 9.* Computational time of our method in log-seconds against other kernel methods.

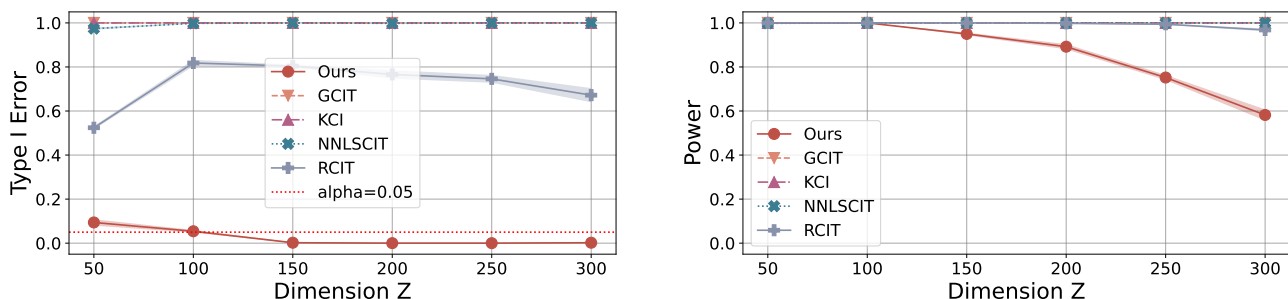

*Figure 10.* Type I error and power of our method (**SpectralCIT**) compared to state-of-the-art CI tests across varying dimensionality of the conditioning set $Z$ on new high-dimensional benchmark. We repeated each experiment 5 times, reporting mean and standard deviation.

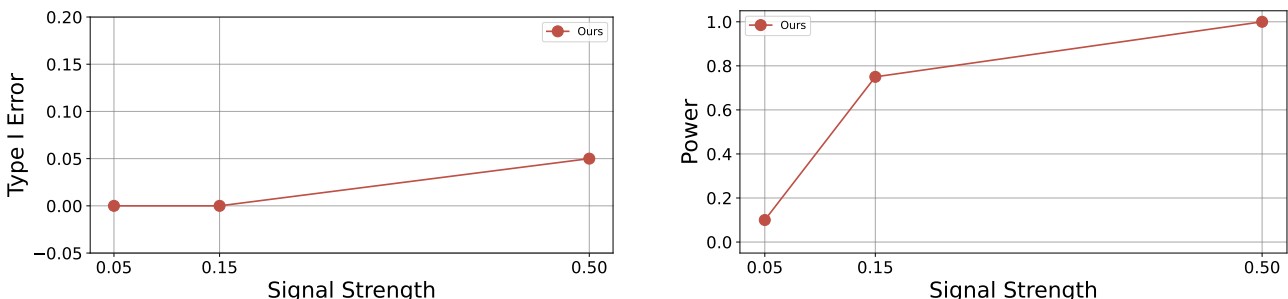

*Figure 11.* Type I error and power of our method (**SpectralCIT**) varying signal strength `str_cond_dep` $\in \{0.05, 0.15, 0.5\}$ on data model with $d_Z = 3$.

**CI queries on a Sachs-inspired model.** We conduct a small-scale experiment to evaluate **SpectralCIT** as a conditional independence (CI) oracle within the PC algorithm. The goal is not a full PC benchmark, but to assess reliability in the regime of large conditioning sets, where kernel-based methods typically degrade. The setup is based on the 7-node Sachs-inspired DAG shown in Fig. 12 (cf. (Sachs et al., 2005)), with post-nonlinear structural equations as follows. Under $\mathcal{H}_0$, there are no additional edges beyond those in $\mathcal{G}$:

$$
\begin{aligned}
Z_1 &\sim \mathcal{N}(0, I_{d_z}), \\
X_i &= \tanh(\bar{Z}_1 + \varepsilon_i), \quad i = 1, 2, \\
X_j &= A_j Z_1 + \varepsilon_j, \quad j = 3, 4, \\
Y_2 &= \cos(\bar{X}_3 + \bar{X}_4 + \varepsilon_{Y_2}),
\end{aligned}
$$

where all noise terms $\varepsilon_. \sim \mathcal{N}(0, 0.25)$ are independent. Under $\mathcal{H}_1$, we plant one additional edge $X_1 - Y_2$ to create a spurious dependency between the two downstream branches by adding a shared noise $\eta \sim \mathcal{N}(0, 0.25)$ between $X_1$ and $Y_2$.

We evaluate three CI queries representative of those a constraint-based algorithm would encounter, under both $\mathcal{H}_0$ and $\mathcal{H}_1$. Under $\mathcal{H}_1$, we plant the direct edge $X_1 \to Y_2$. Results are in Tab. 6. Type I error is 0.00 on all three queries. For $X_1 \perp Y_2 \mid X_3, X_4$ and $X_1 \perp Y_2 \mid X_3, X_4, Z_1$, the planted edge renders $\mathcal{H}_1$ non-trivial: power is 1.00 and 0.75 respectively, the latter at $d_z = 100$ where adding the hub to the conditioning set makes detection harder. For $Y_1 \perp Y_2 \mid X_1, X_2, X_3, X_4$, the planted edge does not alter the null, so $\mathcal{H}_1$ coincides with $\mathcal{H}_0$ and only Type I error is relevant.

While limited in scope, these results support the relevance of SpectralCIT to constraint-based causal discovery, particularly in the large-conditioning-set regime.

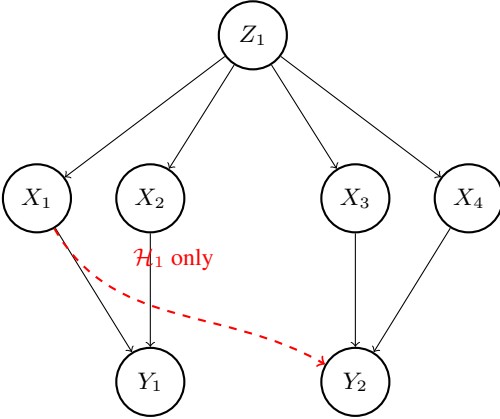

*Figure 12.* Sachs-inspired DAG on 7 nodes. $Z_1$ acts as a central hub (analogous to PKA), with four children $X_1, \ldots, X_4$. The nodes $Y_1$ and $Y_2$ are downstream outcomes with two parents each, creating v-structures. The dashed red edge $X_1 \to Y_2$ is present only under the alternative hypothesis $\mathcal{H}_1$.

*Table 6.* Results for three CI queries over the Sachs-inspired causal graph with $d_z = 100$.

| CI query | Type I error ($\mathcal{H}_0$) | Power ($\mathcal{H}_1$) |
|---|---|---|
| $X_1 \perp\!\!\!\perp Y_2 \mid \{X_3, X_4\}$ | 0.00 | 1.00 |
| $X_1 \perp\!\!\!\perp Y_2 \mid \{X_3, X_4, Z_1\}$ | 0.00 | 0.75 |
| $Y_1 \perp\!\!\!\perp Y_2 \mid \{X_1, X_2, X_3, X_4\}$ | 0.00 | – |

