# OpenReview forum: "Toward Scalable and Valid Conditional Independence Testing with Spectral Representations"
_ICML.cc/2026/Conference — ICML 2026 regular_

### Official Review · Reviewer_CgHX · 2026-02-23

**Soundness:** 2
**Presentation:** 2
**Significance:** 2
**Originality:** 2
**Overall Recommendation:** 4
**Confidence:** 2

**Summary:**

This paper develops a scalable and statistically valid CI test for nonparametric models by combining operator-theoretic kernel ideas with modern spectral representation learning. The proposed CI test  learns low-rank spectral representations of the partial covariance operator and then performing CI testing using a simple chi-square statistic constructed from learned representations. Theoretical guarantees on type I error and power, explicitly linked to representation learning error, are provided. Empirical validation on synthetic and real data is given.

**Compliance With Llm Reviewing Policy:**

Affirmed.

**Final Justification:**

The authors have addressed my concerns in the rebuttal. I keep my positive score unchanged. I hope the arthors include these discussions in the revised manuscript.

**Key Questions For Authors:**

- What is $\widehat{C}_{UU}$ in the defn of $\widehat{\Omega}$? I guess it is the smaple covariance.
- What is the difference between the experiments in Fig 2 and Fig 5? What is the purpose of Fig 5 that is not illustrated by Fig 2?
- In Fig 2, it seems most methods are independent with the dimension of $Z$, actually it looks like their performances are improving with dimension of $Z$ growing?
- For the experiments in Fig 3, why not compare all the CI tests?
- One of the motivations of the paper is to overcome the scalability and adaptivity issue of kernel methods.  The experiments only show for testing performance. How is the comparison on computation?

**Limitations:**

The paper did not discuss limitations.

**Strengths And Weaknesses:**

**Strength:**
- Classical kernel-based tests suffer from scalability and slow convergence.These are the main motivations of the paper. The operator-theoretic framing is conceptually clean and powerful.
- The asymptotic distribution of the statistic under null is chi squared, which is easy to use and avoids permutations and heavy sampling.
- The empirical results provide evidence of the proposed test showing advantage over existing baselines.

**Weakness:**
- The proposed CI test involves a tuning parameter $d$, rank of approximation. How do we choose it in practice?
- The authors want to justify the intuition why using covariance of representation embedding can give a consistent CI test, How they can help address the issues of kernel based methods.
- It is better to use $X,Y,Z$  instead of $A,B,C$ for random variables in section 2 for notation consistency and avoid confusion.

---

> ### Author Rebuttal · Authors · 2026-03-31
>
> We thank the reviewer for the positive assessment of the paper and for highlighting several important aspects of the method. We appreciate the thoughtful questions and clarify them below.
>
> ## On the choice of the truncation parameter d
> The hyperparameter d reflects the effective rank of the partial covariance operator, i.e., the number of dominant components carrying dependence. We tune it as an hyperparameter of our method. See App C for details.
>
> ## On the intuition behind consistency
> The method can be viewed as learning a low-dimensional feature space in which conditional independence reduces to testing whether a covariance vanishes.
>
> Under the null, the partial covariance operator is zero. In this framework, conditional independence is equivalent to the vanishing of this operator, so testing reduces to checking whether a covariance is zero in the learned representation. Any approximately orthonormal representation aligned with the geometry of the underlying $L^2$ spaces then yields a statistic with the correct $\chi^2$ behavior. Under alternatives, dependence is captured by the leading spectral components, and consistency follows from their approximation.
>
> This provides a scalable alternative to kernel methods, replacing fixed kernels and large Gram matrices with a learned low-dimensional representation aligned with the data.
>
> ## On notation
> We thank the reviewer for this suggestion and will standardize notation in Section 2.
>
> ## Q1. On $\widehat{C}_{UU}$
> Thanks for pointing this out. $\widehat{C}_{UU}$ denotes the sample covariance of the learned features $\widehat{u}(X)$. While it is defined in Appendix, we will make sure to write this explicitly in main body.
>
> ## Q2. On Figures 2 and 5
> Both figures follow the same protocol but use different data-generating models.
> Figure 2 corresponds to a standard benchmark used in prior work. Figure 5 uses a more challenging model to assess robustness to increasing $\dim(Z)$.
>
> In Figure 5, our method maintains controlled Type I error and high power (above 80% up to $\dim(Z)\leq 250$), while RCIT shows inflated Type I error and reduced power, and KCIT loses validity. We will clarify the role of each figure.
>
> ## Q3. On the effect of increasing dim(Z)
> Thank you for raising this point. We agree that the post-nonlinear benchmark in Fig. 2 is not especially challenging: under this data model the alternative is strong, so several methods attain power close to 1, and the main differences are therefore visible in Type I error rather than power. We include this benchmark for comparability with existing CI-testing work, where this synthetic setting is commonly used (Yang 2025; Scetbon 2021; Li 2023; Ren 2025; Zhao 2026).
>
> To strengthen the empirical evaluation, we also introduce a new high-dimensional benchmark in which the dimensions of X and Y scale with Z and the shared noise has low variance (std err of 0.15). For each dimension, we repeat every method 500 times and report both the mean and standard error of Type I error and power. These results are shown in Fig. 7 of the anonymous repository. Hyperparameters are kept the same as in the original benchmark, with full details given in App. C. In this more challenging regime, competing methods lose validity, whereas SpectralCIT remains valid, albeit slightly conservative.
>
> ## Q4. On comparison with CI tests (Figure 3)
> Figure 3 highlights robustness to changes in the data-generating assumptions. NNLSCIT relies on structural assumptions (e.g., smooth conditional relationships) that are violated in this setting, leading to Type I error close to 1 across all sample sizes, indicating a loss of validity. In contrast, our method maintains controlled Type I error while retaining competitive power across all settings considered.
>
> ## Q5. On computational comparison
> Our method shifts cost to a one-time representation learning phase, after which testing is low-dimensional. Kernel methods require large matrix operations and additional calibration steps. As described in our answer to reviewer GEcD, experimentally our method is faster than KCIT, but slower than its random Fourier feature version, which we stress loses power in high-dimensional regimes. Thus SpectralCIT offers a good trade-off between time complexity and statistical power/validity in high-dimensional settings. We will clarify this trade-off.
>
> ## On limitations
> We will add a discussion on dependence on representation learning quality and potential finite-sample conservativeness.
>
> ## Additional references
> - Ren, Yixin, et al. "Score-based Generative Modeling for Conditional Independence Testing." Proceedings of the 31st ACM SIGKDD Conference on Knowledge Discovery and Data Mining V. 2. 2025.
> - Zhao, Shunyu, et al. "Fast Flow Matching based Conditional Independence Tests for Causal Discovery." arXiv preprint arXiv:2602.08315 (2026).
>
> ## Rebuttal figures
> - https://anonymous.4open.science/r/SCIT-ICML-Rebuttal-2026-F07F/ICML_Rebuttal_2026_CIT.pdf

---

> > ### Author Rebuttal · Reviewer_CgHX · 2026-04-03
> >
> > Thanks to the authors from the response, which has addressed my questions. The manuscript would benefit from adding these discussions. I keep my score unchanged.

---

### Official Review · Reviewer_GEcD · 2026-03-11

**Soundness:** 3
**Presentation:** 2
**Significance:** 3
**Originality:** 3
**Overall Recommendation:** 4
**Confidence:** 3

**Summary:**

This paper aims to address the scalability and adaptivity issues in Conditional Independence (CI) testing. The author(s) introduce a spectral representation learning procedure, SpectralCIT, to construct a kernel-based test using the low-rank approximation of the partial covariance operator. The proposed approach addresses the adaptivity issue in kernel-based conditional independent test by adding a flexible rank-d approximation. The introduction of representation learning to find the low-rank approximation is a key contribution. The proposed method shows promising Type I error control even under high-dimensional settings. While the idea introduced in the paper is novel, there are several points regarding theoretical results, presentation, implementation, and hyperparameter sensitivity that require further clarification.

**Compliance With Llm Reviewing Policy:**

Affirmed.

**Final Justification:**

The paper addresses the adaptivity of kernel-based methods for conditional independence testing by introducing a spectral representation learning approach. The methodology is novel in the context of conditional independence testing. However, the technical conditions underlying the theoretical results remain high-level and depend on the learned representations, leaving the detectable rates unspecified. Given the overall contribution, I will maintain my original recommendation.

**Key Questions For Authors:**

1. The assumption that the approximation error $\mathcal{\varepsilon}_m$ goes to zero is crucial, but it is quite vague in the sense that no rate is specified, and in what sense (in probability, almost surely or what), for what functional spaces, the proposed representation learning can approximate the partial covariance operator sufficiently well.

2. The theoretical assumptions in Theorems 4.1 and 4.2 are rather higher level, in the sense that the assumptions are directly imposed on the learned low-rank representations, rather than the observed data, the architecture structure of model, and the underlying functional space.

3. Tuning Parameter Selection and Sensitivity: (1) Two key areas regarding the method’s robustness would benefit from further discussion: The Truncation Dimension ($d$): This parameter is central to the balance between theory and practice. How did you settle on the specific values used in the experiments, and is there a recommended heuristic for practitioners? (2) Hyperparameters: Beyond the dimension $d$, how robust is the model to variations in regularization strength and optimization settings? A brief sensitivity analysis would significantly strengthen the practical utility of the work.

4. Computational Trade-offs: A primary motivation for this work is the scalability limitation of kernel-based tests. However, the bi-level optimization and whitening steps introduce their own overhead. Could you provide a more granular breakdown of the computational costs compared to standard kernel CI tests? This would help clarify the regimes where SpectralCIT offers the greatest practical advantage.

5. Some details are not included in the main paper, making it difficult to understand the theoretical results and applicable settings. For example, what are the dimensions of $X$, $Y$, and $Z$? Can $X$, $Y$, and $Z$ be of arbitrary dimensions? What are the assumptions on the approximation order $d$?

6. The current presentation of Figure 2 makes it difficult to verify the paper’s core claims: Type I Error (Left Panel): With the y-axis set to $[0, 1]$, the $0.05$ significance level is compressed at the bottom, making it impossible to tell if the test is well-calibrated or overly conservative (near zero). Please provide a zoomed-in plot (e.g., $[0, 0.1]$) and clarify if the observed error is statistically indistinguishable from $\alpha = 0.05$. Statistical Power (Right Panel): Most methods appear to hit a power of $1$ almost immediately. This "ceiling effect" hides any relative performance gains. Would it be possible to compare methods under a more challenging signal regime (e.g., weaker alternatives) where the methods exhibit visible separation?

7. Throughout the empirical section (Figures 2 and 3), results are reported as averages over 100 repetitions without uncertainty quantification. Please include standard errors or confidence intervals. Given that the paper’s primary contribution is rooted in statistical validity, demonstrating the stability and significance of these empirical results is essential.

**Limitations:**

Yes.

**Strengths And Weaknesses:**

Strengths:
1. Conditional independence testing is an important and challenging problem in statistical learning. The paper proposes a novel approach to address key challenges in conditional independence testing.
2. The paper develops a general framework for the proposed conditional independence test and derives the asymptotic distribution of the test statistic under the null hypothesis, as well as the asymptotic power under alternative hypotheses.
3. The work combines several existing techniques in a novel way to address the difficulties in conditional independence testing, which may provide useful insights for related problems.

Weaknesses:
1. Some theoretical assumptions are not clearly specified and appear somewhat vague. The conditions required for the theoretical results should be stated more precisely.
2. Several notations and conditions are not clearly presented (see questions below). In addition, some key concepts are insufficiently explained, which makes parts of the paper difficult to follow.
3. Some steps in the reasoning and derivations are not well explained, and additional clarification would help improve the readability and rigor of the presentation.
4. The implementation details could be more transparent. For example, the paper does not clearly describe hyperparameter selection or discuss the robustness of the proposed method to different parameter choices.
5. The empirical evaluation could be improved. In particular, uncertainty measures (e.g., standard errors) should be reported. In addition, the simulation settings could be better designed to provide more informative and meaningful comparisons with competing methods.

---

> ### Author Rebuttal · Authors · 2026-03-31
>
> We thank the reviewer for the thorough assessment of our work and for the insightful questions, which highlight key aspects of our method. These questions helped us better identify where the presentation can be improved, and we will revise the paper accordingly.
>
> Before we address comments by topic, note that additional plots are at https://anonymous.4open.science/r/SCIT-ICML-Rebuttal-2026-F07F/ICML_Rebuttal_2026_CIT.pdf. Also note additional benchmark in reply to CgHX.
>
> ## W1/Q1: approximation error $ℰ_m$
> We would like to emphasise that our goal in this paper was to **make explicit which properties of the learned representation are required for the CI test**, and **develop principled way to learn them**. To that end, instead of providing architecture-specific guarantees for the representation learning procedure, we decided to state condition on ℰ_m intentionally at a high level. Our asymptotic results are thus conditional on representation quality driven by the empirical loss minimisation that yields  $ℰ_m \to 0$ in probability as $m \to \infty$. The term $ℰ_m$ aggregates both approximation and estimation aspects: it reflects (i) how well a finite-dimensional representation approximates the operator, and (ii) how accurately this representation is learned from data. While providing explicit convergence rates for spectral representations remains an open problem in modern representation learning,  our contribution is to isolate how representation quality impacts validity and power. That said, we agree that this should be explained more clearly, and commit to do it in the revision.
>
> ## W1&3/Q2&5: assumptions, dimensions, and separation rate
> The assumptions are formulated at the level of the learned representation by design. Our method follows a two-stage structure: a representation learning phase followed by a testing phase. The CI problem itself is defined via $\Sigma_{X,\ddot{Y}\cdot Z}$, while Theorems 4.1–4.2 characterize the test conditional on the learned representation.
>
> This separation allows the analysis to remain agnostic to the choice of architecture while making explicit which properties of the learned features are required.
> - **Null validity (Thm 4.1)**: requires that the learned features form an approximately orthonormal system (via whitening) in the relevant $L^2$ spaces of $X, \ddot{Y}$, and $Z$. Under the null, a stable finite-dimensional basis suffices to ensure validity.
> - **Power (Thm 4.2)**: additionally requires that the signal in the leading components of the partial covariance operator is sufficiently strong. This is captured by the separation rate $\varepsilon_n$, which must dominate both statistical estimation error and the representation error $\mathcal{E}_m$.
>
> Thus, the **guarantees distinguish between a validity regime** (driven by representation normalization) and a **power regime** (driven by signal strength relative to approximation error).
>
> Importantly, $(X,Y,Z)$ may lie in arbitrary measurable spaces. The analysis depends on the intrinsic complexity of the distribution (via spectral decay), rather than on ambient dimension. The truncation level $d$ reflects this effective rank, and $\varepsilon_n$ corresponds to the minimal signal strength.
>
> ## W4/Q3&4: hyperparameters and computational trade-offs
> We would like to emphasise that we did substantial effort to design fair hyperparameter selection pipeline  (including dimension $d$) across all baselines, see App. C. That said, as requested be performed additional:
> - Sensitivity to $d$ (Fig. 2, rebuttal): for $d \geq 8$, Type I error is well-controlled and power remains competitive across all conditioning dimensions.
> - Sensitivity to learning rate (Fig. 3, rebuttal): multiplying η by α ∈ {0.1, 0.3, 1, 3, 10}, Type I error remains controlled for $α \geq 0.3$, with modest power variation at extreme values.
> - Sensitivity to regularization (Fig. 4, rebuttal): results are stable across all tested values.
>
> Further, we investigated computational complexity (Fig. 5, rebuttal): SpectralCIT is approx 2x faster than KCIT across all conditioning dimensions. This is expected as the additional cost of representation learning is amortized over test evaluation, and the overall complexity is dominated by low-rank operations in dimension $d$, rather than by kernel matrix computations in $n$. While, RCIT is faster, it unfortunately fails to maintain Type I error control in high dimensions (as shown in Figs. 6–7). **SpectralCIT offers the best trade-off between cost and validity**.
>
> ## W5/Q6&7: additional evaluations
> We provide zoomed-in Type I error plots (Fig. 6, rebuttal): SpectralCIT stays near or below $α=0.05$ across all dimensions, while NNLSCIT, GCIT, and RCIT all inflate substantially. Fig. 7 shows the new high-dimensional benchmark: SpectralCIT is the only method jointly maintaining valid Type I error control and non-trivial power across all conditioning dimensions. Confidence intervals and the revised paper will include these results.

---

> > ### Author Rebuttal · Reviewer_GEcD · 2026-04-01
> >
> > Thank you for addressing the concerns and comments. While I understand the responses regarding the technical conditions, they remain somewhat high-level and are not easy to verify in practice. I recognize that fully resolving these issues may require more time. So, I'll keep my original score.

---

> > > ### Author Response · Authors · 2026-04-07
> > >
> > > We thank the reviewer for the clarification. These conditions are natural and principled within the operator framework, as they correspond directly to the geometric, spectral, and signal-strength properties governing CI testing. We agree, however, that we have not sufficiently explained how to assess them in practice.
> > >
> > > In the revision, we will make explicit how these conditions relate to quantities that can be assessed empirically by separating them into validity and power components:
> > >
> > >  - **Validity (Type I error control)** requires that the learned features are approximately orthonormal, i.e., their empirical covariance operators are close to identity. This can be directly evaluated on held-out (validation) samples by measuring the conditioning of the empirical covariance matrices of the learned features. This property reflects alignment with the geometry of the underlying $L^2_X, L^2_{\ddot{Y}}, (L^2_Z$ spaces and is what guarantees the $\chi^2$  null distribution (Theorem 4.1).
> > >
> > > -  **Power** is governed by the signal strength $\epsilon_n$ (Eq. 9) and the separation condition in Theorem 4.2. In practice, signal strength and approximation quality can be assessed on validation samples by comparing the magnitude of the empirical test statistic to the statistical estimation error (e.g., scaling as $d^2/n$). This provides a direct way to evaluate whether the signal dominates the relevant error terms.
> > >
> > > For the representation learning phase, selecting the truncation level $d$ based on spectral decay (as in deep CCA and related methods) is a natural heuristic. While this approach has been effective in other spectral representation learning settings, we found that it is not yet fully reliable in our current bi-level formulation. We have identified concrete directions to improve the optimization procedure, and expect that these will make such spectral heuristics more stable in this setting.
> > > We will incorporate these diagnostics and guidelines into the revision so that the theoretical conditions are connected to concrete quantities that can be monitored in practice.

---

### Official Review · Reviewer_q36H · 2026-03-13

**Soundness:** 3
**Presentation:** 2
**Significance:** 3
**Originality:** 3
**Overall Recommendation:** 4
**Confidence:** 3

**Summary:**

The paper proposes a new approach for conditional independence (CI) testing based on the singular value decomposition of the partial covariance operator. The authors learn representations that approximate the operator and construct a test statistic based on them. To obtain the representations, they introduce a bi-level contrastive learning algorithm. Experiments on both synthetic and real datasets suggest that the method achieves good Type I error control while maintaining strong test power.

**Compliance With Llm Reviewing Policy:**

Affirmed.

**Final Justification:**

The paper proposes a new approach to conditional independence testing. Its main strength is that the method is potentially more applicable in high-dimensional settings than the kernel-based approach, although kernel methods could also be combined with representation learning. The rebuttal addressed my main concerns, and I now find the contribution technically sound and interesting. My remaining reservation is mainly about presentation: the method, theory, and its connection to regression-based approaches could be explained more clearly. I also feel that the claims about validity control and the advantages over regression-based methods are somewhat overstated and should be presented more carefully. Overall, however, I believe the paper would make a useful contribution to the field, and I recommend acceptance.

**Key Questions For Authors:**

1. Why is the partial cross-covariance operator defined as $\Sigma_{AB} - \Sigma_{AC}\Sigma_{CB}$ instead of $\Sigma_{AB} - \Sigma_{AC}\Sigma_{CC}^{-1}\Sigma_{CB}$ as in Strobl et al. (2019)? How does this modification affect the interpretation of the operator and the resulting test statistic? Does $A ⊥ B\mid C$ iff $∥ΣA(B,C)·C ∥_{HS} = 0$ still hold?
2. Could the authors clarify the claim that this null validity requirement is simpler than the conditional mean embedding estimation required by regression-based tests? Conceptually, the low-rank auxiliary problem still appears to involve estimating components predictable from Z.
3. It would be helpful to evaluate the method on the synthetic benchmark introduced by He et al. (2025), with single- and multi-dimensional conditioning variables. This dataset is low-dimensional but known to be very challenging for CI tests, especially for maintaining Type I error control.

**Limitations:**

yes

**Strengths And Weaknesses:**

Strengths:

The paper proposes a novel approach to approximate the partial cross-covariance operator for CI testing. The method relies on the assumption that the partial cross-covariance operator is compact, which allows it to be approximated arbitrarily well by a sufficiently large finite-rank empirical operator. This formulation avoids the use of kernels and operates in finite-dimensional representations, which may improve computational efficiency and scalability to large-scale or high-dimensional data. Empirical results show good Type I error control and competitive power on the evaluated benchmarks. Overall, the approach is interesting and may open a new direction for CI testing methods based on learned representations.

Weaknesses:

While the overall approach is interesting, the theoretical development and methodological details are not always easy to follow, making it difficult to fully assess the proposed framework.

One unclear point is the definition of the partial cross-covariance operator. The paper defines
$\Sigma_{AB\cdot C} = \Sigma_{AB} - \Sigma_{AC}\Sigma_{CB}$,
whereas the cited reference (Strobl et al., 2019) defines
$\Sigma_{AB\cdot C} = \Sigma_{AB} - \Sigma_{AC}\Sigma_{CC}^{-1}\Sigma_{CB}$,
where $\Sigma_{CC}^{-1}\Sigma_{CB}$ corresponds to the conditional expectation operator $\mathbb{E}_{B\mid C}[\cdot]$.

The paper does not explain why the inverse term is omitted. This distinction is important, as it directly affects Equation (5) through the $\Sigma_{XZ}\Sigma_{ZY}$ term and the associated low-rank auxiliary problem. Since this operator is central to the proposed method, a clear justification or derivation is necessary; as currently presented, this point is unclear and potentially problematic.

The paper also claims that the proposed requirement is simpler than the uniform conditional mean embedding rates needed in regression-based tests. However, the operator formulation does not eliminate the conditional expectation; it merely hides it inside operator notation. Note that the partial cross-covariance operator can be written as
$\mathbb{E}[(X - \mathbb{E}[X|Z])(Y - \mathbb{E}[Y|Z])]$ (footnote 3, Strobl et al. (2019)). Although the conditional expectation is not written explicitly in the operator derivation, conceptually it still plays the same role: removing the component explained by Z. The regression interpretation therefore remains valid. In practice, the low-rank auxiliary problem should capture the parts of X and Y that are predictable from Z, and the learned representations $C_{WU}, C_{VW}$ need to implicitly approximate quantities close to
$\mathbb{E}[U(X)\mid Z], \mathbb{E}[V(\ddot{Y})\mid Z]$.
From this perspective, the method still resembles regression-based tests, so the claim that it avoids such requirements may be overstated.

Besides, some key steps rely heavily on prior work (e.g., Kostic et al., 2024), but several details are omitted, making it challenging to understand for readers who are not familiar with the refered work.
For example, in Section 2 (“Spectral representation learning”), the authors state that Kostic et al. (2024) minimize a regularized loss to learn $\Sigma_{AB}$ using two neural networks, but the paper does not explain how the outputs of these networks are related to $\Sigma_{AB}$. The referred details in Section 6 mainly provide high-level context rather than derivation details.
Equation (5) also appears to hold only when u and v are centered, but this assumption is introduced only in the appendix rather than in the main text. And I think the derivation of the losses in the appendix should be presented in more detail.

---

> ### Author Rebuttal · Authors · 2026-03-31
>
> We thank the reviewer for their thoughtful assessment and for recognizing the novelty of the proposed approach and its potential to open a new direction for CI testing based on learned representations.
>
> ## Q1 Definition of the partial cross-covariance operator
> We thank the reviewer for raising this point. The difference comes from the ambient function spaces in which the operator is defined. In contrast to the RKHS formulation of Strobl et al. (2019) and related kernel works, we define the partial cross-covariance operator directly between $L^2$ spaces. In this setting, the covariance operator $\Sigma_{ZZ}$ acts as the identity on the codimension-1 subspace of centered functions and no explicit inverse term is needed.
>
> However, note that both operators share the same null space in RKHS, hence, they vanish iff $X \perp Y \mid Z$ and the characterization of conditional independence is preserved. We will clarify this in the revision.
>
> ## Q2 Comparison with regression-based approaches
> We agree that the auxiliary problem involves components predictable from $Z$, and that a connection with regression exists. The key distinction lies in the requirement for validity.
>
> In regression-based CI tests (e.g., GCM; Shah and Peters, 2020), validity depends on accurate estimation of conditional expectations, requiring $L^2$-consistent estimation of $\mathbb{E}[X \mid Z]$ and $\mathbb{E}[Y \mid Z]$ at rate at least $o_P(n^{-1})$.
>
> In contrast, our framework does not require estimating conditional expectations at any prescribed rate. Theorem 4.1 only requires approximate orthonormality of the learned features. Under the null, it suffices to learn a suitable basis of the relevant function spaces; recovering full conditional expectations is not necessary. Approximation of the partial covariance operator affects only power (Theorem 4.2).
>
> The proof in Appendix B is correct; the issue stems from presentation. For brevity, we grouped two distinct quantities—one governing validity and one governing power—into a single error term $\mathcal{E}_m$, obscuring this distinction. We will separate these terms explicitly: a validity component (orthonormality gaps) and a power component (operator approximation error).
>
> More broadly, regression-based methods reduce CI testing to estimation and inherit its guarantees. Since testing can often be performed at faster rates than estimation (Ingster, 1993), this can be restrictive. Our approach instead relies on eigenspace approximation without reducing CI testing to full regression. We will clarify this in Section 4.
>
> - Ingster, Yu. I. (1993). Asymptotically minimax hypothesis testing for nonparametric alternatives I–III. Mathematical Methods of Statistics, 2(2–4).
>
> ## Q3 Benchmark of He et al. (2025)
>
> We thank the reviewer for this suggestion and agree that this benchmark provides a valuable stress test for Type I error control.
>
> This synthetic model is challenging because dependence varies with $z$ but largely cancels on average, making it difficult to distinguish from conditional independence. As expected, this setting is pathological for standard CI tests: since dependence cancels globally, methods based on global discrepancies—including ours and those in the original benchmark—fail to control Type I error. The method of He et al. performs well on this specific construction, consistent with its design.
>
> However, on our new high-dimensional benchmark (Fig. 1, anonymised repository), which we describe in our answer to Reviewer CgHX, their method loses Type I error control for $\dim(Z) \geq 150$, while SpectralCIT maintains validity and competitive power.
>
> We will integrate these additional experiments in the final version to provide a more complete and balanced evaluation.
>
> Our goal is to develop a CI test that remains valid across a broad range of high-dimensional settings. The benchmark of He et al. highlights a specific regime where dependence is structured and cancels globally.
>
> Overall, our results demonstrate that SpectralCIT achieves reliable Type I error control and competitive power across a wide range of high-dimensional settings, which is the primary objective of this work.
>
> The anonymised repository:
> https://anonymous.4open.science/r/SCIT-ICML-Rebuttal-2026-F07F/ICML_Rebuttal_2026_CIT.pdf
>
> ## Clarity
> We thank the reviewer for these suggestions. We will improve readability by clarifying the definition of the partial covariance operator, explicitly stating centering assumptions (e.g., Eq. (5)), providing additional intuition on the spectral objective, and moving selected derivations into the main text.
>
> We would be grateful if the reviewer might consider increasing their score should they find this response satisfactory.

---

> > ### Author Rebuttal · Reviewer_q36H · 2026-04-01
> >
> > Thank the authors for clarifying my understanding of the partial cross-covariance operator. It is indeed consistent with Strobl et al. (2019). The discussion comparing it with regression-based tests also resolved my earlier concern.
> >
> > I am still confused by the claim on the benchmark of He et al. (2025):
> > “This synthetic model is challenging because dependence varies with z but largely cancels on average, making it difficult to distinguish from conditional independence. As expected, this setting is pathological for standard CI tests: since dependence cancels globally, methods based on global discrepancies—including ours and those in the original benchmark—fail to control Type I error.”
> >
> > My understanding is that, in this setting, global cancellation of dependence should mainly cause methods based on global discrepancy, such as GCM, to lose power. For regression-based methods, inflated Type I error would instead seem to arise from spurious dependence induced by poor estimation of the conditional expectations.
> >
> > By contrast, if your statistic is based on the partial cross-covariance operator, which satisfies
> > $||\Sigma_{A(B,C)\cdot C}||_{HS}=0 \text{ if and only if } A \perp B \mid C$,
> > then it should still be sensitive to local conditional dependence, and its failure mode should therefore differ from that of GCM.
> >
> > Moreover, under the null, the data are simply of the form
> > $A=f(C)+\varepsilon_A, B=g(C)+\varepsilon_B$,
> > where $\varepsilon_A$ and $\varepsilon_B$ are independent Gaussian noises. In this relatively simple low-dimensional setting, it seems more likely that the issue arises from the learned representation or the estimation procedure, rather than from the global cancellation structure of the alternative itself. So my question is whether the method is less suitable for low-dimensional data, or whether there is another explanation.
> >
> > I would be willing to raise my score if the authors could further clarify this point.

---

> > > ### Author Response · Authors · 2026-04-07
> > >
> > > # On the He et al. (2025) benchmark — Follow-up
> > >
> > > We thank the reviewer for this clarification and we agree.
> > >
> > > We ran SpectralCIT on the He et al. benchmark at $d_z = 3$. With the same representation-learning and regularization settings used for our high-dimensional benchmark, we were not able to obtain both strong power and satisfactory calibration in this low-dimensional weak-signal setting. After retraining representations for this regime, we recover Type I error $0.00 \pm 0.00$ and power $0.34 \pm 0.12$ (mean $\pm$ standard deviation across runs). This suggests that the difficulty lies not in the operator formulation itself, but in the sensitivity of the learned representation to hyperparameters in this regime.
> > >
> > > We also argue that the limited power is not due to low dimensionality per se, but to weak signal strength under the alternative. This interpretation is consistent with Theorem 4.2: power is guaranteed when the signal strength (leading singular values of the partial covariance operator) dominates the representation error. In the He et al. setting, the conditional dependence is deliberately weak, so this condition may not be met with the current representation-learning procedure.
> > >
> > > To test this hypothesis directly, we constructed a low-dimensional model ($d_z = 3$) with controlled signal strength:
> > >
> > > $$\mathcal{H}0:\quad X=\sin(Z+\varepsilon_X),\qquad Y=\cos(Z+\varepsilon_Y),$$
> > >
> > > $$\mathcal{H}1:\quad X=\sin(Z+\varepsilon_X)+\eta,\qquad Y=\cos(Z+\varepsilon_Y)+\eta,$$
> > >
> > > where $Z \sim \mathcal{N}(0, \sigma_Z^2 I_{d_z})$, $\varepsilon_X, \varepsilon_Y \sim \mathcal{N}(0, \sigma_{\text{noise}}^2 I_{d_z})$, and $\eta \sim \mathcal{N}(0, \sigma_{\eta}^2 I_{d_z})$ are independent.
> > >
> > > We vary $\sigma_{\eta} \in [0.05, 0.15, 0.5]$ with $\sigma_Z = 0.1$ and $\sigma_{\text{noise}} = 0.25$. Results are shown in Fig. 9: as signal strength increases, power increases from 0.10 to 0.75 to 1.00, while Type I error remains controlled at or below 0.05. This is fully consistent with Theorem 4.2 and confirms that the difficulty in the He et al.\ benchmark is weak signal rather than low dimensionality.
> > >
> > > We see improving the representation-learning phase to reliably capture weaker conditional dependence structures as an interesting direction for future work.
> > >
> > > New experiment (Fig. 9) is available at: https://anonymous.4open.science/r/SCIT-ICML-Rebuttal-2026-F07F/ICML_Rebuttal_2026_CIT.pdf
> > >
> > > We hope this clarification, together with the additional experiment, addresses the reviewer's concern, and we would be grateful if they would consider raising their score.

---

### Official Review · Reviewer_anoW · 2026-03-13

**Soundness:** 2
**Presentation:** 2
**Significance:** 3
**Originality:** 3
**Overall Recommendation:** 4
**Confidence:** 3

**Summary:**

The paper introduces a scalable conditional-independence test that learns low-dimensional neural representations to approximate the leading spectral components of a partial covariance operator for $(X,Y,Z)$. A bi-level contrastive/whitening training procedure yields features for $X$, $(Y,Z)$, and $Z$, after which the test uses a simple low-dimensional statistic $n\|\widehat C_{UV}-\widehat C_{UW}\widehat C_{WV}\|_F^2$ with asymptotic guarantee under vanishing representation error. Experiments show strong Type I control and good power with large conditioning sets.

**Compliance With Llm Reviewing Policy:**

Affirmed.

**Final Justification:**

Despite remaining concerns (notably conservative finite-sample calibration and theory that depends on an unquantified representation-learning error), the approach is technically strong and tackles an important scalability bottleneck in CI testing. The rebuttal and added causal-graph simulation help clarify relevance to causal discovery; however, without comparisons to alternative CI tests within PC/FCI (or a clear explanation of why such comparisons were infeasible), this additional evidence is limited.

**Key Questions For Authors:**

1. The experimental results show that the observed Type I error is significantly below the nominal level $\alpha$ (i.e., being overly conservative). Do the authors agree that this gap implies a potential loss of statistical power? Furthermore, could the authors discuss any practical calibration methods (e.g., specific bootstrapping or asymptotic adjustments) to bring the empirical Type I error closer to $\alpha$ to fully leverage the power of the proposed test?
2. The paper motivates the research through the lens of causal discovery, yet the connection remains largely conceptual. There is a lack of concrete discussion or experiments showing how the proposed CI test integrates into standard constraint-based pipelines (e.g., PC, FCI) or addresses specific bottlenecks like high-dimensional conditioning sets in those algorithms. If the authors view this as a causality contribution, could they provide a downstream experiment or a detailed discussion on its impact on causal structure learning? Otherwise, the work currently reads as a pure statistical CI-testing paper, for which 'Statistics' might be a more technically accurate framing for its core contribution.

**Limitations:**

The paper would benefit from an explicit limitations/impact discussion, e.g., discussing potential misuse in high-stakes domains (e.g., biomedical or social settings) where a CI “non-rejection” could be over-interpreted and lead to unreliable decisions.

**Strengths And Weaknesses:**

**Strength:**

The  paper use an interesting spectural representation learning tool to tackle conditional independence test problems, which is original. The problem of unscalability is rather urgent, so I acknowledge the significance of this work. The listed theoretical results seem solid.

**Weakness:**

1. **Theory relies on an unquantified representation-learning error.** The main validity/power guarantees are conditioned on the neural feature-learning error $E_m \to 0$, but the paper does not provide a concrete bound or convergence rate for learning $(u,v,w)$ under the proposed bi-level contrastive training. As a result, the theoretical claims are somewhat “if the network learns well, then…” and it remains unclear when/why this condition should hold in practice.
2. **Representation-learning theory appears heavily inherited from prior “spectral representation learning” work** (Kostic et al., 2025). The paper would benefit from a clearer delineation of what is genuinely new in the learning-theoretic derivations (beyond adapting the framework to the partial/conditional setting) versus what is essentially a direct reuse of existing arguments.
3. **Calibration appears conservative despite nominal $\alpha=0.05$.** Although the method is advertised as valid, the reported Type I error is often well below the target significance level (sometimes essentially zero, e.g., Fig. 3). This suggests the asymptotic $\chi^2(d^2)$ null leads to an overly conservative test. In turn, it is plausible that better calibration (bringing Type I error closer to 0.05) would yield noticeably higher power, so the current empirical gains may not reflect the method’s full potential and the comparison may depend on conservativeness rather than pure detection ability.
4. **Unclear presentation**. The paper is not self-contained, that means you have to have find other papers and try to get enough background knowledge to understand some part of it, which causes inconvenience for the readers. Even if the results come from previous works, please explain one sentence or two about it (or provide links to the corresponding appendix sections), especially when the results are one of the main parts of your work.

---

> ### Author Rebuttal · Authors · 2026-03-31
>
> We thank the reviewer for their thoughtful and constructive assessment of our work, and for recognizing both the originality of the proposed approach and the importance of addressing scalability in conditional independence (CI) testing. We address the reviewer’s concerns below.
>
> ## On representation learning error
> We agree that our guarantees are conditional on the quality of the learned representation, as captured by $\mathcal{E}_m$. Our goal is to make this dependence explicit and isolate precisely where it enters the analysis, rather than tie the theory to a specific architecture or optimization scheme. The term $\mathcal{E}_m$ aggregates both approximation and estimation aspects arising from the representation learning phase.
>
> Providing explicit non-asymptotic guarantees or convergence rates for such learned spectral representations is a challenging open problem, not specific to our method (see, e.g., [1]). We will clarify this point and better explain the role of $\mathcal{E}_m$ in the theoretical results.
>
> - [1] Meunier et al. Demystifying Spectral Feature Learning for Instrumental Variable Regression. NeurIPS 2025
>
> ## On novel theoretical contributions
>  A key conceptual contribution of this work is to show that CI testing can be approached without relying on accurate estimation of conditional expectations, by instead _grounding the test in eigenspace (spectral) approximation_.
>
> We agree that the distinction from prior spectral representation learning is not sufficiently clear and will make it more explicit in the revision.
>
> Prior work focuses on estimation or uncertainty quantification and does not address independence or conditional independence testing. Extending this framework to the conditional setting is nontrivial: the partial covariance operator involves residualization with respect to Z, which is not directly observable from data and must be handled implicitly. This fundamentally changes the representation learning problem and motivates the bilevel formulation in Section 3.
> As a result, both the representation learning phase and the downstream analysis differ.
>
> On the learning side, the objective must capture conditional structure through an auxiliary low-rank problem. Importantly, the representation is learned specifically for the testing task: rather than estimating the full operator, we recover a finite-dimensional spectral subspace adapted to CI testing. Under the null, learning approximately orthonormal basis of proper L2 spaces suffices for correct Type I error (Thm 4.1), while under alternatives, power depends on accurately capturing the leading components of the partial covariance operator (Thm 4.2).
>
> ## On conservative calibration and its impact on power (W3, Q1)
>  We agree that the empirical Type I error appears conservative in some settings, and that this may in principle reduce statistical power. CI testing is intrinsically challenging, and calibration under the null is known to be difficult in conditional settings (Shah and Peters, 2020), particularly in high-dimensional regimes or with complex conditional structure. Many existing approaches rely on regression, conditional mean embedding estimation, or resampling-based procedures (e.g., bootstrap, conditional permutations, or generative-model-based sampling), whose validity and performance can be sensitive to modeling assumptions and estimation error.
>
> Our approach instead works in a learned finite-dimensional representation, yielding a test statistic with a tractable asymptotic chi2 null and avoids repeated resampling or conditional simulation. The observed conservativeness can be understood as a finite-sample effect of combining an asymptotic reference distribution with learned representations. We view this as reflecting a trade-off between calibration sharpness and robustness. In the revision, we will make this trade-off more explicit. Zoomed-in Type I error plots and uncertainty measures are available in the anonymous repository. Please see our reply to Reviewer GEcD Q6.
>
> ## On the causal discovery framing (Q2)
> The reference to causal discovery is intended as an example: CI testing is a core component of methods such as PC and FCI and is often a major statistical and computational bottleneck, particularly with large conditioning sets or high-dimensional data (Shah and Peters. 2020; Glymour et al. 2019). Our primary contribution, however, is a conditional independence testing method. We do not study causal discovery pipelines directly, and we agree that evaluating integration into methods such as PC or FCI would be valuable. However, such an evaluation would require a dedicated study. We will clarify this scope more explicitly in the revision.
>
> - Glymour et al. Review of Causal Discovery Methods Based on Graphical Models. Sec. Statistical Genetics and Methodology, 2019
>
> We would be grateful if the reviewer might consider increasing their score should they find this response satisfactory.

---

> > ### Author Rebuttal · Reviewer_anoW · 2026-04-04
> >
> > Thank you for the detailed rebuttal. The responses partly address my concerns: the explanation for conservative calibration and the lack of explicit convergence rates for $E_m$ is reasonable, although these issues remain unresolved. However, my **presentation/clarity concerns (raised in the original review) were not addressed** in the rebuttal.
> >
> > **Follow-up question**: Given that large conditioning sets are a key bottleneck in constraint-based causal discovery (e.g., PC/FCI) due to the sheer number of CI tests, in which setting your approach sounds particularly promising, did you consider running a downstream PC experiment? Even a small-scale study would be informative. If not, could you clarify the main blocker (e.g., per-test runtime, whether training cost can be amortized across many tests, hyperparameterstability, or implementation complexity)?

---

> > > ### Author Response · Authors · 2026-04-06
> > >
> > > # On presentation and clarity
> > > We appreciate you highlighting the presentation issues. Due to rebuttal length limits, we did not fully address them earlier. We will revise the paper to make it more self-contained and reduce reliance on prior work by adding the necessary explanations. Related improvements were also discussed in our responses to Reviewers q36H and GEcD and will be incorporated into a unified revision:
> > >
> > > *Section 2 — Spectral representation learning.* The main text introduces the contrastive loss of Kostic et al. (2024) without explaining its connection to the SVD. We will add the following intuition inline: the loss is a U-statistic estimator of $\Vert \Sigma_{AB} - UMV^*\Vert^2_{HS}-\Vert\Sigma_{AB} \Vert^2_{HS}$, whose minimizer is the rank-$d$ truncated SVD by the Eckart–Young–Mirsky theorem. The formal statement is in Appendix A (Theorem A.1), and we will add a pointer from Section 2.
> > >
> > > *Section 3 — Bi-level formulation.* The need for the inner optimization problem is currently not well motivated. A reader unfamiliar with Kostic et al. (2024) may not see why a single-level loss is insufficient. We will clarify that the variational formulation of the truncated SVD of $\Sigma_{X\tilde{Y} \cdot Z}$ involves the term $\Sigma_{XZ}\Sigma_{Z\tilde{Y}}$, which cannot be directly estimated from data. The inner problem addresses this by introducing a low-rank operator $WNW^*$ that captures this composition implicitly. The full derivation, including the symmetrization step and resulting bi-level formulation, is given in Appendix A (eqs. 13 and 15), and we will add a pointer from Section 3.
> > >
> > > *Theorem 4.1 — Chi-squared null distribution.* The convergence to $\chi^2(d^2)$ is currently stated without intuition. We will add the following explanation inline: after whitening, the learned features are approximately orthonormal, so the empirical covariance matrix $\hat C_{\hat u \hat v} - \hat C_{\hat u \hat w} \hat C_{\hat w \hat v}$ has approximately identity covariance structure under the null. The test statistic $\hat{T}_n$ is then approximately a sum of $d^2$ squared independent standard Gaussians, which is a $\chi^2(d^2)$ random variable. The formal proof is in Appendix B.7, and we will add an explicit pointer.
> > >
> > > *Error term $\mathcal{E}_m$ — Validity vs Power.* As also noted by Reviewer q36H, the current presentation groups two conceptually distinct quantities into a single error term $\mathcal{E}_m$, obscuring the roles they play. We will separate these explicitly in the revision:
> > >
> > > - A validity component capturing orthonormality gaps like in this term $\Vert \hat{C}_{\hat{u}\hat{u}} - I_d \Vert$, which governs Type I error control (Theorem 4.1, proved in Appendix B.7)
> > > - A power component capturing operator approximation error $\Vert T_d(\Sigma_{X\tilde{Y}\cdot Z}) - U_\theta M_\theta V_\theta^* \Vert$ (where $ T_d$ is the rank d truncation), which governs detection under the alternative (Theorem 4.2, proved in Appendix B.9)
> > >
> > > This separation will be stated explicitly in Section 4 before the theorems, making clear that validity and power depend on different aspects of the learned representation.
> > >
> > > # On the PC experiment
> > >
> > > We added a small downstream study on the 7-node Sachs-inspired DAG shown in Fig. 8 and Table 1 of the anonymous repository ($Z_1$ hub, post-nonlinear structural equations as in Fig. 2). We evaluate three CI queries representative of those a constraint-based algorithm would encounter, with conditioning sets of size 2–4, under both $\mathcal{H}_0$ and $\mathcal{H}_1$. Under $\mathcal{H}_1$, we plant the direct edge $X_1 \to Y_2$. Results are in Table 1. Type I error is 0.00 on all three queries. For $X_1 \perp Y_2 \mid X_3, X_4$ and $X_1 \perp Y_2 \mid X_3, X_4, Z_1$, the planted edge renders $\mathcal{H}_1$ non-trivial: power is 1.00 and 0.75 respectively, the latter at $d_z = 100$ where adding the hub to the conditioning set makes detection harder. For $Y_1 \perp Y_2 \mid X_1, X_2, X_3, X_4$, the planted edge does not alter the null, so $\mathcal{H}_1$ coincides with $\mathcal{H}_0$ and only Type I error is relevant. While this is not yet a full PC/FCI benchmark, these results support the relevance of SpectralCIT to constraint-based causal discovery, particularly in the large-conditioning-set regime.
> > >
> > > The anonymised repository:
> > > https://anonymous.4open.science/r/SCIT-ICML-Rebuttal-2026-F07F/ICML_Rebuttal_2026_CIT.pdf
> > >
> > > We hope these revisions address your concern that the paper relies on prior familiarity. They will make the paper self-contained for readers new to spectral representation learning without significantly increasing the length.
> > >
> > > We would be grateful if the reviewer might consider increasing their score in light of these commitments and the PC experiment results.

---

### Decision · Program_Chairs · 2026-04-30

**Decision:**

Accept (regular)

**Comment:**

The submission is recommended for Acceptance because all reviewers were impressed by its "conceptually clean" way of mixing spectral representation learning with operator-theoretic CI testing. This approach is a big step forward because it fixes the scalability issues of kernel methods while keeping Type I error control solid, even when conditioning dimensions get very high. While the theory is a bit high-level, the authors really proved the method's worth in the rebuttal by adding a tough high-dimensional benchmark, showing it isn't too sensitive to hyperparameters, and even running a downstream causal study.